# Nonconvex-nonconcave min-max optimization on Riemannian manifolds

**Andi Han**                                                   *andi.han@sydney.edu.au*
*University of Sydney*

**Bamdev Mishra**                                              *bamdevm@microsoft.com*
*Microsoft India*

**Pratik Jawanpuria**                                 *pratik.jawanpuria@microsoft.com*
*Microsoft India*

**Junbin Gao**                                              *junbin.gao@sydney.edu.au*
*University of Sydney*

**Reviewed on OpenReview:** *https://openreview.net/forum?id=EDVIHPZhFo*

## Abstract

This work studies nonconvex-nonconcave min-max problems on Riemannian manifolds. We first characterize the local optimality of nonconvex-nonconcave problems on manifolds with a generalized notion of local minimax points. We then define the stability and convergence criteria of dynamical systems on manifolds and provide necessary and sufficient conditions of strictly stable equilibrium points for both continuous and discrete dynamics. Additionally, we propose several novel second-order methods on manifolds that provably converge to local minimax points asymptotically. We validate the empirical benefits of the proposed methods with extensive experiments.

## 1 Introduction

Min-max optimization has been the central focus in a variety of machine learning applications, including generative adversarial networks (GANs) (Goodfellow et al., 2020), adversarial training (Madry et al., 2017), optimal transport (Lin et al., 2020a; Jawanpuria et al., 2021), low-rank matrix/tensor completion (Jawanpuria & Mishra, 2018; Nimishakavi et al., 2018), robust learning (El Ghaoui & Lebret, 1997), non-linear feature learning (Rakotomamonjy et al., 2008; Aflalo et al., 2011; Jawanpuria et al., 2011; 2015b), online learning (Bubeck et al., 2012), multi-task learning (Jawanpuria & Nath, 2012; Jawanpuria et al., 2015a), fair inference (Madras et al., 2018), and reinforcement learning (Busoniu et al., 2008). When the objective function is convex-concave, the well-known minimax theorem (Neumann, 1928; Sion, 1958) guarantees the existence of a global (Nash) saddle point and there exist algorithms that converge to such global solutions (Tseng, 1995; Nemirovski, 2004; Mokhtari et al., 2020a).

Despite the noteworthy progress in solving convex-concave min-max problems, many real-life applications do not have the convex-concave structure. For general nonconvex-nonconcave min-max problems, even the proper notions of local optimality are critically debated. Under such settings, the order of optimization matters, e.g., in GANs and adversarial training, and this renders the notion of Nash saddle points improperly. The recent work by Jin et al. (2020) introduces the notion of (local) minimax points, which better suits the sequential nature of the problems and bears a close relation to the stable limiting points of the gradient descent ascent dynamics (with timescale separation).

Another research direction on nonconvex-nonconcave optimization is to formulate min-max problems on Riemannian manifolds (Zhang et al., 2022c; Jordan et al., 2022; Han et al., 2023b; Huang & Gao, 2023). The

generalized Sion's minimax theorem (Zhang et al., 2022c) allows many intractable nonconvex-nonconcave problems to be solved efficiently on Riemannian manifolds, provided the objectives satisfy a generalized notion of convexity and concavity along geodesics on manifolds. Nevertheless, there exists several problems on manifolds without such generalized convexity or concavity structures, e.g., the robust Fréchet mean (Zhang et al., 2022c), projection robust Wasserstein distance (Lin et al., 2020a), and orthonormal GAN (Müller et al., 2019) problems, to name a few. It is, therefore, natural to ask:

*how to define local optimality for general nonconvex-nonconcave problems on Riemannian manifolds and what are algorithms that can converge to such local optimality?*

We attempt to address the above questions by generalizing the notion of (local) minimax points to Riemannian manifolds and studying the behaviors of various algorithms on manifolds relating to such optimal points. To this end, our contributions are as follows.

- We generalize the concepts of minimax points for defining local optimality for nonconvex-nonconcave min-max problems on Riemannian manifolds from the Euclidean space and explore various properties. We also illustrate examples of such points.

- We introduce notions of stability and convergence for both continuous and discrete dynamical systems on manifolds. We then show the relation between the limiting behaviors of the Riemannian gradient descent ascent method and the generalized minimax points.

- We propose several novel second-order methods and prove that they asymptotically converge to local minimax points.

- We demonstrate the efficacy of the proposed algorithms against existing baselines on a variety of nonconvex-nonconcave problems, including the robust Fréchet mean, robust maximum likelihood estimation, projection robust Wasserstein distance, and orthonormal GAN problems.

## 2 Related works

In the **Euclidean space**, most studies focus on (strongly)-convex-(strongly)-concave settings (Nemirovski, 2004; Mokhtari et al., 2020a;b; Golowich et al., 2020; Yoon & Ryu, 2021; Zhang et al., 2022b). However, without the convex-concave structure, the problem is in general intractable. Thus, efforts have been devoted to solving structured nonconvex-nonconcave problems, including assuming (strong)-concavity in the max-variable (Thekumparampil et al., 2019; Lin et al., 2020b;c), imposing variational inequality conditions on the min-max gradient operator (Song et al., 2020; Diakonikolas et al., 2021; Lee & Kim, 2021), and assuming a PL-like condition (Nouiehed et al., 2019; Abernethy et al., 2019). For general nonconvex-nonconcave settings, existing works have defined various notions of local optimality, such as local Nash saddle points (Daskalakis & Panageas, 2018; Adolphs et al., 2019; Mazumdar et al., 2020), and local minimax points (Jin et al., 2020; Zhang et al., 2022a), and have analyzed (local) convergence of algorithms to such local solutions (Daskalakis & Panageas, 2018; Wang et al., 2019; Zhang et al., 2020; Fiez et al., 2020; Gao et al., 2022; Zhang et al., 2022a). The work (Daskalakis et al., 2021) studies the complexity of identifying and solving constrained nonconvex nonconcave problems. Daskalakis et al. (2021), in particular, focus on approximate local Nash saddle points and linear convex constraints.

On **Riemannian manifolds**, min-max problems have attracted increasing attention. Zhang et al. (2022c) generalize Sion's minimax theorem to Riemannian manifolds and propose a Riemannian extra-gradient method (namely RCEG) for geodesic-convex-geodesic-concave problems. Jordan et al. (2022) complete the analysis of both Riemannian gradient descent ascent (RGDA) and RCEG for geodesic-(strongly)-convex-geodesic-(strongly)-concave settings. Han et al. (2023b) extend the Hamiltonian gradient methods to Riemannian min-max optimization and show convergence under the Riemannian PL condition on a proxy function. Huang & Gao (2023); Wu et al. (2023) consider the nonconvex-(strongly)-concave min-max problems on manifolds.

## 3 Preliminaries and notations

This section provides a summary of concepts we require in this work with a focus on notations. We defer the detailed introduction to Appendix B.

A Riemannian manifold $\mathcal{M}$ is a manifold with a *Riemannian metric* on each tangent space $T_z\mathcal{M}$, $\forall z \in \mathcal{M}$, i.e., $\langle u, v \rangle_z$ for any $u, v \in T_z\mathcal{M}$. The induced norm is denoted as $\| \cdot \|_z$. We also use $T\mathcal{M}$ to denote the *tangent bundle*, the disjoint union of all tangent spaces. The *geodesics* $\gamma : [0, 1] \to \mathcal{M}$ generalizes the straight line segment, and the *exponential map* $\mathrm{Exp}_z : T_z\mathcal{M} \to \mathcal{M}$ generalizes the vector addition in Euclidean space. *Riemannian distance* $\mathrm{dist}(z_1, z_2)$ is the length of the shortest geodesic connecting $z_1, z_2 \in \mathcal{M}$. The (smooth) inverse of the exponential map (when exists), namely *the logarithm map*, is denoted as $\mathrm{Log}_{z_1} : \mathcal{M} \to T_{z_1}\mathcal{M}$. *Parallel transport* $\Gamma_{z_1}^{z_2} : T_{z_1}\mathcal{M} \to T_{z_2}\mathcal{M}$ transports tangent vectors between tangent spaces. The *covariant derivative* induced from the Riemannian connection is denoted as $\nabla$, which allows differentiation of vector fields on manifolds.

A *Riemannian product manifold* $\mathcal{M} = \mathcal{M}_x \times \mathcal{M}_y$ admits a Riemannian manifold structure with induced Riemannian metric from $\mathcal{M}_x, \mathcal{M}_y$. For a linear operator $H$ on $T_z\mathcal{M}$, we can decompose its operation on $\xi = (u, v) \in T_x\mathcal{M}_x \times T_y\mathcal{M}_y$ as $H[\xi] = \begin{pmatrix} H_{xx} & H_{xy} \\ H_{yx} & H_{yy} \end{pmatrix} \begin{pmatrix} u \\ v \end{pmatrix} = \begin{pmatrix} H_{xx}[u] + H_{xy}[v] \\ H_{yx}[u] + H_{yy}[v] \end{pmatrix}$ where $H_{xx} : T_x\mathcal{M}_x \to T_x\mathcal{M}_x, H_{yy} : T_y\mathcal{M}_y \to T_y\mathcal{M}_y, H_{xy} : T_y\mathcal{M}_y \to T_x\mathcal{M}_x, H_{yx} : T_x\mathcal{M}_x \to T_y\mathcal{M}_y$ are linear operators. For a linear operator $G$ between tangent spaces, we denote $G^\dagger$ as its *adjoint operator*.

For a bifunction $f : \mathcal{M}_x \times \mathcal{M}_y \to \mathbb{R}$, we denote $\nabla_x f(x, y), \nabla_y f(x, y)$ as the *Euclidean (partial) gradient* and $\mathrm{grad}_x f(x, y), \mathrm{grad}_y f(x, y)$ as the *Riemannian (partial) gradient*. Furthermore, we let $\mathrm{Hess}_x f(x, y), \mathrm{Hess}_y f(x, y)$ be the Riemannian Hessian and $\mathrm{grad}_{xy}^2 f(x, y), \mathrm{grad}_{yx}^2 f(x, y)$ be the *Riemannian cross-derivatives* defined as $\mathrm{grad}_{xy}^2 f(x, y)[v] = \mathrm{D}_y \mathrm{grad}_x f(x, y)[v]$ for any $v \in T_y\mathcal{M}_y$ and similarly for $\mathrm{grad}_{yx}^2 f(x, y)$. *Geodesic (strong) convexity* is a generalized notion of convexity along geodesics on manifolds. A function is called geodesic linear if it is both geodesic convex and geodesic concave. A bifunction $f(x, y)$ is called geodesic (strongly) convex (strongly) concave if $f$ is geodesic (strongly) convex in $x$ and geodesic (strongly) concave in $y$.

**Other notations.** We use $a \wedge b := \min\{a, b\}$ and $a \vee b := \max\{a, b\}$ for $a, b \in \mathbb{R}$. We use $[n] := \{1, ..., n\}$ for $n \in \mathbb{Z}_+$, the set of positive integers. Suppose $\lambda_i, i \in [n]$ are real eigenvalues of a self-adjoint operator. We order the eigenvalues as $\lambda_1 \geq \lambda_2 \geq \cdots \geq \lambda_n$. We denote $\| \cdot \|_2, \| \cdot \|_\mathrm{F}$ as the Euclidean norm and the Frobenius norm, respectively. We denote $\mathrm{logm}(\cdot)$ as the principal matrix logarithm and define $\log \det(\cdot) := \log(\det(\cdot))$.

## 4 Minimax point for nonconvex-nonconcave min-max problems on manifolds

We consider the min-max problems on Riemannian manifolds as

$$\min_{x \in \mathcal{M}_x} \max_{y \in \mathcal{M}_y} f(x, y),$$

where $f$ is at least twice differentiable and in general nonconvex-nonconcave in both Euclidean and geodesic senses. Under such settings, the generalized Sion's minimax theorem does not hold (Zhang et al., 2022c), and the order of optimization matters. Without loss of generality, we assume minimization takes place first, followed by maximization.

Recently, Zhang et al. (2022c); Jordan et al. (2022) have studied global saddle points on manifolds. Below, the classic definitions of global and local *Nash saddle points* are generalized from the Euclidean space to general manifolds.

**Definition 1** (Global and local Nash saddle point)**.** A point $(x^*, y^*)$ is called a *global* Nash saddle point if, for any $(x, y) \in \mathcal{X} \times \mathcal{Y}$, we have $f(x^*, y) \leq f(x^*, y^*) \leq f(x, y^*)$. A point $(x^*, y^*)$ is called a *local* Nash saddle point if there exists $\epsilon > 0$ such that for any $(x, y)$ satisfying $\mathrm{dist}(x, x^*) \leq \epsilon, \mathrm{dist}(y, y^*) \leq \epsilon$, we have $f(x^*, y) \leq f(x^*, y^*) \leq f(x, y^*)$.

The conditions of optimality for local saddle points on manifolds are similar to those in Euclidean space.

**Proposition 1** (First-order and second-order conditions of local Nash saddle points). *If a point $(x^*, y^*)$ is a local Nash saddle point of a twice continuously differentiable function $f$, then it satisfies (1) $\mathrm{grad}_x f(x^*, y^*) = 0, \mathrm{grad}_y f(x^*, y^*) = 0$, and (2) $\mathrm{Hess}_x f(x^*, y^*) \succeq 0$ and $\mathrm{Hess}_y f(x^*, y^*) \preceq 0$.*

As shown in Jin et al. (2020), the Nash saddle point, be it local or global, may not exist in general for nonconvex-nonconcave functions. Hence, Jin et al. (2020) have proposed the notion of global and local *minimax points* in the Euclidean setting. In the following, we generalize it to Riemannian manifolds.

**Definition 2** (Global minimax point). Consider two subsets $\mathcal{X} \times \mathcal{Y} \subseteq \mathcal{M}_x \times \mathcal{M}_y$, $(x^*, y^*)$ is called a *global minimax point* of $f$ if, for any $(x, y) \in \mathcal{X} \times \mathcal{Y}$, it satisfies $f(x^*, y) \leq f(x^*, y^*) \leq \max_{y' \in \mathcal{Y}} f(x, y')$. Equivalently, $(x^*, y^*)$ satisfies $y^* = \arg\max_{y \in \mathcal{Y}} f(x^*, y)$ and $x^* = \arg\min_{x \in \mathcal{X}} \phi(x)$, where $\phi(x) := \max_{y \in \mathcal{Y}} f(x, y)$.

**Definition 3** (Local minimax point). A point $(x^*, y^*)$ is called a *local minimax point* of $f : \mathcal{X} \times \mathcal{Y} \to \mathbb{R}$ if there exists a constant $\delta_0 > 0$ and a function $h$ satisfying $\lim_{\delta \to 0} h(\delta) = 0$ for any $\delta \in (0, \delta_0]$ and any $(x, y) \in \mathcal{X} \times \mathcal{Y}$ satisfying $\mathrm{dist}(x, x^*) \leq \delta$, $\mathrm{dist}(y, y^*) \leq \delta$, we have $f(x^*, y) \leq f(x^*, y^*) \leq \max_{y' : \mathrm{dist}(y', y^*) \leq h(\delta)} f(x, y')$.

**Proposition 2** (Equivalent characterization of local minimaxity). *Suppose $f$ is continuous on $\mathcal{X} \times \mathcal{Y}$. A point $(x^*, y^*)$ is a local minimax point if and only if $y^*$ is a local maximum of function $f(x^*, y)$ and there exists a $\delta_0 > 0$ such that $x^*$ is a local minimum of $g_\delta(x) := \max_{y' : \mathrm{dist}(y', y^*) \leq \delta} f(x, y')$ for any $\delta \in (0, \delta_0]$.*

A game-theoretic intuition for minimax points is that the max-player always maximizes its payoff based on the observation of the min-player's action, while the min-player chooses an overall optimal strategy without knowing the action of the max-player. In particular, (Jin et al., 2020, Proposition 11) show that there always exists a global minimax point due to the extreme value theorem on a compact domain, unlike the case for Nash saddle points. However, the local minimax point may not exist. In addition, any local Nash saddle point is a local minimax point.

Next, we provide first-order and second-order characterizations of local minimaxity on general Riemannian manifolds, where we make use of the Riemannian Hessian and cross-derivative operators on tangent spaces.

**Proposition 3** (First-order and second-order conditions). *Suppose $f$ is continuous and at least twice differentiable and if $(x^*, y^*)$ is a local minimax point, then it satisfies $\mathrm{grad}_x f(x^*, y^*) = 0$, $\mathrm{grad}_y f(x^*, y^*) = 0$. In addition, $(x^*, y^*)$ satisfies $\mathrm{Hess}_y f(x^*, y^*) \preceq 0$. If further $\mathrm{Hess}_y f(x^*, y^*) \prec 0$, then $\mathrm{Hess}_x f(x^*, y^*) - \mathrm{grad}_{xy}^2 f(x^*, y^*) \circ [\mathrm{Hess}_y f(x^*, y^*)]^{-1} \circ \mathrm{grad}_{yx}^2 f(x^*, y^*) \succeq 0$.*

For completeness, we show the sufficient first-order and second-order conditions for local Nash and minimax saddle points in Proposition 15 (Appendix). We now discuss several examples to elucidate the concepts of local minimax points on Riemannian manifolds.

**Proposition 4** (Examples of local minimax points). *Let $\mathbb{S}_{++}^d := \{\mathbf{X} \in \mathbb{R}^{d \times d} : \mathbf{X}^\top = \mathbf{X}, \mathbf{X} \succ 0\}$ be the symmetric positive definite (SPD) manifold endowed with the affine-invariant metric (detailed in Appendix C.2). Consider $\mathbf{X}, \mathbf{Y} \in \mathbb{S}_{++}^d$ and define the set $S^* := \{\mathbf{A} \mid \mathbf{A} \in \mathbb{S}_{++}^d, \det(\mathbf{A}) = 1\}$.*

*(1) Any $(\mathbf{X}^*, \mathbf{Y}^*) \in S^* \times S^*$ is a global Nash saddle point and hence a local minimax point to function $f(\mathbf{X}, \mathbf{Y}) = c_1 (\log \det(\mathbf{X}))^2 + c_2 \log \det(\mathbf{X}) \log \det(\mathbf{Y}) - c_3 (\log \det(\mathbf{Y}))^2$. for any $c_1, c_2, c_3 > 0$.*

*(2) Any $(\mathbf{X}^*, \mathbf{Y}^*) \in S^* \times S^*$ is a local minimax point but **not** a local Nash saddle point to function $f(\mathbf{X}, \mathbf{Y}) = -c_1 (\log \det(\mathbf{X}))^2 + c_2 \log \det(\mathbf{X}) \log \det(\mathbf{Y}) - c_3 (\log \det(\mathbf{Y}))^2$, for any $c_1, c_2, c_3 > 0$ such that $c_2 \geq 2\sqrt{c_1 c_3}$.*

*(3) Any $(\mathbf{X}^*, \mathbf{Y}^*) \in S^* \times \{\mathbf{I}\}$ is a local minimax point but **not** a local Nash saddle point to function $f(\mathbf{X}, \mathbf{Y}) = -c_1 (\log \det(\mathbf{X}))^2 + c_2 \log \det(\mathbf{X}) \log \det(\mathbf{Y}) - c_3 \|\mathrm{logm}(\mathbf{Y})\|_{\mathrm{F}}^2$ for any $c_1, c_2, c_3 > 0$ such that $c_2 \geq \frac{2\sqrt{c_1 c_3}}{\sqrt{d}}$.*

Proposition 4 makes use of "quadratic" functions $(\log \det(\cdot))^2$ and $\|\mathrm{logm}(\mathbf{Y})\|_{\mathrm{F}}^2$ on the SPD manifold. $(\log \det(\cdot))^2$ is quadratic due to geodesic linearity of $\log \det(\cdot)$ (Vishnoi, 2018). $\|\mathrm{logm}(\mathbf{Y})\|_{\mathrm{F}}^2$ is equivalent to $\mathrm{dist}^2(\mathbf{I}, \mathbf{Y})$, i.e., the squared Riemannian distance. Hence, the functions are analogs of quadratic functions in Euclidean space. However, we highlight that the example functions of the form (2) in Proposition 4 are geodesic concave but not necessarily strongly concave in $\mathbf{Y}$, unlike the quadratic functions in Euclidean space. A consequence is that the local minimax point is not strict and there exists a continuum of such points.

# 5 Stability and convergence of dynamics on manifolds

In this section, we introduce the notions of stability and asymptotic convergence rate of both continuous and discrete dynamics on manifolds, which are essential for subsequent convergence analysis around minimax points. Given the analysis is local around an equilibrium point, we make the following standard assumption.

**Assumption 1.** Dynamics considered in this paper are contained in a neighborhood of an equilibrium point where the exponential map has a smooth inverse.

Such an assumption allows the linearization of arbitrary dynamics on manifolds around the equilibrium point, in a similar vein as the linearization of nonlinear dynamics in Euclidean space.

## 5.1 Stability of continuous dynamics

We consider a continuous autonomous dynamical system $\dot{z}(t) = G(z(t))$ on a Riemannian manifold $\mathcal{M}$, where $z(t) \in \mathcal{M}$, $\dot{z}(t) \in T_{z(t)}\mathcal{M}$ represents the time-derivatives, and $G : \mathcal{M} \to T\mathcal{M}$ is a vector field. We say $z^*$ is an equilibrium point (or fixed point) if it satisfies $G(z^*) = 0$.

In Euclidean space, i.e., $z(t) \in \mathbb{R}^d$, the stability of the dynamical system can be characterized via the (real-parts) of Jacobian eigenvalues of $G(z^*)$ at equilibrium (Khalil, 2002). Nevertheless, on general Riemannian manifolds, the derivative of $G(z^*)$ is not necessarily self-adjoint, and its eigenvalue/eigenvector may not exist due to the lack of well-defined complex eigenvalue/eigenvector on tangent spaces.

Here, we approach the notion of stability via linearization of the dynamics and Lyapunov analysis.

**Proposition 5.** *Consider a dynamics $\dot{z}(t) = G(z(t))$ on $\mathcal{M}$, with an equilibrium point $z^*$. Let $V(z(t)) \coloneqq \langle \mathrm{Log}_{z^*}(z(t)), H[\mathrm{Log}_{z^*}(z(t))]\rangle_{z^*}$ be a Lyapunov function for some self-adjoint positive definite operator $H$ on $T_{z^*}\mathcal{M}$. Then, we have $\frac{d}{dt}\mathrm{Log}_{z^*}(z(t)) = \boldsymbol{\nabla}G(z^*)[\mathrm{Log}_{z^*}(z(t))] + o(\|\mathrm{Log}_{z^*}(z(t))\|_{z^*})$ and $\frac{d}{dt}V(z(t)) = \langle \mathrm{Log}_{z^*}(z(t)), \left(H \circ \boldsymbol{\nabla}G(z^*) + (\boldsymbol{\nabla}G(z^*))^\dagger \circ H\right)[\mathrm{Log}_{z^*}(z(t))]\rangle_{z^*} + o(\|\mathrm{Log}_{z^*}(z(t))\|_{z^*}^2)$.*

From Proposition 5, we see any dynamical system can be linearized around a fixed point, which is sufficient for the asymptotic analysis in subsequent sections. In addition, $\frac{d}{dt}V(z(t)) < 0$ if and only if $H \circ \boldsymbol{\nabla}G(z^*) + (\boldsymbol{\nabla}G(z^*))^\dagger \circ H$ is negative definite for some positive definite operator $H$. This leads to the notion of equilibrium stability on manifolds, which generalizes the classic notion of Lyapunov (asymptotic) stability in Euclidean space.

**Definition 4** (Strictly stable equilibrium)**.** Consider any (autonomous) dynamical system on a Riemannian manifold $\mathcal{M}$ as $\dot{z}(t) = G(z(t))$. An equilibrium point $z^*$ is called strictly stable if and only if $H \circ \boldsymbol{\nabla}G(z^*) + (\boldsymbol{\nabla}G(z^*))^\dagger \circ H$ is negative definite for some positive definite self-adjoint operator $H$ on $T_{z^*}\mathcal{M}$.

We next show that a strictly stable equilibrium can be equivalently characterized via the eigenvalues of a matrix representation of $H \circ \boldsymbol{\nabla}G(z^*) + (\boldsymbol{\nabla}G(z^*))^\dagger \circ H$. The matrix representation is obtained with respect to an orthonormal basis on tangent space. In this paper, this is also followed for any linear operator on a tangent space.

**Proposition 6.** *For a dynamical system $\dot{z}(t) = G(z(t))$ on a manifold $\mathcal{M}$, an equilibrium $z^*$ is strictly stable if and only if the matrix representation of $\boldsymbol{\nabla}G(z^*)$ in an orthonormal basis of $T_{z^*}\mathcal{M}$ has all real parts of eigenvalues negative.*

**Remark 1.** Proposition 6 is mainly due to the energy preservation under an orthonormal coordinate transform. We highlight that this result is independent of the choice of orthonormal basis because, as shown in the proof, the change-of-basis transformation forms an isomorphism between the dynamics under different coordinate systems and the stability can be studied under any basis.

## 5.2 Stability of discrete dynamics

In this section, we consider the asymptotic stability of a discrete dynamical system on manifolds, namely $z_{k+1} = \mathrm{Exp}_{z_k}(\eta\, G(z_k))$ for some stepsize $\eta > 0$ with a fixed point $z^*$ satisfying $G(z^*) = 0$. This corresponds to the geometric Euler discretization of the continuous dynamics $\dot{z}(t) = G(z(t))$ discussed in the previous

section. The following proposition shows that similar to the continuous case, the discrete dynamics is also locally linear around a fixed point.

**Proposition 7.** *Consider a discrete dynamical system on $\mathcal{M}$ as $z_{k+1} = \mathrm{Exp}_{z_k}(\eta\, G(z_k))$ for some $\eta > 0$. Define the Lyapunov function $V(z_k) := \langle \mathrm{Log}_{z^*}(z_k), H[\mathrm{Log}_{z^*}(z_k)]\rangle_{z^*}$ for some self-adjoint positive definite operator $H$ on $T_{z^*}\mathcal{M}$. Then $\mathrm{Log}_{z^*}(z_{k+1}) = (\mathrm{id} + \eta\boldsymbol{\nabla}G(z^*))[\mathrm{Log}_{z^*}(z_k)] + o(\|\mathrm{Log}_{z^*}(z_k)\|_{z^*})$, and $V(z_{k+1}) - V(z_k) = \langle \mathrm{Log}_{z^*}(z_k), \big((\mathrm{id} + \eta\boldsymbol{\nabla}G(z^*))^{\dagger} \circ H \circ (\mathrm{id} + \eta\boldsymbol{\nabla}G(z^*)) - H\big)[\mathrm{Log}_{z^*}(z_k)]\rangle_{z^*} + o(\|\mathrm{Log}_{z^*}(z_k)\|_{z^*}^2)$.*

Hence, to ensure a decrease in the Lyapunov function near the fixed point $z^*$, we require $(\mathrm{id} + \eta\boldsymbol{\nabla}G(z^*))^{\dagger} \circ H \circ (\mathrm{id} + \eta\boldsymbol{\nabla}G(z^*)) - H$ to be negative definite for some positive definite $H$. Here, id represents the identity operator. Following similar arguments in Proposition 6, this is equivalent to requiring the matrix representation of $\mathrm{id} + \eta\boldsymbol{\nabla}G(z^*)$ in an orthonormal basis to have a spectral radius less than one.

**Proposition 8.** *For a discrete dynamical system $z_{k+1} = \mathrm{Exp}_{z_k}(\eta\, G(z_k))$, a fixed point $z^*$ is called strictly stable if and only if the matrix representation of $\mathrm{id} + \eta\boldsymbol{\nabla}G(z^*)$ has all the eigenvalues with magnitude less than one for some $\eta > 0$.*

### 5.3 Stability equivalence and asymptotic convergence

The next proposition proves an equivalence between Propositions 6 and 8.

**Proposition 9.** *For discrete dynamics $z_{k+1} = \mathrm{Exp}_{z_k}(\eta\, G(z_k))$ on manifolds, the following are equivalent.*

*(1) $z^*$ is a strictly stable equilibrium point.*
*(2) All the eigenvalues of the matrix representation of $\mathrm{id} + \eta\boldsymbol{\nabla}G(z^*)$ have a magnitude lower than one for some $\eta > 0$.*
*(3) All the eigenvalues of the matrix representation of $\boldsymbol{\nabla}G(z^*)$ have real parts negative.*

Proposition 9 allows us to define the asymptotic convergence rate of dynamical systems on Riemannian manifolds.

**Definition 5** (Asymptotic convergence rate). Consider a dynamical system $z_{k+1} = \mathrm{Exp}_{z_k}(\eta\, G(z_k))$ on $\mathcal{M}$. Suppose $z^*$ is a strictly stable fixed point. We say $\rho := \max_i |1 + \eta\lambda_i|$ is the asymptotic convergence rate of the system for some $\eta > 0$ such that $|1 + \eta\lambda_i| < 1$ for all $i$, where $\lambda_i$ are the eigenvalues of the matrix representation of $\boldsymbol{\nabla}G(z^*)$ in an orthonormal basis.

**Remark 2.** We remark that the notions of stability and asymptotic convergence are generalizations of the counterparts in Euclidean space via coordinate representation. The notion of asymptotic convergence on manifolds follows from the linearization in Proposition 7 where the dynamics evolve according to $\mathrm{Log}_{z^*}(z_{k+1}) = (\mathrm{id} + \eta\boldsymbol{\nabla}G(z^*))[\mathrm{Log}_{z^*}(z_k)]$ around a local neighborhood of $z^*$. Such a claim is formalized in the following result, where the asymptotic convergence rate is related to the local linear convergence rate.

**Proposition 10.** *Consider the discrete dynamical system $z_{k+1} = \mathrm{Exp}_{z_k}(\eta\, G(z_k))$. Let $\rho = 1 - \kappa < 1$ be the asymptotic convergence rate of the system. Then, there exists a local neighborhood $\mathcal{U} \subseteq \mathcal{M}$ around $z^*$ such that for any $z_0 \in \mathcal{U}$, it satisfies $\mathrm{dist}(z_k, z^*) \le C(1 - \kappa/2)^k \mathrm{dist}(z_0, z^*)$ for some constant $C > 0$.*

In the next section, we use the above results to analyze the asymptotic convergence behavior of timescale-separated Riemannian gradient descent ascent (TSRGDA) method.

### 5.4 Asymptotic analysis of TSRGDA

In the Euclidean setup, Jin et al. (2020) analyze the limit points of (timescale-separated) gradient descent ascent (TSGDA) and establish a relationship between stable limit points of TSGDA with local minimax point and Nash saddle points. We attempt to generalize this analysis to the Riemannian gradient descent ascent (RGDA) method on manifolds (Huang & Gao, 2023). In Figure 1, we show a schematic view of an iterative algorithm on manifolds.

We consider timescale-separated RGDA (TSRGDA) with different stepsizes for descent and ascent directions, i.e.,

$$\text{TSRGDA:} \quad z_{k+1} = \mathrm{Exp}_{z_k}\big(\eta\, G_\tau(z_k)\big) = \mathrm{Exp}_{z_k}\begin{pmatrix} -\eta\tau\, \mathrm{grad}_x f(x_k, y_k) \\ \eta\, \mathrm{grad}_y f(x_k, y_k) \end{pmatrix}, \tag{1}$$

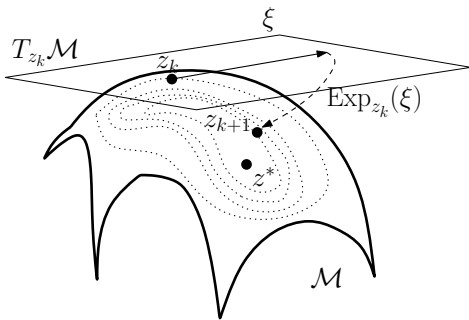

Figure 1: Schematic view of Riemannian optimization. The algorithm iteratively updates $z$ following a direction $\xi \in T_{z_k}\mathcal{M}$ with the goal to reach $z^*$. This scheme encompasses all the proposed methods considered in this paper.

where $z_k = (x_k, y_k)$ and $\tau > 0$ is the separation parameter. From Proposition 9, it is equivalent to studying the continuous version of such dynamics ($\eta \to 0$) as $\dot{z}(t) = G_\tau(z(t)), G_\tau(z(t)) = \begin{pmatrix} -\tau \operatorname{grad}_x f(x(t), y(t)) \\ \operatorname{grad}_y f(x(t), y(t)) \end{pmatrix}$.

Let TSRGDA denote the set of strictly stable equilibrium points of dynamics (1), with parameter $\tau$, and LNS $\coloneqq \{z = (x, y) \mid \operatorname{grad}_z f(z) = 0, \operatorname{Hess}_x f(z) \succ 0, \operatorname{Hess}_y f(z) \prec 0\}$, i.e., the set of strictly local Nash saddle points and LMiniMax $\coloneqq \{z = (x, y) \mid \operatorname{grad}_z f(z) = 0, (\operatorname{Hess}_x f - \operatorname{grad}^2_{xy} f \circ \operatorname{Hess}^{-1}_y f \circ \operatorname{grad}^2_{yx} f)(z) \succ 0, \operatorname{Hess}_y f(z) \prec 0\}$ be the set of strictly local minimax points. The following result claims for any fixed timescale separation parameter $\tau$, there exist certain functions, where TSRGDA may converge to unmeaning points, which are neither local Nash saddle nor local minimax points.

**Proposition 11.** *For any fixed $\tau$, we have* LNS $\subset$ TSRGDA, *while there exists some functions such that* TSRGDA $\not\subset$ LNS. *Furthermore, for any fixed $\tau$, there exists some functions such that* LMiniMax $\not\subset$ TSRGDA *and* TSRGDA $\not\subset$ LMiniMax.

The proof of Proposition 11 relies on Proposition 9 where we work with a matrix representation of $\boldsymbol{\nabla}G_\tau(z)$. It has been further shown by Jin et al. (2020) that provided $\tau$ sufficiently small, the stable equilibrium points of TSRGDA are strictly local minimax points (up to some degenerate points with respect to $\operatorname{Hess}_y f(z)$). A follow-up work (Fiez & Ratliff, 2021) further derives a non-asymptotic construction of an upper bound to $\tau$, which we extend to the Riemannian manifold settings as follows.

**Proposition 12.** *Suppose $z$ is a non-degenerate critical point of $f$, i.e., $\operatorname{grad}_z f(z) = 0$, $\operatorname{Hess}_y f(z)$ is non-degenerate. Then there exists $\tau_0 > 0$ such that for all $\tau \in (0, \tau_0)$, $z \in$ TSRGDA if and only if $z \in$ LMiniMax.*

In addition, the asymptotic convergence rate is derived for TSRGDA, motivated by Zhang et al. (2020).

**Theorem 1** (Asymptotic convergence of TSRGDA). *Suppose $z^* \in$ LMiniMax and for some $\tau$ sufficiently small, $z^* \in$ TSRGDA. Then the asymptotic convergence rate of TSRGDA to $z^*$ is given by $\max_i |1 + \eta\nu_i|$, where $\{\nu_i\}_{i=1}^{d_x+d_y}$ are the eigenvalues of $\boldsymbol{\nabla}G_\tau(z^*)$ in an orthonormal basis, and $\eta < \min_i\left(-2\mathcal{R}(\nu_i)/|\nu_i|^2\right)$, with $\mathcal{R}(\cdot)$ denoting the real part.*

**Remark 3.** From (Jin et al., 2020, Lemma 40), we see when $\tau \to 0$, we have the eigenvalues of matrix representation of $\boldsymbol{\nabla}G_\tau(z^*)$ asymptotically splits and converges to $\{-\tau\lambda_i\}_{i=1}^{d_x}$ and $\{-\mu_j\}_{j=1}^{d_y}$, where $\lambda_1 \geq \ldots \geq \lambda_{d_x} > 0$, $\mu_1 \geq \ldots \geq \mu_{d_y} > 0$ are eigenvalues of $(\operatorname{Hess}_x f - \operatorname{grad}^2_{xy} f \circ \operatorname{Hess}^{-1}_y f \circ \operatorname{grad}^2_{yx} f)(z^*)$ and $-\operatorname{Hess}_y f(z^*)$, respectively. In this limiting case, we see the convergence rate reduces to $\max_i |1 - \eta\tau\lambda_i| \vee \max_j |1 - \eta\mu_j|$ with $\eta < \frac{2}{\tau\lambda_1 \vee \mu_1}$.

# 6 Second-order methods for nonconvex-nonconcave problems on Riemannian manifolds

This section proposes various second-order methods that are guaranteed to converge to the local minimax point, without requiring a sufficiently small stepsize (i.e., $\tau$ sufficiently small) for the minimization variable as in TSRGDA. We also analyze a few recent Riemannian min-max algorithms proposed in the literature.

## 6.1 Riemannian follow-the-ridge and total gradient descent ascent methods

**Riemannian follow-the-ridge.** The Riemannian follow-the-ridge (RFR) generalizes the idea proposed by Wang et al. (2019). Based on the definition of a strictly local minimax point $(x^*, y^*)$, $x^*$ is a local minimum of $f(x, \psi(x))$ where $\psi : \mathcal{X} \to \mathcal{Y}$ is an implicit function such that $\mathrm{grad}_y f(x, \psi(x)) = 0$ for any $x$ in a neighborhood of $x^*$ (due to implicit function theorem on manifolds, i.e., Theorem 5 in Appendix). In order to ensure the condition $\mathrm{grad}_y f(x_k, \psi(x_k)) = 0$ is satisfied (approximately) at every iteration, a correction term for the $y$ player is added as $\mathrm{D}_x \psi(x_k)[\mathrm{Log}_{x_k}(x_{k+1})] \approx \mathrm{Log}_{y_k}(y_{k+1})$. Specifically, each iteration starts by updating $x_k$ as $x_{k+1} = \mathrm{Exp}_{x_k}(-\eta\tau\mathrm{grad}_x f(x_k, y_k))$ and then updates $y_k$ as $y_{k+1} = \mathrm{Exp}_{y_k}(\eta\,\mathrm{grad}_y f(x_k, y_k) + \mathrm{D}_x \psi(x_k)[\mathrm{Log}_{x_k}(x_{k+1})])$ where the latter term is computed following again the implicit function theorem on manifolds, as $\mathrm{D}_x \psi(x_k)[\mathrm{Log}_{x_k}(x_{k+1})] = \eta\tau([\mathrm{Hess}_y f]^{-1} \circ \mathrm{grad}^2_{yx} f)(x_k, y_k)[\mathrm{grad}_x f(x_k, y_k)]$. Suppose at iteration $k$, we have $\mathrm{grad}_y f(x_k, y_k) = 0$, then we can show $y_{k+1} \approx \psi(x_{k+1})$ with $\mathrm{grad}_y f(x_{k+1}, \psi(x_{k+1})) = 0$.

Using similar notations as before, we can write the dynamics compactly as

$$\text{RFR:} \quad z_{k+1} = \mathrm{Exp}_{z_k}\big(\eta\, G_{\mathrm{RFR}}(z_k)[\mathrm{grad}f(z_k)]\big), \qquad G_{\mathrm{RFR}}(z_k) = \begin{pmatrix} -\tau\mathrm{id} & 0 \\ \tau\big(\mathrm{Hess}_y^{-1} f \circ \mathrm{grad}^2_{yx} f\big)(z_k) & \mathrm{id} \end{pmatrix}.$$

**Riemannian total gradient descent ascent.** Total gradient descent ascent (Fiez et al., 2020) is closely related to the follow-the-ridge update, where both methods have been shown to precondition the GDA update with preconditioner being the transpose of each other (Zhang et al., 2020). The update of the Riemannian version is given by

$$\text{RTGDA:} \quad z_{k+1} = \mathrm{Exp}_{z_k}\big(\eta\, G_{\mathrm{RTGDA}}(z_k)[\mathrm{grad}f(z_k)]\big), \qquad G_{\mathrm{RTGDA}}(z_k) = \begin{pmatrix} -\tau\mathrm{id} & \tau\big(\mathrm{grad}^2_{xy} f \circ \mathrm{Hess}_y^{-1} f\big)(z_k) \\ 0 & \mathrm{id} \end{pmatrix}.$$

It can be seen that $G_{\mathrm{RTGDA}}(z_k) = (G_{\mathrm{RFR}}(z_k))^\dagger$ due to that $\mathrm{grad}^2_{xy} f = (\mathrm{grad}^2_{yx} f)^\dagger$. Hence, RFR and RTGDA share similar convergence properties as stated below.

**Theorem 2** (Asymptotic convergence of RFR and RTGDA). *For any $\tau > 0$, the following three statements are equivalent: (1) $z^*$ is a strictly local minimax point; (2) $z^*$ is a strictly stable fixed point of RFR; and (3) $z^*$ is a strictly stable fixed point of RTGDA. Moreover, the asymptotic convergence rate of RFR and RTGDA to a strictly local minimax point is $\rho = \max_i |1 - \eta\tau\lambda_i| \vee \max_j |1 - \eta\mu_j|$ for $\eta < \frac{2}{\tau\lambda_1 \vee \mu_1}$, where $\lambda_i, \mu_j$ are the same eigenvalues as in Theorem 1.*

**Remark 4.** Compared to TSRGDA in Theorem 1 and Remark 3, we see RTGDA and RFR achieve the same (limiting) asymptotic convergence rate with an arbitrary choice of $\tau$, which is in general numerically favorable.

## 6.2 Riemannian Newton gradient descent and Newton follow-the-ridge.

The Euclidean Newton gradient descent method (Zhang et al., 2020) updates the max player with a Newton step, which allows the maximization step to be solved accurately and is more favorable when the maximization problem is ill-conditioned, i.e., $\mathrm{Hess}_y f(z^*)$ has a large eigenvalue gap. The Riemannian Newton gradient descent (RNGD) is given by

$$\text{RNGD:} \quad x_{k+1} = \mathrm{Exp}_{x_k}\big(-\eta\tau\mathrm{grad}_x f(z_k)\big), \quad y_{k+1} = \mathrm{Exp}_{y_k}\big(-\eta\mathrm{Hess}_y^{-1} f(z_k)[\mathrm{grad}_y f(x_{k+1}, y_k)]\big),$$

where the negative sign in the max variable is because the Newton step is the same for both minimization and maximization problems.

Another related algorithm, known as the Newton follow-the-ridge (Gao et al., 2022) method, presents a first-order approximation to the maximization step of the Newton gradient descent. On general Riemannian manifolds, by Taylor approximation of $\mathrm{grad}_y f(x_{k+1}, y_k)$, we have $\mathrm{grad}_y f(x_{k+1}, y_k) \approx \mathrm{grad}_y f(z_k) - \eta\tau\mathrm{grad}_{yx}^2 f(z_k)[\mathrm{grad}_x f(z_k)]$ where substituting this approximation to RNGD yields the Riemannian Newton follow-the-ridge (RNFR) as

$$\text{RNFR:} \quad z_{k+1} = \mathrm{Exp}_{z_k}\big(\eta\, G_{\mathrm{RNFR}}(z_k)[\mathrm{grad}f(z_k)]\big), \quad G_{\mathrm{RNFR}}(z_k) = \begin{pmatrix} -\tau\mathrm{id} & 0 \\ \tau(\mathrm{Hess}_y^{-1}f \circ \mathrm{grad}_{yx}^2 f)(z_k) & -\zeta\mathrm{Hess}_y^{-1}f(z_k) \end{pmatrix}.$$

We remark that Gao et al. (2022) use $\zeta'\mathrm{id} - \zeta\mathrm{Hess}_y^{-1}f(z_k)$ in place of $-\zeta\mathrm{Hess}_y^{-1}f(z_k)$ for the max variable. Since we motivate our RNFR from Taylor approximation, we decide to exclude $\zeta'\mathrm{id}$, where the subsequent analysis does not differ significantly. In addition, based on the discussion in the previous section, we propose a Riemannian Newton total gradient descent ascent (RNTGDA) by simply taking the adjoint of $G_{\mathrm{RNFR}}$. Specifically, let $G_{\mathrm{RNTGDA}}(z_k) = (G_{\mathrm{RNFR}}(z_k))^\dagger$ we directly consider the update on general Riemannian manifolds as

$$\text{RNTGDA:} \quad z_{k+1} = \mathrm{Exp}_{z_k}(\eta\, G_{\mathrm{RNTGDA}}(z_k)[\mathrm{grad}f(z_k)]) = \mathrm{Exp}_{z_k}(\eta\,(G_{\mathrm{RNFR}}(z_k))^\dagger[\mathrm{grad}f(z_k)]).$$

Below, we present the convergence guarantees for RNFR and RNTGDA.

**Theorem 3** (Asymptotic convergence of RNFR and RNTGDA)**.** *For any $\tau > 0$, the following three statements are equivalent: (1) $z^*$ is a strictly local minimax point; (2) $z^*$ is a strictly stable fixed point of RNFR; and (3) $z^*$ is a strictly stable fixed point of RNTGDA. When choosing $\zeta = 1/\eta$, the asymptotic convergence rate of RNFR and RNTGDA to a strictly local minimax point is $\rho = \max_i |1 - \eta\tau\lambda_i|$ where $\eta < \frac{2}{\tau\lambda_1}$ and $\lambda_i s$ are the eigenvalues of $\left(\mathrm{Hess}_x f - \mathrm{grad}_{xy}^2 \circ \mathrm{Hess}_y^{-1}f \circ \mathrm{grad}_{yx}^2\right)(z^*)$.*

**Remark 5.** Compared to the convergence of RFR and RTGDA, the asymptotic convergence rate of RNFR and RNTGDA does not depend on the eigenvalues of $\mathrm{Hess}_y f(z^*)$. Thus, when $\mathrm{Hess}_y f(z^*)$ is ill-conditioned, we obtain an improved convergence rate with the Newton step for the max player.

### 6.3 Riemannian Hamiltonian and consensus methods (RHM/RCON)

Riemannian Hamiltonian and consensus methods (Han et al., 2023b) are motivated by the minimization of gradient norm as a proxy problem to the original min-max problem. The updates of the methods are

$$\text{RHM/RCON:} \quad z_{k+1} = \mathrm{Exp}_{z_k}\big(\eta H(z_k)\big) = \mathrm{Exp}_{z_k}\big(\eta(\beta G(z_k) - (\boldsymbol{\nabla}G(z_k))^\dagger[G(z_k)])\big), \tag{2}$$

where $G(z_k) = (-\mathrm{grad}_x f(z_k), \mathrm{grad}_y f(z_k))$ and $\eta > 0, \beta \geq 0$. We also let $H(z_k) = \beta G(z_k) - (\boldsymbol{\nabla}G(z_k))^\dagger[G(z_k)]$ Particularly the update (2) is called the Riemannian Hamiltonian method (RHM) when $\beta = 0$ and called the Riemannian consensus method (RCON) when $\beta > 0$. The methods are shown to converge linearly to stationary points of the min-max problem, provided the Riemannian Hamiltonian function, i.e., $\|G(z_k)\|_{z_k}^2$, satisfies the general Riemannian PL condition.

The below result shows that for general nonconvex-nonconcave functions, RHM/RCON may converge to undesirable stable fixed points.

**Proposition 13.** *The strictly stable fixed points of RHM and RCON may contain local minima, maxima, minimax, or maximin points.*

## 7 Experiments

In this section, we evaluate the proposed second-order algorithms – RFR, RTGDA, RNGD, RNFR, and RNTGDA (proposed and analyzed in Section 6) – on a variety of problems and compare them against the below baselines. The code is available at `https://github.com/andyjm3/nonconvex-nonconcave-mfd`.

- **RGDA** (Jordan et al., 2022): It is a first-order method with same stepsize for the min and max variables.

- **TSRGDA** (studied in Section 5.4): variant of the RGDA method but with *different* stepsizes for min and max variables, i.e., timescale separated.
- **RCEG** (Zhang et al., 2022c; Jordan et al., 2022): The Riemannian corrected extra-gradient method is a first-order method where the search direction is modified by taking gradient from the past iterate.
- **RHM** (Han et al., 2023b): the Riemannian Hamiltonian method with fixed stepsize.
- **RCON** (Han et al., 2023b): the Riemannian consensus method with fixed stepsize. It is based on the RHM search direction but now combined with the gradient descent accent direction.

All our experiments are done in Matlab with the Manopt package (Boumal et al., 2014) except for the GAN experiments (Section 7.4), where we use the Geoopt package with Pytorch (Kochurov et al., 2020). For the proposed second-order methods, the Hessian inverse is computed via conjugate gradient on tangent space (Boumal, 2020, Algorithm 6.2), with practical implementation details given in Appendix C. For all the experiments, we stop the algorithms either when the optimality gap (or gradient norm wherever applicable) falls below a threshold or the maximum iteration count is reached. The experiments in Section 7.1 show that RTGDA and RNTGDA empirically perform similarly to RFR and RNFR, respectively (this also follows from the theoretical analysis in Sections 6.1 and 6.2). Therefore, for all other subsequent experiments, we remove RTGDA and RNTGDA from our comparisons.

## 7.1 Quadratic optimization

We evaluate the algorithms on the functions discussed in Proposition 4.

**Logdet quadratic problem.** The first example is the following quadratic optimization problem with the log-determinant (logdet) function, i.e.,

$$\min_{\mathbf{X} \in \mathbb{S}_{++}^d} \max_{\mathbf{Y} \in \mathbb{S}_{++}^d} c_1 (\log \det(\mathbf{X}))^2 + c_2 \log \det(\mathbf{X}) \log \det(\mathbf{Y}) + c_3 (\log \det(\mathbf{Y}))^2 \tag{3}$$

for $c_1, c_2, c_3 \in \mathbb{R}$. We consider both nonconvex-geodesic-concave as well as geodesic-convex-concave settings, i.e., $c_1 = -1, c_2 = 5, c_3 = -1$ and $c_1 = 1, c_2 = 10, c_3 = -1$. We respectively label the two problems as `LD:NCGC` and `LD:GCC`, representing nonconvex-geodesic-concave and geodesic convex-concave logdet quadratic problems. The Riemannian gradient and second-order derivatives for the problem are derived in Lemma 4.

From Proposition 4, we see that the local minimax points $(\mathbf{X}^*, \mathbf{Y}^*)$ of problem `LD:NCGC` satisfy $\det(\mathbf{X}^*) = \det(\mathbf{Y}^*) = 1$. For the problem `LD:GCC`, the global saddle points are shown to also satisfy the same conditions as in (Han et al., 2023b, Proposition 8). Hence for both problems, we use the same criterion $|\det(\mathbf{X}) - 1| + |\det(\mathbf{Y}) - 1|$ for measuring the optimality gap as in Han et al. (2023b).

**Riemannian distance quadratic problem.** We now consider a modification to (3), where $(\log \det(\mathbf{Y}))^2$ is replaced with $\text{dist}^2(\mathbf{I}, \mathbf{Y})$, i.e.,

$$\min_{\mathbf{X} \in \mathbb{S}_{++}^d} \max_{\mathbf{Y} \in \mathbb{S}_{++}^d} c_1 (\log \det(\mathbf{X}))^2 + c_2 \log \det(\mathbf{X}) \log \det(\mathbf{Y}) + c_3 \text{dist}^2(\mathbf{I}, \mathbf{Y}), \tag{4}$$

where we recall the Riemannian distance on $\mathbb{S}_{++}^d$ with the affine-invariant metric is $\text{dist}^2(\mathbf{A}, \mathbf{B}) = \|\text{logm}(\mathbf{A}^{-1/2}\mathbf{B}\mathbf{A}^{-1/2})\|_F^2$. For this problem instance, we consider the setting $c_1 = -1, c_2 = 5, c_3 = -1$, which we label as `RD:NCGC`, which is a nonconvex-geodesic-(strongly)-concave problem. The Riemannian gradient and second-order derivatives are given in Lemma 5. From Proposition 4, we see the local minimax points $(\mathbf{X}^*, \mathbf{Y}^*)$ satisfy $\det(\mathbf{X}^*) = 1, \mathbf{Y}^* = \mathbf{I}$. We use $|\det(\mathbf{X}) - 1| + \text{dist}^2(\mathbf{Y}^*, \mathbf{I})$ to measure the optimality gap.

**Experiment setup and results.** For experiments, we consider $d = 30$ and tune stepsizes for all the methods. For RCON, we also tune $\gamma$ after fixing the same stepsize as RHM. We fix $\zeta = 1$ for RNFR and RNTGDA. The results are plotted in Figure 2, where we show the optimality gap as a function of both iteration counts and time. For the `RD:NCGC` problem, we also plot the gradient norm against time. Examining the plots, we make the following observations.

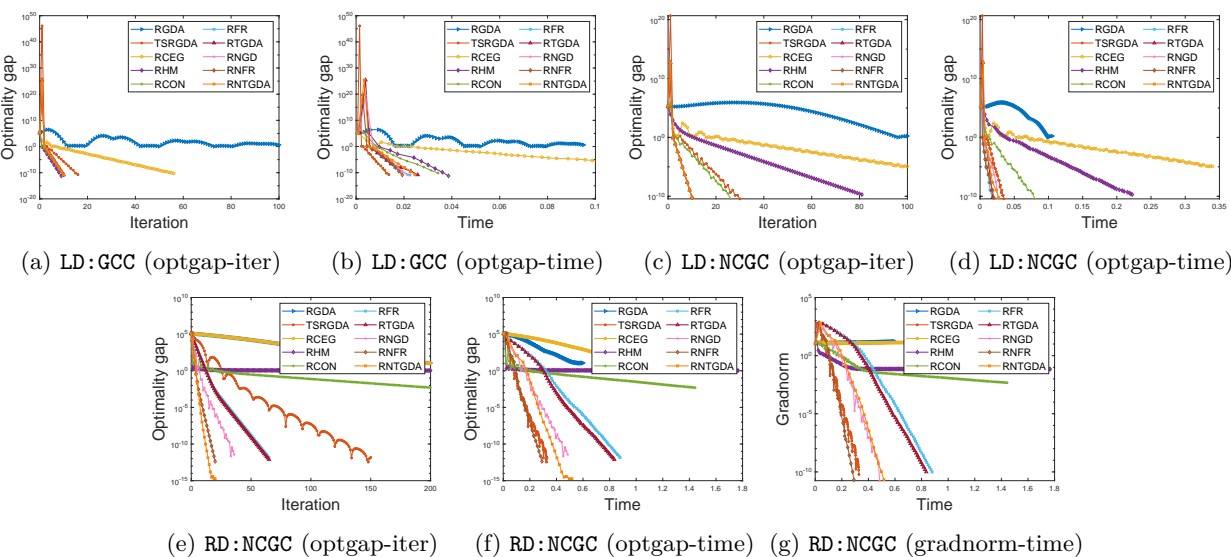

(a) `LD:GCC` (optgap-iter)  (b) `LD:GCC` (optgap-time)  (c) `LD:NCGC` (optgap-iter)  (d) `LD:NCGC` (optgap-time)

(e) `RD:NCGC` (optgap-iter)  (f) `RD:NCGC` (optgap-time)  (g) `RD:NCGC` (gradnorm-time)

Figure 2: Experiments on the quadratic optimization problems (3) and (4) on the SPD manifold. We observe faster convergence of proposed second-order methods against the benchmarks particularly for nonconvex-nonconcave settings (`LD:NCGC`, `RD:NCGC`).

- For all the problem instances, our proposed second-order methods perform competitively among the baselines. They perform the best in terms of both iteration and runtime for the two nonconvex-geodesic-concave problems (`LD:NCGC`, `RD:NCGC`).
- We observe very similar convergence behavior of RFR and RTGDA as well as RNFR and RNTGDA. This empirical observation matches the theoretical analysis as they share the same preconditioner (up to a transpose) and have the same asymptotic convergence rate.
- For the geodesic convex-concave problem (`LD:GCC`), Figure 2(a), we see RHM/RCON outperforms other methods in terms of the number of iterations. As the proxy problem for RHM/RCON satisfies the Riemannian PL condition as demonstrated in (Han et al., 2023b, Proposition 8), RHM and RCON have better convergence rates.
- Among the methods that use first-order information, we observe that TSRGDA outperforms RGDA and RCEG. This suggests using different stepsizes for minimization and maximization updates is essential to improve the convergence.

### 7.2 Robust Fréchet mean

The robust Fréchet mean (`RFM`) computation problem on the SPD manifold (Zhang et al., 2022c; Han et al., 2023b) can be formulated as the task of finding the Fréchet mean (of a set of SPD matrices) with eigenvalues bounded away from zero, i.e.,

$$\min_{\mathbf{M} \in \mathbb{S}_{++}^d} \max_{\mathbf{x} \in \mathcal{S}^d} \mathbf{x}^\top \mathbf{M} \mathbf{x} + \frac{\alpha}{n} \sum_{i=1}^n \mathrm{dist}^2(\mathbf{M}, \mathbf{M}_i), \tag{5}$$

where $\alpha > 0$ is the regularization parameter and $\mathcal{S}^d := \{\mathbf{x} \in \mathbb{R}^d : \mathbf{x}^\top \mathbf{x} = 1\}$ is the sphere manifold. This problem is geodesic strongly convex in $\mathbf{M}$ and in general nonconcave for $\mathbf{x}$. We derive the Riemannian gradient and second-order derivatives in Lemma 6. We use the gradient norm as a measure of distance to optimality.

**Experiment setup and results.** We follow the experimental setting in Zhang et al. (2022c); Han et al. (2023b) for generating the sample SPD matrices with bounded eigenvalues in $[\lambda_0, \lambda_1]$. We choose $d = 50$, $n = 40$, $\lambda_0 = 0.2, \lambda_1 = 4.5, \alpha = 0.1$ as in Han et al. (2023b).

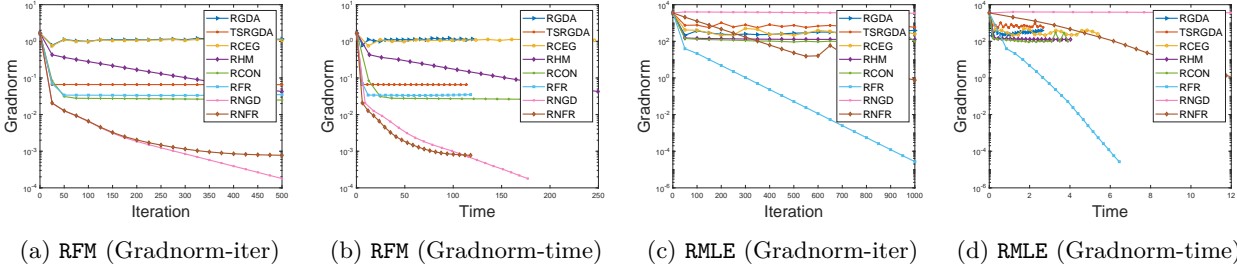

| (a) `RFM` (Gradnorm-iter) | (b) `RFM` (Gradnorm-time) | (c) `RMLE` (Gradnorm-iter) | (d) `RMLE` (Gradnorm-time) |

Figure 3: Experiments on the robust Fréchet mean (`RFM`) and robust maximum likelihood estimation (`RMLE`) problems. For `RFM`, we observe that RNGD and RNFR achieve the fastest convergence. The other converging method is RHM, although its convergence is slow. All other baselines converge to undesired points that are not stationary. For `RMLE`, the only two converging algorithms are RFR and RNFR, while first-order methods (RGDA, TSRGDA, and RCEG), RNGD, and RHM/RCON either do not converge or converge slowly.

From Figures 3a and 3b, we see that RNGD and RNFR outperform others in both in number of iterations required and runtime. RHM converges as well, but slowly. RFR and all other first-order methods fail to converge suggesting that the Newton step provides a better-conditioned maximization update.

### 7.3 Robust maximum likelihood estimation

The robust maximum likelihood estimation problem (Bertsimas & Nohadani, 2019; Zhang et al., 2022c) is formulated as follows. Given an observed data matrix $\mathbf{X} \in \mathbb{R}^{n \times d}$ where $n, d$ are respectively the number of samples and feature dimension, the objective is

$$\min_{\boldsymbol{\delta} \in \mathcal{S}^d} \max_{\boldsymbol{\mu} \in \mathbb{R}^d, \boldsymbol{\Sigma} \in \mathbb{S}^d_{++}} -\frac{nd}{2} \log(2\pi) - \frac{n}{2} \log \det(\boldsymbol{\Sigma}) - \frac{1}{2} \sum_{i=1}^n (\mathbf{x}_i - \boldsymbol{\delta} - \boldsymbol{\mu})^\top \boldsymbol{\Sigma}^{-1} (\mathbf{x}_i - \boldsymbol{\delta} - \boldsymbol{\mu}), \tag{6}$$

where we choose the constraint set for the perturbation as a sphere, i.e., the perturbed samples are unit-distance away from the original ones. We adopt the reformulation in (Hosseini & Sra, 2020, Theorem 1) to transform problem (6) to be geodesic concave in the maximization variable. If $\mathbf{y}_i^\top = [\mathbf{x}_i^\top \ 1] \in \mathbb{R}^{d+1}$ for all $i$ be the augmented data, then an equivalent problem to (6) is

$$\min_{\boldsymbol{\delta} \in \mathcal{S}^{d+1}} \max_{\mathbf{S} \in \mathbb{S}^{d+1}_{++}} -\frac{n}{2} \log \det(\mathbf{S}) - \frac{1}{2} \sum_{i=1}^n (\mathbf{y}_i - \boldsymbol{\delta})^\top \mathbf{S}^{-1} (\mathbf{y}_i - \boldsymbol{\delta}), \tag{7}$$

where we ignore the constants. Although the problem is geodesic concave in $\mathbf{S}$, it is in general nonconvex in $\boldsymbol{\delta}$. The Riemannian gradient and second-order derivatives are derived in Lemma 7.

**Experiment setup and results.** We use the `fisheriris` dataset from Matlab, which consists of $n = 150$ samples in $d = 4$ dimension. We augment the given data matrix with a column of ones and solve the reformulated problem (7). We observe in Figures 3c, 3d that RFR and RNFR converge faster, whereas others either do not converge or converge slowly.

### 7.4 Orthonormal generative adversarial networks

Lastly, we evaluate the proposed second-order methods on the application of generative adversarial network (GAN), which optimizes a game between a generator and a discriminator network. In particular, we consider the orthonormal GAN (Brock et al., 2019; Müller et al., 2019), where the weights of the discriminator are constrained to be orthonormal. Such a constraint has shown to improve training stability and generalization capability (Müller et al., 2019). To this end, we formulate the objective of the orthonormal GAN (`OGAN`) problem as

$$\min_{\{\mathbf{W}_\ell^G\}} \max_{\{\mathbf{W}_\ell^D\}: \mathbf{W}_\ell^D \in \text{St}(d_\ell, d_{\ell+1})} \frac{1}{n} \sum_{i=1}^n \Big( \log\big(\sigma(D(\mathbf{x}_i))\big) + \log\big(1 - \sigma(D(G(\mathbf{z}_i)))\big) \Big),$$

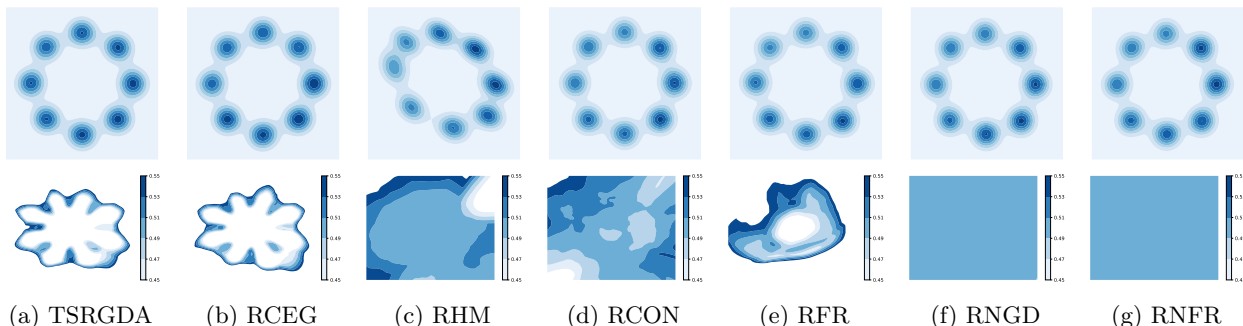

(a) TSRGDA      (b) RCEG      (c) RHM      (d) RCON      (e) RFR      (f) RNGD      (g) RNFR

Figure 4: `OGAN` on a mixture of Gaussians. The **top** row plots samples generated from the generator and the **bottom** row plots prediction from the discriminator on the entire 2d domain where the pixel intensity represents the predicted probability. Although all the algorithms achieve good sample quality for the generator, only the discriminator optimized by RNGD and RNFR is fooled completely by the generator, suggesting that the Newton step is crucial for stabilizing training.

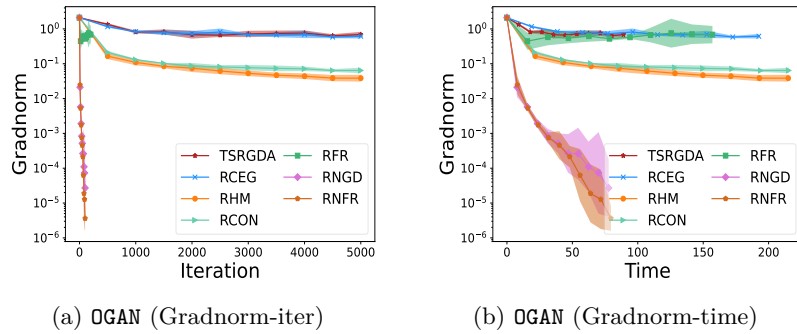

(a) `OGAN` (Gradnorm-iter)         (b) `OGAN` (Gradnorm-time)

Figure 5: Experiments on orthonormal generative adversarial network (`OGAN`) on a mixture of Gaussians where we see RNGD and RNFR converge within a few iterations, while the others fail to converge.

where we denote $\sigma(\cdot)$ as the sigmoid function, $\mathbf{x}_i$ as the input samples, $\mathbf{z}_i$ as the input noise, usually following a Gaussian distribution, We also use $D$ and $G$ to represent the discriminator and generator networks, respectively. The weights of $l$-th layer of generator and $\ell$-the layer of discriminator are denoted as $\mathbf{W}_l^G, \mathbf{W}_\ell^D$, and $\mathrm{St}(d, r)$ (with $r \leq d$) represents the set of column orthonormal matrices (i.e., the Stiefel manifold).

**Experiment setup and results.** We consider the mixture of 8 Gaussians example as in Zhang et al. (2020) and use the same network architecture where both the generator and discriminator have 3 hidden layers, each with 256 units and ReLU activation. The dimension of the prior $\mathbf{z}_i$ is 100 and the batch size is set as 512. Following Han et al. (2023b), we parameterize the orthonormal weights only for the second last layer of the discriminator. To examine the local convergence, we use the pre-trained network parameters in Zhang et al. (2020) (after projecting to the manifold) as initialization. For RFR, RNGD, and RNFR, we use 20 iterations for the conjugate gradient method (that is used to compute the Hessian inverse). We fix the stepsize for the generator to be 0.001 for all methods and tune the stepsize of the discriminator. We set the maximum number of iterations to 5000 for RGDA, RCEG, RHM, and RCON, 200 for RFR, and 100 for RNGD and RNFR.

Figures 5a and 5b show the gradient norm in terms of both iteration and runtime, respectively, averaged over five random runs. We observe that RNGD and RNFR converge within a few iterations while other methods do not. The significant convergence speed justifies the additional per-iteration cost in computing the Hessian inverse. We further plot the generated samples and discriminator prediction in Figure 4. We notice that despite generating high-quality samples by all methods, only RNGD and RNFR trick the discriminator to predict uniformly almost everywhere.

# 8 Concluding remarks

This work studies nonconvex-nonconcave min-max Riemannian optimization and introduces the notion of minimax points for defining local optimality on manifolds. It generalizes the analysis from the Euclidean space. We also quantify the stability and asymptotic convergence for dynamics on manifolds that allow studying the limiting behaviors of various existing algorithms under the nonconvex-nonconcave regimes. As an additional novel contribution, we propose a class of second-order methods with provable asymptotic convergence to local minimax points. The experiments show the merits of the proposed methods. In particular, for many nonconvex-nonconcave problems – robust Fréchet mean, robust maximum likelihood estimation, and orthonormal generative adversarial networks – the only methods converging to the local minimax are the proposed second-order methods.

To the best of our knowledge, this is the first work that systematically studies nonconvex-nonconcave min-max optimization problems on manifolds. However, there remain many open challenges and questions. One question is whether non-asymptotic convergence to local minimax points can be shown for the methods proposed on manifolds. Convergence under stochastic settings can be also an interesting direction. Furthermore, there exist various other notions of local optimality for min-max problems in Euclidean space. It would be interesting to study those on manifolds and verify the (non-)convergence of existing methods to such optimality. Similarly, developing acceleration schemes for solving min-max optimization on manifolds is a challenging but interesting research problem.

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

## A  Organization of Appendix

The appendix sections are organized as follows.

- **Appendix B** provides a preliminary introduction to Riemannian geometry and Riemannian optimization.

- **Appendix C** discusses the practical implementation of the proposed algorithms in the main text, including the evaluation of the Hessian inverse and ingredients for the Riemannian geometries considered in this work.

- **Appendix D** includes additional experimental results on the problem of computing projection robust Wasserstein distance.

- **Appendix E** derives some useful lemmas for the proofs of the main results.

- **Appendices F-J** present the proofs for the main results.

## B  Preliminaries on Riemannian geometry and Riemannian optimization

This section introduces the basics of Riemannian geometry and optimization with a focus on the essential concepts required for this work. We refer readers to Absil et al. (2009); Boumal (2020) for a detailed exposition of the topics.

**Riemannian geometry.** A Riemannian manifold $\mathcal{M}$ is a smooth manifold equipped with a smooth inner product structure, i.e., a Riemannian metric on each tangent space $T_z\mathcal{M}$, $\forall z \in \mathcal{M}$, which is denoted as $\langle u, v \rangle_z$ for any $u, v \in T_z\mathcal{M}$. Riemannian metric induces a norm $\|u\|_z := \sqrt{\langle u, u \rangle_z}$. We also use $T\mathcal{M}$ to denote the tangent bundle, which is the disjoint union of all tangent spaces. A *geodesic* $\gamma : [0, 1] \to \mathcal{M}$ generalizes the straight line segment in Euclidean space, which is the locally shortest curve with zero acceleration. The *exponential map*, $\mathrm{Exp}_z : T_z\mathcal{M} \to \mathcal{M}$ and for any $u \in T_z\mathcal{M}$, $\mathrm{Exp}_z(u)$ is defined such that there exists a geodesic $\gamma$ with $\gamma(0) = z, \gamma'(0) = u$ and $\gamma(1) = \mathrm{Exp}_z(u)$. Riemannian distance $\mathrm{dist}(z_1, z_2)$ is defined as the length of the distance minimizing geodesic connecting $z_1, z_2 \in \mathcal{M}$. In a uniquely geodesic neighborhood $\mathcal{U}$, the exponential map has a smooth inverse, namely the logarithm map, i.e., $\mathrm{Log}_{z_1} : \mathcal{U} \to T_{z_1}\mathcal{M}$. In such a neighbourhood, we can express $\mathrm{dist}(z_1, z_2) = \|\mathrm{Log}_{z_1}(z_2)\|_{z_1} = \|\mathrm{Log}_{z_2}(z_1)\|_{z_2}$. *Parallel transport* $\Gamma_{z_1}^{z_2} : T_{z_1}\mathcal{M} \to T_{z_2}\mathcal{M}$ is a linear isometric map between tangent spaces such that the vector field along a curve connecting $z_1, z_2$ is parallel.

This paper also considers the product of Riemannian manifolds, i.e., $\mathcal{M} = \mathcal{M}_x \times \mathcal{M}_y$, which also admits a Riemannian manifold structure, with the Riemannian metric induced from $\mathcal{M}_x, \mathcal{M}_y$. That is, for any $z = (x, y) \in \mathcal{M}$, $(u, v), (u', v') \in T_z\mathcal{M}$, $\langle (u, v), (u', v') \rangle_z = \langle u, u' \rangle_x^{\mathcal{M}_x} + \langle v, v' \rangle_y^{\mathcal{M}_y}$ where we denote $\langle \cdot, \cdot \rangle^{\mathcal{M}_x}, \langle \cdot, \cdot \rangle^{\mathcal{M}_y}$ as Riemannian metrics on $\mathcal{M}_x, \mathcal{M}_y$, respectively. We will make use of a linear operator on the tangent space of Riemannian product manifolds, i.e., $H : T_z\mathcal{M} \to T_z\mathcal{M}$. We can write out its operation on $\xi = (u, v) \in T_x\mathcal{M}_x \times T_y\mathcal{M}_y$ as $H[\xi] := \begin{pmatrix} H_{xx} & H_{xy} \\ H_{yx} & H_{yy} \end{pmatrix} \begin{pmatrix} u \\ v \end{pmatrix} = \begin{pmatrix} H_{xx}[u] + H_{xy}[v] \\ H_{yx}[u] + H_{yy}[v] \end{pmatrix}$ where $H_{xx} : T_x\mathcal{M}_x \to T_x\mathcal{M}_x, H_{yy} : T_y\mathcal{M}_y \to T_y\mathcal{M}_y$, $H_{xy} : T_y\mathcal{M}_y \to T_x\mathcal{M}_x$, $H_{yx} : T_x\mathcal{M}_x \to T_y\mathcal{M}_y$ are linear operators. Here we slightly abuse the block-matrix notation for representing the decomposition. For a linear operator $G : T_x\mathcal{M}_x \to T_y\mathcal{M}_y$ between tangent spaces, we denote its adjoint operator (with respect to Riemannian metrics) as $G^\dagger : T_y\mathcal{M}_y \to T_x\mathcal{M}_x$ such that for any $u_x \in T_x\mathcal{M}_x, u_y \in T_y\mathcal{M}_y$, $\langle u_y, G[u_x] \rangle_y^{\mathcal{M}_y} = \langle u_x, G^\dagger[u_y] \rangle_x^{\mathcal{M}_x}$. We use $\boldsymbol{\nabla}$ to denote the *covariant derivative* induced from the Riemannian connection. On a product manifold, its Riemannian connection is derived from the product connection (Boumal, 2020, Exercise 5.4).

**Riemannian optimization.** For a real-valued function $f : \mathcal{M} \to \mathbb{R}$, its Riemannian gradient $\mathrm{grad} f(z)$ is the tangent vector that satisfies $\langle \mathrm{grad} f(z), \xi \rangle_z = \mathrm{D} f(z)[\xi]$ for any $\xi \in T_z\mathcal{M}$. The Riemannian Hessian $\mathrm{Hess} f(z) : T_z\mathcal{M} \to T_z\mathcal{M}$ is a self-adjoint, linear operator, defined as the covariant derivative of the Riemannian gradient, i.e., $\mathrm{Hess} f(z) = (\boldsymbol{\nabla} \mathrm{grad} f)(z)$. A geodesic convex set $\mathcal{U} \subseteq \mathcal{M}$ is a subset where every two points are joined with a geodesic that lies entirely in the set. A function $f : \mathcal{M} \to \mathbb{R}$ is called $\mu$-geodesic strongly

convex in $\mathcal{U}$ is for any $z_1, z_2 \in \mathcal{U}$, there exists a geodesic $\gamma$ connecting them such that $\frac{d^2 g(\gamma(t))}{dt^2} \geq \mu$ for all $t \in [0,1]$ for some $\mu \geq 0$. If $\mu = 0$, the function is called geodesic convex. A function is called geodesic linear if it is both geodesic convex and geodesic concave.

**Riemannian min-max optimization.** For a bifunction $f : \mathcal{M}_x \times \mathcal{M}_y \to \mathbb{R}$, we denote $\nabla_x f(x,y), \nabla_y f(x,y)$ as the Euclidean (partial) gradient and $\operatorname{grad}_x f(x,y), \operatorname{grad}_y f(x,y)$ as the Riemannian (partial) gradient with respect to $x, y$ respectively. A bifunction $f(x,y)$ is geodesic (strongly) convex (strongly) concave if it is geodesic (strongly) convex in $x$ and geodesic (strongly) concave in $y$. For the bifunction $f(x,y)$, its Riemannian Hessian is derived in (Han et al., 2023b, Proposition 9), given by $\operatorname{Hess}_{(x,y)} f(x,y)[\xi] = \begin{pmatrix} \operatorname{Hess}_x f(x,y) & \operatorname{grad}^2_{xy} f(x,y) \\ \operatorname{grad}^2_{yx} f(x,y) & \operatorname{Hess}_y f(x,y) \end{pmatrix} \begin{pmatrix} u \\ v \end{pmatrix}$, where $\operatorname{grad}^2_{xy} f(x,y) : T_y \mathcal{M}_y \to T_x \mathcal{M}_x$ is the Riemannian cross derivatives, computed as $\operatorname{grad}^2_{xy} f(x,y)[v] = \operatorname{D}_y \operatorname{grad}_x f(x,y)[v]$ and similarly for $\operatorname{grad}^2_{yx} f(x,y)$. It has been shown in (Han et al., 2023b, Proposition 10) that $\operatorname{grad}^2_{xy} f(x,y), \operatorname{grad}^2_{yx} f(x,y)$ are adjoint operators due to self-adjointness of Riemannian Hessian.

# C   Implementation and Riemannian geometries

## C.1   Practical considerations

The second-order methods introduced in this section all require evaluating the Hessian inverse. In order to ensure efficient computation in real applications, we adopt the following practical implementation strategies whenever necessary.

**Conjugate gradient and regularization.** In order to compute $\operatorname{Hess}_y^{-1} f(x,y)[v]$, we consider solving a linear system $(\operatorname{Hess}_y f)^2(x,y)[s] = \operatorname{Hess}_y f(x,y)[v]$ for $s \in T_y \mathcal{M}_y$. This can be solved via a conjugate gradient algorithm on tangent space (Boumal, 2020, Section 6.3). To ensure $(\operatorname{Hess}_y f)^2(x,y)$ is strictly positive definite, we add a regularization term to the linear system as $((\operatorname{Hess}_y f)^2 + \epsilon \operatorname{id})(x,y)[s] = \operatorname{Hess}_y f(x,y)[v]$ for some sufficiently small $\epsilon > 0$. Such a strategy has also been considered in the Euclidean space (Wang et al., 2019; Zhang et al., 2020; Gao et al., 2022). In all the experiments, we simply set $\epsilon = 0$.

**Use of general retraction in place of exponential map.** When the exponential map is inaccessible or computationally costly to evaluate, a general retraction $(\operatorname{Retr} : T\mathcal{M} \to \mathcal{M})$ can be used to approximate the exponential map, usually to the first order. With retraction, the resulting dynamics is $z_{k+1} = \operatorname{Retr}_{z_k}(\eta\, G(z_k))$, and can be similarly linearized using inverse retraction $(\operatorname{Retr}^{-1} : \mathcal{M} \to T\mathcal{M})$, and Proposition 7 holds, i.e., $\operatorname{Retr}^{-1}_{z^*}(z_{k+1}) = (\operatorname{id} + \eta \boldsymbol{\nabla} G(z^*))[\operatorname{Retr}^{-1}_{z^*}(z_k)] + o(\|\operatorname{Retr}^{-1}_{z^*}(z_k)\|_{z^*})$ (see Han et al. (2023a) for more details). Hence the same notions of stability and asymptotic convergence in Section 5.3 apply, and the convergence analysis for various methods also holds.

For the experiments, we use the exponential map whenever such an operation is well-defined, e.g., on SPD and sphere manifolds. For the Stiefel and doubly stochastic manifolds, we use the retraction operation.

## C.2   Riemannian geometries of manifolds of interest

**Symmetric positive definite (SPD) manifold.** The SPD manifold of size $d$ is denoted as $\mathbb{S}^d_{++} := \{\mathbf{X} \in \mathbb{R}^{d \times d} : \mathbf{X}^\top = \mathbf{X}, \mathbf{X} \succ 0\}$ where the affine-invariant metric $\langle \mathbf{U}, \mathbf{V} \rangle_{\mathbf{X}} = \operatorname{tr}(\mathbf{X}^{-1}\mathbf{U}\mathbf{X}^{-1}\mathbf{V})$ is considered (Bhatia, 2009), for $\mathbf{U}, \mathbf{V} \in T_{\mathbf{X}} \mathbb{S}^d_{++}$. The exponential map is given by $\operatorname{Exp}_{\mathbf{X}}(\mathbf{U}) = \mathbf{X}\operatorname{expm}(\mathbf{X}^{-1}\mathbf{U})$ where $\operatorname{expm}(\cdot)$ denotes the principal matrix exponential. The corresponding logarithm map is given by $\log_{\mathbf{X}}(\mathbf{Y}) = \mathbf{X}\operatorname{logm}(\mathbf{X}^{-1}\mathbf{Y})$. Its Riemannian gradient of a real-valued function $f$ is derived as $\operatorname{grad} f(\mathbf{X}) = \mathbf{X}\nabla f(\mathbf{X})\mathbf{X}$ and the Riemannian Hessian is $\operatorname{Hess} f(\mathbf{X})[\mathbf{U}] = \operatorname{Dgrad} f(\mathbf{X})[\mathbf{U}] - \{\mathbf{U}\mathbf{X}^{-1}\operatorname{grad} f(\mathbf{X})\}_{\mathrm{S}} = \mathbf{X}\nabla^2 f(\mathbf{X})[\mathbf{U}]\mathbf{X} + \{\mathbf{U}\nabla f(\mathbf{X})\mathbf{X}\}_{\mathrm{S}}$ where we use $\{\mathbf{A}\}_{\mathrm{S}} := (\mathbf{A} + \mathbf{A}^\top)/2$.

**Sphere manifold.** The sphere of size $d$ is $\mathcal{S}^d := \{\mathbf{x} \in \mathbb{R}^d : \mathbf{x}^\top \mathbf{x} = 1\}$. The sphere manifold is a Riemannian manifold with the Euclidean metric. The exponential map is given by $\operatorname{Exp}_{\mathbf{x}}(\mathbf{u}) = \cos(\|\mathbf{u}\|_2)\mathbf{x} + \sin(\|\mathbf{u}\|_2)\frac{\mathbf{u}}{\|\mathbf{u}\|_2}$ and the logarithm map is $\operatorname{Log}_{\mathbf{x}}(\mathbf{y}) = \arccos(\mathbf{x}^\top \mathbf{y})\frac{(\mathbf{I} - \mathbf{x}\mathbf{x}^\top)(\mathbf{y} - \mathbf{x})}{\|(\mathbf{I} - \mathbf{x}\mathbf{x}^\top)(\mathbf{y} - \mathbf{x})\|_2}$. The Riemannian gradient and Riemannian

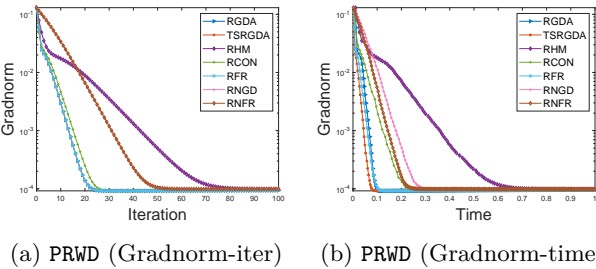

(a) `PRWD` (Gradnorm-iter)  (b) `PRWD` (Gradnorm-time)

Figure 6: Experiments on the projection robust Wasserstein distance (`PRWD`) computation problem on the fragmented hypercube. We see all methods, except RHM and RNFR, perform competitively.

Hessian are respectively $\text{grad}f(\mathbf{x}) = (\mathbf{I} - \mathbf{x}\mathbf{x}^\top)\nabla f(\mathbf{x})$ and $\text{Hess}f(\mathbf{x})[\mathbf{u}] = (\mathbf{I} - \mathbf{x}\mathbf{x}^\top)\nabla^2 f(\mathbf{x})[\mathbf{u}] - \mathbf{x}^\top\nabla f(\mathbf{x})\mathbf{u}$ for any $\mathbf{u} \in T_\mathbf{x}\mathcal{S}^d$.

**Stiefel manifold.** The Stiefel manifold $\text{St}(d,r) \coloneqq \{\mathbf{X} \in \mathbb{R}^{d\times r} : \mathbf{X}^\top\mathbf{X} = \mathbf{I}\}$ with the Euclidean inner product. We consider the QR-based retraction in the experiment, which is $\text{Retr}_\mathbf{X}(\mathbf{U}) = \text{qf}(\mathbf{X} + \mathbf{U})$ where $\text{qf}(\cdot)$ extracts the Q-factor from the QR decomposition. Let the orthogonal projection to the tangent space be denoted as $\text{P}_\mathbf{X}(\mathbf{U}) = \mathbf{U} - \mathbf{X}\{\mathbf{X}^\top\mathbf{U}\}_\text{S}$. Then, the Riemannian gradient and Riemannian Hessian are given by $\text{grad}f(\mathbf{X}) = \text{P}_\mathbf{X}(\nabla f(\mathbf{X}))$ and $\text{Hess}f(\mathbf{X})[\mathbf{U}] = \text{P}_\mathbf{X}(\nabla^2 f(\mathbf{X})[\mathbf{U}] - \mathbf{U}\{\mathbf{X}^\top\nabla f(\mathbf{X})\}_\text{S})$.

**Doubly stochastic manifold.** The doubly stochastic manifold (or coupling manifold) between two discrete probability measures $\mu, \nu$ with marginals $\mathbf{a} \in \mathbb{R}^m, \mathbf{b} \in \mathbb{R}^n$ is the set $\Pi(\mu,\nu) = \{\boldsymbol{\Gamma} \in \mathbb{R}^{m\times n} : \Gamma_{ij} > 0, \boldsymbol{\Gamma}\mathbf{1}_n = \mathbf{a}, \boldsymbol{\Gamma}^\top\mathbf{1}_m = \mathbf{b}\}$. It can be equipped with the Fisher information metric, defined as $\langle\mathbf{U}, \mathbf{V}\rangle_\boldsymbol{\Gamma} = \sum_{i,j}(U_{ij}V_{ij})/\Gamma_{ij}$ for any $\mathbf{U}, \mathbf{V} \in T_\boldsymbol{\Gamma}\Pi(\mu,\nu)$. The retraction is given by $\text{Retr}_\boldsymbol{\Gamma}(\mathbf{U}) = \text{Sinkhorn}(\boldsymbol{\Gamma} \odot \exp(\mathbf{U} \oslash \boldsymbol{\Gamma}))$ where $\exp, \odot, \oslash$ are elementwise exponential, product, and division operations.

## D  Additional experiments on projection robust Wasserstein distance problem

We consider the problem of computing the subspace robust Wasserstein distance between two discrete probability measures (Paty & Cuturi, 2019; Lin et al., 2020a; Huang et al., 2021), i.e.,

$$\min_{\boldsymbol{\Gamma}\in\Pi(\mu,\nu)} \max_{\mathbf{U}\in\text{St}(d,r)} \sum_{i=1}^m\sum_{j=1}^n \Big(\frac{1}{mn}\Gamma_{ij}\|\mathbf{U}^\top\mathbf{x}_i - \mathbf{U}^\top\mathbf{y}_j\|_2^2 + \epsilon\,\Gamma_{i,j}\big(\log(\Gamma_{ij}) - 1\big)\Big),$$

where $\mu = \sum_{i=1}^m a_i\delta_{\mathbf{x}_i}, \nu = \sum_{j=1}^n b_j\delta_{\mathbf{y}_j}$ are discrete probability measures. We denote $\Pi(\mu,\nu) = \{\boldsymbol{\Gamma} \in \mathbb{R}^{m\times n} : \Gamma_{ij} > 0, \sum_i\Gamma_{ij} = b_j, \forall j \in [n], \sum_j\Gamma_{ij} = a_i, \forall i \in [m]\}$ as the set of doubly stochastic matrices, which form a Riemannian manifold (Douik & Hassibi, 2019; Mishra et al., 2021). The Riemannian gradient and second-order derivatives are given in Lemma 8. The problem is nonconvex-nonconcave, and hence, we track the gradient norm for comparing convergence behaviors of various algorithms.

**Experiment setup and results.** We consider the same fragmented hypercube example as in Lin et al. (2020a); Han et al. (2023b), concerning a uniform distribution on $[-1, 1]^d$ and a push-forward map $T(\mathbf{x}) = \mathbf{x}+2\text{sign}(\mathbf{x})\odot(\sum_{i=1}^k \mathbf{e}_i)$, where $\text{sign}(\cdot)$ denotes the elementwise sign of $\mathbf{x}$ and $\mathbf{e}_i$ are the $i$-th canonical basis in $\mathbb{R}^d$. We consider the same choice of parameters in generating the samples as $d = 30, r = 5, k = 2, n = 100, \epsilon = 0.2$. We exclude RCEG in the experiment because there is no well-defined logarithm map (or inverse retraction) operations for double stochastic manifold. In Figures 6a, 6b, we see that all methods, except RNFR and RHM, perform competitively. Even though we see matching convergence in iterations, TSRGDA is slightly superior due to its less costly per-iteration computation.

## E  Useful results

This section derives several useful results for the analysis presented in this paper. Appendices E.1 and E.2 allow to relate the spectrum of any linear operator on tangent space to its matrix representation via a

coordinate transform. This is crucial for quantifying local minimax points. Appendix E.3 provides inverse and implicit function theorem on manifolds, which proves to be useful for relating dynamical systems under different representations (i.e., Proposition 6).

### E.1  Spectrum of tangent space linear operators

First, we recall the definition of eigenvalues and eigenvectors of a tangent space operator.

**Definition 6.** For some $x \in \mathcal{M}$, let $A : T_x\mathcal{M} \to T_x\mathcal{M}$ be a linear operator. A tuple $(\lambda, u)$ for some $\lambda \in \mathbb{R}, u \in T_x\mathcal{M}$ is called an eigenpair of $A$ if $A[u] = \lambda u$.

It is important to notice here that the eigenvalues need to be real. Thus, based on Definition 6, a general non-self-adjoint operator can have no eigenpair, unlike the case in Euclidean space where complex eigenvalues and eigenvectors are well-defined.

The next lemma shows that for eigenpairs that exist for a linear operator (not necessarily self-adjoint), they can be equivalently written in terms of a matrix representation of the operator on an orthonormal basis. It should be emphasized that only real eigenpairs are preserved when switching back and forth between the matrix representation and the abstract operator.

**Lemma 1** (Eigenpairs in basis). *Suppose we are given an orthonormal basis $(e_1, ..., e_d)$ of a tangent space $T_x\mathcal{M}$. Then for some $\lambda \in \mathbb{R}$, $u \in T_x\mathcal{M}$, we have $A[u] = \lambda u$ if and only if $\vec{A}\vec{u} = \lambda\vec{u}$ where $\vec{u} = [\langle u, e_i \rangle_x]_i, \vec{A} = [\langle e_i, A[e_j] \rangle_x]_{ij}$ are the vector/matrix representations in the basis.*

*Proof of Lemma 1.* Taking the inner product with $e_i, i = 1, ..., d$ for $A[u] = \lambda u$ on both sides gives $\langle A[u], e_i \rangle_x = \lambda\vec{u}_i$, for all $i$ where $\vec{u}_i = \langle u, e_i \rangle_x$ is the $i$-th component of vector $\vec{u}$. Hence it remains to show the $i$-th component of $\vec{A}\vec{u}$ is equal to $\langle A[u], e_i \rangle_x$. This can be seen from the linearity of $A$, i.e.,

$$[\vec{A}\vec{u}]_i = \sum_{j'=1}^{d} \langle e_i, A[e_{j'}] \rangle_x \langle u, e_{j'} \rangle_x = \left\langle e_i, A\Big[ \sum_{j'=1}^{d} \langle u, e_{j'} \rangle e_{j'} \Big] \right\rangle_x = \langle e_i, A[u] \rangle_x,$$

where we note $u = \sum_{i=1}^{d} \langle u, e_i \rangle_x e_i$. □

Furthermore, we can extend such a result to tangent space operators on a Riemannian product manifold.

**Lemma 2** (Eigenpairs in basis on product manifold). *Consider a product Riemannian manifold $\mathcal{M} = \mathcal{M}_x \times \mathcal{M}_y$ of dimension $d_x \times d_y$. Let a linear operator be $T : T_z\mathcal{M} \to T_z\mathcal{M}$ and we denote its elements $T_{xx} : T_x\mathcal{M} \to T_x\mathcal{M}, T_{yy} : T_y\mathcal{M} \to T_y\mathcal{M}, T_{xy} : T_y\mathcal{M} \to T_x\mathcal{M}, T_{yx} : T_x\mathcal{M} \to T_y\mathcal{M}$ be linear operators, not necessarily self-adjoint.*

*Suppose that we are given an orthonormal basis $\mathcal{B}_x = (e_1, ..., e_{d_x}), \mathcal{B}_y = (w_1, ..., w_{d_y})$. Also suppose that for some $\lambda \in \mathbb{R}, \xi \in T_z\mathcal{M}$ such that $(\lambda, \xi)$ is an eigenpair of $T_z$. Then, we have $T_z[\xi] = \lambda\xi$ if and only if $\vec{T_z}\vec{\xi} = \lambda\vec{\xi}$ where $\vec{\xi} = [\vec{u}, \vec{v}] \in \mathbb{R}^{d_x+d_y}$ where $\vec{u} = [\langle u, e_1 \rangle_x, ..., \langle u, e_{d_x} \rangle_x]$ and $\vec{v} = [\langle v, w_1 \rangle_y, ..., \langle v, w_{d_y} \rangle_y]$ and $\vec{T_z} = \begin{bmatrix} \vec{T}_{xx}, & \vec{T}_{xy} \\ \vec{T}_{yx}, & \vec{T}_{yy} \end{bmatrix} \in \mathbb{R}^{(d_x+d_y) \times (d_x+d_y)}$ with $\vec{T}_{xx} = [\langle e_i, T_{xx}[e_j] \rangle_x]_{ij}, \vec{T}_{xy} = [\langle e_i, T_{xy}[w_j] \rangle_x]_{ij}, \vec{T}_{yx} = [\langle w_i, T_{yx}[e_j] \rangle_y]_{ij}, \vec{T}_{yy} = [\langle w_i, T_{yy}[w_j] \rangle_x]_{ij}.$*

*Proof of Lemma 2.* If $(\lambda, \xi)$ is an eigenpair of $T_z$, then $T_z[\xi] = \lambda\xi$ and we set $\xi = (u, v)$ with $u = \sum_{i=1}^{d_x} \langle u, e_i \rangle_x e_i, v = \sum_{j=1}^{d_y} \langle v, w_j \rangle_y w_j$. Substituting the expressions to the $T_z[\xi]$ and taking inner product with

basis in $\mathcal{B}_x, \mathcal{B}_y$ respectively yields $\overrightarrow{T_z[\xi]} := \begin{bmatrix} \left[\overrightarrow{T_{xx}[u]} + \overrightarrow{T_{xy}[v]}\right] \\ \left[\overrightarrow{T_{yx}[u]} + \overrightarrow{T_{yy}[v]}\right] \end{bmatrix}$ where

$$
\begin{aligned}
\left[\overrightarrow{T_{xx}[u]} + \overrightarrow{T_{xy}[v]}\right]_i &= \left\langle e_i, T_{xx}\big[\sum_{j=1}^{d_x}\langle u, e_j\rangle_x e_j\big]\right\rangle_x + \left\langle e_i, T_{xy}\big[\sum_{j=1}^{d_y}\langle v, w_j\rangle_y w_j\big]\right\rangle_x \\
&= \left\langle e_i, \sum_{j=1}^{d_x}\langle u, e_j\rangle_x T_{xx}[e_j] + \sum_{j=1}^{d_y}\langle v, w_j\rangle_y T_{xy}[w_j]\right\rangle_x \\
&= \sum_{j=1}^{d_x}\langle e_i, T_{xx}[e_j]\rangle_x\langle u, e_j\rangle_x + \sum_{j=1}^{d_y}\langle e_i, T_{xy}[w_j]\rangle_x\langle v, w_j\rangle_y \\
&= \left[\vec{T}_{xx}[\vec{u}] + \vec{T}_{xy}[\vec{v}]\right]_i, \ \forall i \in [1, d_x], \\
\left[\overrightarrow{T_{yx}[u]} + \overrightarrow{T_{yy}[v]}\right]_i &= \left[\vec{T}_{yx}[\vec{u}] + \vec{T}_{yy}[\vec{v}]\right]_i, \ \forall i \in [d_x + 1, d_y],
\end{aligned}
$$

where we apply the linearity of the operators. Hence, $\overrightarrow{T_z[\xi]} = \vec{T}_z\vec{\xi}$. Similarly taking the inner product for the RHS (i.e., $\lambda\xi$), we have $\overrightarrow{\lambda\xi} = \lambda\vec{\xi}$. For the converse, we can show if $\vec{T}_z\vec{\xi} = \lambda\vec{\xi}$, $T_z\xi = \lambda\xi$ using similar arguments via linearity of operators. $\qquad\square$

We also remind of a result for similar operators.

**Proposition 14** (Similar operators). *Consider a Riemannian manifold $\mathcal{M}$ and $x \in \mathcal{M}$, let $A, B$ be linear operators on $T_x\mathcal{M}$. We say $A, B$ are similar if there exists a non-degenerate operator $C$ such that $A = C^{-1} \circ B \circ C^{-1}$. In addition, if $A, B$ are similar, then they share the same (real) eigenvalues.*

### E.2 Coordinate transform

**Lemma 3** (Coordinate transform is isometric isomorphism). *Given an orthonormal basis $\{e_1, ..., e_d\}$ on a tangent space $T_z\mathcal{M}$, the coordinate transform $\phi : T_x\mathcal{M} \to \mathbb{R}^d$, defined by $\phi(u) = [\langle u, e_i\rangle_z]_{i=1}^d$ is an isometric isomorphism (i.e., a bijective isometry) between the tangent space and the Euclidean space.*

*Proof of Lemma 3.* First, given the coordinate transform, we can write for any $u \in T_z\mathcal{M}$, $u = \sum_{i=1}^d\langle u, e_i\rangle_z e_i$. The mapping $\phi$ it is invertible and we can write the inverse $\phi^{-1}(\vec{u}) = \sum_{i=1}^d \vec{u}_i e_i$ such that $(\phi^{-1} \circ \phi)(u) = \sum_{i=1}^d\langle u, e_i\rangle_z u = u$ and $(\phi \circ \phi^{-1})(\vec{u}) = \vec{u}$. In addition, we have for any $u, v \in T_z\mathcal{M}$, $\vec{u}, \vec{v} \in \mathbb{R}^d$,

$$
\langle \phi(u), \phi(v)\rangle_2 = \sum_{i=1}^d\langle u, e_i\rangle_z\langle v, e_i\rangle_z = \langle u, \sum_{i=1}^d\langle v, e_i\rangle_z e_i\rangle_z = \langle u, v\rangle_z,
$$

$$
\langle \phi^{-1}(\vec{u}), \phi^{-1}(\vec{v})\rangle_z = \langle \sum_{i=1}^d\vec{u}_i e_i, \sum_{j=1}^d\vec{v}_j e_j\rangle_z = \sum_{i=1}^d\vec{u}_i\vec{v}_i = \langle \vec{u}, \vec{v}\rangle_2,
$$

where we use the orthonormality of the basis. $\qquad\square$

### E.3 Inverse and implicit function theorem on manifolds

**Theorem 4** (Inverse function theorem on smooth manifolds (Boumal, 2020, Theorem 4.16)). *Let $G : \mathcal{M}_x \to \mathcal{M}_y$ be a smooth map on two smooth manifolds $\mathcal{M}_x, \mathcal{M}_y$. If $\mathrm{D}G(x)$ is invertible at $x \in \mathcal{M}_x$, then there exists an open subset $\mathcal{U}_x \subset \mathcal{M}_x$ that contains $x$ and $\mathcal{U}_y \subset \mathcal{M}_y$ that contains $G(x)$ such that $G|_{\mathcal{U}_x} : \mathcal{U}_x \to \mathcal{U}_y$ is a diffeomorphism.*

**Theorem 5** (Implicit function theorem on smooth manifolds). *Consider a differentiable function $F : \mathcal{M}_x \times \mathcal{M}_y \to T\mathcal{M}_y$ defined on the smooth product manifolds $\mathcal{M}_x \times \mathcal{M}_y$. Suppose $z^* = (x^*, y^*) \in \mathcal{M}_x \times \mathcal{M}_y$ such that $F(x^*, y^*) = 0$ and $\boldsymbol{\nabla}_y F(x^*, y^*)$ is invertible where $\boldsymbol{\nabla}_y$ denotes the covariant derivative on $\mathcal{M}_y$.*

*Then, there exists an open subset $\mathcal{U} = \mathcal{U}_x \times \mathcal{U}_y \subset \mathcal{M}_x \times \mathcal{M}_y$ where $x^* \in \mathcal{U}_x, y^* \in \mathcal{U}_y$ and a differentiable function $\Psi : \mathcal{M}_x \to \mathcal{M}_y$ such that $\Psi(x^*) = y^*$ and for all $x \in \mathcal{U}_x$, $\Psi(x)$ is the unique point in $\mathcal{U}_y$ such that $F(x, \Psi(x)) = 0$. In addition, we have $\mathrm{D}_x\Psi(x^*) = -\left((\boldsymbol{\nabla}_y F)^{-1} \circ \mathrm{D}_x F\right)(z^*)$.*

*Proof of Theorem 5.* The proof is motivated by (Lebl, 2009, Theorem 8.5.6) in Euclidean space. We start by defining a map $\bar{F} : \mathcal{U} \to \mathcal{M}_x \times T\mathcal{M}_y$ as $\bar{F}(x, y) := (x, F(x, y))$. We recall that the tangent bundle $T\mathcal{M}_y$ is also a smooth manifold, and therefore, $\mathcal{M}_x \times T\mathcal{M}_y$ too is a smooth product manifold. The derivative of map $\bar{F}$, for any $\xi = (u, v) \in T_z\mathcal{U}$, is derived as $\mathrm{D}\bar{F}(z) : T_z\mathcal{U} \to T_x\mathcal{M}_x \times T_{F(x,y)}\mathcal{M}_y$

$$\mathrm{D}\bar{F}(z)[\xi] = \begin{pmatrix} u \\ \mathrm{D}_x F(x, y)[u] + \boldsymbol{\nabla}_y F(x, y)[v] \end{pmatrix}.$$

Evaluated at $z^*$, it is easy to see when $\mathrm{D}\bar{F}(z^*)[\xi] = 0$ if and only if $\xi = 0$ (because $\boldsymbol{\nabla}_y F(z^*)$ is invertible). Thus, it follows that $\mathrm{D}\bar{F}(z)$ is invertible. By the inverse function theorem (Theorem 4), there exists an open subset $\mathcal{V}$ that contains $\bar{F}(z^*) = (x^*, 0)$ and $\bar{F}|_{\mathcal{U}} : \mathcal{U} \to \mathcal{V}$ is invertible. That is, for any $(x, v) \in \mathcal{V}$, we have $\bar{F}(\bar{F}^{-1}(x, v)) = (x, v)$. Let $\bar{F}^{-1} = (\bar{F}_1^{-1}, \bar{F}_2^{-1})$ such that $\bar{F}(\bar{F}_1^{-1}(x, v), F(\bar{F}_1^{-1}(x, v), \bar{F}_2^{-1}(x, v))) = (x, v)$. This leads to $F(x, \bar{F}_2^{-1}(x, v)) = v$ for any $v \in \mathcal{V}$. Set $v = 0$ yields $F(x, \bar{F}_2^{-1}(x, 0)) = 0$. Let $\Psi(x) := \bar{F}_2^{-1}(x, 0)$ and it can be verified that $\Psi(x)$ is one-to-one as $\bar{F}_2^{-1}(x, 0)$ is one-to-one and hence the uniqueness.

Finally, we take derivative of $F(x, \Psi(x))$, which is the zero map. This leads to

$$0 = \mathrm{D}_x F(x, \Psi(x))[u] = \mathrm{D}_x F(x, \Psi(x))[u] + \boldsymbol{\nabla}_y F(x, \Psi(x))\big[\mathrm{D}_x\Psi(x)[u]\big]$$

for any $u \in T_x\mathcal{M}_x$. Thus it follows that for $x^*, y^* = \Psi(x^*)$, $\mathrm{D}_x\Psi(x^*) = -((\boldsymbol{\nabla}_y F)^{-1} \circ \mathrm{D}_x F)(z^*)$. $\qquad\square$

# F  Proofs of Section 4

*Proof of Proposition 2.* The proof can be trivially adapted from the Euclidean counterpart in Jin et al. (2020). $\qquad\square$

*Proof of Proposition 3.* (1) First, we prove the first-order condition. By Proposition 2, $y^*$ is the local maximizer of $f(x^*, y)$, and therefore (based on Proposition 4.5 in Boumal (2020)), we have $\mathrm{grad}_y f(x^*, y^*) = 0$. From the definition of a local minimax point, we know that $f(x^*, y^*) \leq \min_{x:\mathrm{dist}(x,x^*)\leq\delta} \max_{y:\mathrm{dist}(y,y^*)\leq h(\delta)} f(x, y)$ for any $\delta \leq \delta_0$. For any $x' = \mathrm{Exp}_{x^*}(u)$ with $\|u\|_{x^*} \leq \delta$, denote $y' = \arg\max_{y:\mathrm{dist}(y,y^*)\leq h(\delta)} f(x', y)$. Then,

$$0 \leq f(x', y') - f(x^*, y^*) = f(x', y') - f(x^*, y') + f(x^*, y') - f(x^*, y^*)$$
$$\leq f(x', y') - f(x^*, y') = \langle \mathrm{grad}_x f(x^*, y'), u \rangle_{x^*} + o(\|u\|_{x^*}) = \langle \mathrm{grad}_x f(x^*, y^*), u \rangle_{x^*} + o(\|u\|_{x^*}).$$

Since the inequality holds for any $u$ with $\|u\|_{x^*} \to 0$, we have $\mathrm{grad}_x f(x^*, y^*) = 0$.

(2) We now prove for the second-order condition. By the fact that $y^*$ is a local maximizer of $f(x^*, y)$, we have by standard result $\mathrm{Hess}_y f(x^*, y^*) \preceq 0$. In addition, by Taylor expansion on $f$, we have

$$f(\mathrm{Exp}_{x^*}(u_x), \mathrm{Exp}_{y^*}(u_y)) = f(x^*, y^*) + \frac{1}{2}\langle u_x, \mathrm{Hess}_x f(x^*, y^*)[u_x] \rangle_{x^*} + \langle u_x, \mathrm{grad}_{xy}^2 f(x^*, y^*)[u_y] \rangle_{x^*}$$
$$+ \frac{1}{2}\langle u_y, \mathrm{Hess}_y f(x^*, y^*)[u_y] \rangle_{y^*} + o(\|u_x\|_{x^*}^2 + \|u_y\|_{y^*}^2),$$

where we use the fact that Riemannian cross derivatives are adjoint with respect to the Riemannian metric. Since $\mathrm{Hess}_y f(x^*, y^*)$ is invertible, define $\tilde{h}(\delta) := 2\|[\mathrm{Hess}_y f(x^*, y^*)]^{-1} \circ \mathrm{grad}_{yx}^2 f(x^*, y^*)\|_{y^*}\delta$, that satisfies $\tilde{h}(\delta) \to 0$ as $\delta \to 0$. Hence it can be shown that $-([\mathrm{Hess}_y f(x^*, y^*)]^{-1} \circ \mathrm{grad}_{yx}^2 f(x^*, y^*))[u_x] + o(\|u_x\|_{x^*}) = \arg\max_{\|u_y\|_{y^*} \leq \max\{h(\delta), \tilde{h}(\delta)\}} f(\mathrm{Exp}_{x^*}(u_x), \mathrm{Exp}_{y^*}(u_y))$ for any $h(\delta)$. By definition of $(x^*, y^*)$ as local minimax point,

$$0 \leq \max_{\|u_y\|_{y^*} \leq h(\delta)} f(\mathrm{Exp}_{x^*}(u_x), \mathrm{Exp}_{y^*}(u_y)) - f(x^*, y^*)$$
$$\leq \max_{\|u_y\|_{y^*} \leq \max\{h(\delta), \tilde{h}(\delta)\}} f(\mathrm{Exp}_{x^*}(u_x), \mathrm{Exp}_{y^*}(u_y)) - f(x^*, y^*)$$
$$= \frac{1}{2}\langle u_x, \left(\mathrm{Hess}_x f(x^*, y^*) - \mathrm{grad}_{xy}^2 f(x, y) \circ [\mathrm{Hess}_y f(x, y)]^{-1} \circ \mathrm{grad}_{yx}^2 f(x, y)\right)[u_x] \rangle_{x^*} + o(\|u_x\|_{x^*}^2).$$

Since the inequality holds for any $u_x$, the proof is complete. $\qquad\square$

**Proposition 15** (Sufficient conditions for local Nash saddle points and local minimax points). *Let a point* $(x^*, y^*)$ *satisfy: (1)* $\mathrm{grad}_x f(x^*, y^*) = 0, \mathrm{grad}_y f(x^*, y^*) = 0$ *and (2)* $\mathrm{Hess}_x f(x^*, y^*) \succ 0, \mathrm{Hess}_y f(x^*, y^*) \prec 0$. *Then, the point is a (strict) local Nash saddle point.*

*If a point* $(x^*, y^*)$ *satisfies* $\mathrm{grad}_x f(x^*, y^*) = 0, \mathrm{grad}_y f(x^*, y^*) = 0$ *and* $\mathrm{Hess}_y f(x^*, y^*) \prec 0$, $\mathrm{Hess}_x f(x, y) - \mathrm{grad}^2_{xy} f(x, y) \circ [\mathrm{Hess}_y f(x, y)]^{-1} \circ \mathrm{grad}^2_{yx} f(x, y) \succ 0$, *then* $(x^*, y^*)$ *is a (strict) local minimax point.*

*Proof of Proposition 15.* The proof follows a similar vein from Proposition 3 and (Jin et al., 2020, Proposition 20). $\qquad\square$

*Proof of Proposition 4.* (1) By the definition of a global Nash saddle point, we see when $(\mathbf{X}^*, \mathbf{Y}^*) \in S^*$, $f(\mathbf{X}^*, \mathbf{Y}^*) = 0$ and it satisfies $-(\log\det(\mathbf{Y}))^2 = f(\mathbf{X}^*, \mathbf{Y}) \leq f(\mathbf{X}^*, \mathbf{Y}^*) \leq f(\mathbf{X}, \mathbf{Y}^*) = (\log\det(\mathbf{X}))^2$. Then, from (Jin et al., 2020, Proposition 11), we can conclude $(\mathbf{X}^*, \mathbf{Y}^*)$ is a local minimax point.

(2) For the second claim, we follow the definition of local minimax point in Definition 3. It is easy to verify that when $(\mathbf{X}^*, \mathbf{Y}^*) \in S^* \times S^*$, $-c_3(\log\det(\mathbf{Y}))^2 = f(\mathbf{X}^*, \mathbf{Y}) \leq f(\mathbf{X}^*, \mathbf{Y}^*) = 0$ for all $\mathbf{Y} \in \mathcal{Y}$. On the other hand, consider $\max_{\mathbf{Y}:\mathrm{dist}(\mathbf{Y}^*, \mathbf{Y}) \leq h(\delta)} f(\mathbf{X}, \mathbf{Y})$ for $\mathbf{X}$ such that $\mathrm{dist}(\mathbf{X}^*, \mathbf{X}) \leq \delta$. First we can derive the Riemannian Hessian as $\mathrm{Hess}_y f(\mathbf{X}, \mathbf{Y})[\mathbf{V}] = -2c_3\mathrm{tr}(\mathbf{Y}^{-1}\mathbf{V})\mathbf{Y}$ for any symmetric matrix $\mathbf{V}$. We can show the Riemannian Hessian is negative semidefinite with respect to the Riemannian metric, i.e., $\langle \mathrm{Hess}_y f(\mathbf{X}, \mathbf{Y})[\mathbf{V}], \mathbf{V} \rangle_{\mathbf{X}} = -2c_3(\mathrm{tr}(\mathbf{Y}^{-1}\mathbf{V}))^2 \leq 0$. Hence $f(\mathbf{X}, \mathbf{Y})$ is geodesic concave in $\mathbf{Y}$ (but not strongly concave, e.g. when $\mathbf{V} = 0$). Hence by first-order conditions and compactness of the domain, we can show the maximum is attained either when $\mathrm{grad}_y f(\mathbf{X}, \mathbf{Y}) = 0$ or on the boundary of the constraint set. The Riemannian gradient can be derived as $\mathrm{grad}_y f(\mathbf{X}, \mathbf{Y}) = (c_2\log\det(\mathbf{X}) - 2c_3\log\det(\mathbf{Y}))\mathbf{Y}$. Given $\mathbf{Y} \in \mathbb{S}^d_{++}$, $\mathrm{grad}_y f(\mathbf{X}, \mathbf{Y}) = 0$ if and only if $\frac{c_2}{2c_3}\log\det(\mathbf{X}) = \log\det(\mathbf{Y})$. The task is thus to show given certain $\delta, h(\delta)$, and $\mathbf{X}$ such that $\mathrm{dist}(\mathbf{X}^*, \mathbf{X}) \leq \delta$ we can always find $\mathbf{Y}$ satisfying $\mathrm{dist}(\mathbf{Y}^*, \mathbf{Y}) \leq h(\delta)$ such that $\mathrm{grad}_y f(\mathbf{X}, \mathbf{Y}) = 0$.

To see this, first we recall that the Riemannian distance is

$$\mathrm{dist}^2(\mathbf{A}, \mathbf{B}) = \|\mathrm{logm}(\mathbf{A}^{-1}\mathbf{B})\|_F^2 = \sum_{i=1}^d \big(\log\big(\lambda_i(\mathbf{A}^{-1}\mathbf{B})\big)\big)^2.$$

We notice that for any $\mathbf{A} \in S^*$, any $\mathbf{B} \in \mathbb{S}^d_{++}$, it satisfies $\log\det(\mathbf{A}^{-1}\mathbf{B}) = \log\det(\mathbf{A}^{-1}) + \log\det(\mathbf{B}) = \log\det(\mathbf{B})$. In addition, we notice that $\log\det(\mathbf{B}) = \sum_{i=1}^d \log(\lambda_i(\mathbf{B}))$. Hence, suppose we choose $\mathbf{Y}$ such that $\log\big(\lambda_i((\mathbf{Y}^*)^{-1}\mathbf{Y})\big) = \frac{c_2}{2c_3}\log(\lambda_i((\mathbf{X}^*)^{-1}\mathbf{X}))$ for all $i \in [d]$. Since both $\mathbf{X}^*, \mathbf{Y}^* \in S^*$, we can verify

$$\log\det(\mathbf{Y}) = \log\det((\mathbf{Y}^*)^{-1}\mathbf{Y}) = \frac{c_2}{2c_3}\log\det((\mathbf{X}^*)^{-1}\mathbf{X}) = \frac{c_2}{2c_3}\log\det(\mathbf{X}),$$

which satisfies $\mathrm{grad}_y f(\mathbf{X}, \mathbf{Y}) = 0$. Now we just need to show $\mathrm{dist}^2(\mathbf{Y}^*, \mathbf{Y})$ is bounded. To this end, we write

$$\mathrm{dist}^2(\mathbf{Y}^*, \mathbf{Y}) = \sum_{i=1}^d \big(\log\big(\lambda_i((\mathbf{Y}^*)^{-1}\mathbf{Y})\big)\big)^2 = \frac{c_2^2}{4c_3^2}\sum_{i=1}^d \big(\log\big(\lambda_i((\mathbf{X}^*)^{-1}\mathbf{X})\big)\big)^2 = \frac{c_2^2}{4c_3^2}\mathrm{dist}^2(\mathbf{X}^*, \mathbf{X}).$$

Hence, we can simply choose $h(\delta) = \frac{c_2}{2c_3}\delta$, and under such choice, we can always find $\mathbf{Y}$ given $\mathbf{X}$ under the constraints such that $\mathrm{grad}_y f(\mathbf{X}, \mathbf{Y}) = 0$. Such choice leads to $\phi(\mathbf{X}) := \max_{\mathbf{Y}:\mathrm{dist}(\mathbf{I}, \mathbf{Y}) \leq h(\delta)} f(\mathbf{X}, \mathbf{Y}) = (\frac{c_2^2}{4c_3} - c_1)(\log\det(\mathbf{X}))^2$. When $c_2 \geq 2\sqrt{c_1 c_3}$, we have $\frac{c_2^2}{4c_3} - c_1 \geq 0$. Hence $\phi(\mathbf{X}) \geq f(\mathbf{X}^*, \mathbf{Y}^*) = 0$.

Finally, to show it is not a local Nash saddle point, it suffices to see that $\langle \mathbf{U}, \mathrm{Hess}_x f(\mathbf{X}^*, \mathbf{Y}^*)[\mathbf{U}] \rangle_{\mathbf{X}^*} = -2c_1(\mathrm{tr}(\mathbf{U}))^2 \leq 0$, which violates the second-order characterization of local Nash saddle points (Proposition 1).

(3) For the third claim, first we see that $f(\mathbf{X}^*, \mathbf{Y}) \leq f(\mathbf{X}^*, \mathbf{Y}^*) = 0$ for all $\mathbf{Y} \in \mathcal{Y}$. Then, we consider $\max_{\mathbf{Y}:\mathrm{dist}(\mathbf{I}, \mathbf{Y}) \leq h(\delta)} f(\mathbf{X}, \mathbf{Y})$ for $\mathbf{X}$ with $\mathrm{dist}(\mathbf{I}, \mathbf{X}) \leq \delta$. From the standard result (e.g. (Alimisis et al., 2020, Lemma 2)), we notice that the $\mathrm{Hess}_y f(\mathbf{X}, \mathbf{Y})[\mathbf{V}]$ is negative definite for SPD manifold with affine-invariant

metric (because the curvature is negative). Hence $f(\mathbf{X}, \mathbf{Y})$ is geodesic strongly concave in $\mathbf{Y}$. We similarly take the first-order conditions where $\mathrm{grad}_y f(\mathbf{X}, \mathbf{Y})$ is computed as

$$\mathrm{grad}_y f(\mathbf{X}, \mathbf{Y}) = c_2 \log\det(\mathbf{X})\mathbf{Y} + 2c_3 \mathrm{Log}_{\mathbf{Y}}(\mathbf{I}) = c_2 \log\det(\mathbf{X})\mathbf{Y} + 2c_3\{\mathbf{Y}\mathrm{logm}(\mathbf{Y}^{-1})\}_{\mathrm{S}}$$
$$= \mathbf{O}\big(c_2 \log\det(\mathbf{X})\mathbf{\Lambda} - 2c_3\mathbf{\Lambda}\log(\mathbf{\Lambda})\big)\mathbf{O}^\top,$$

where we consider the eigendecomposition of $\mathbf{Y} = \mathbf{O}\mathbf{\Lambda}\mathbf{O}^\top$ ($\mathbf{\Lambda}$ is a positive diagonal matrix) and denote $\{\mathbf{A}\}_{\mathrm{S}} := (\mathbf{A} + \mathbf{A}^\top)/2$ and $\mathrm{logm}(\mathbf{A})$ as the principal matrix logarithm of $\mathbf{A} \in \mathbb{S}_{++}^d$. We also use $\log(\mathbf{A})$ as elementwise logarithm. It can be verified that $\mathrm{grad}_y f(\mathbf{X}, \mathbf{Y}) = 0$ if and only if $c_2 \log\det(\mathbf{X})\mathbf{\Lambda} - 2c_3\mathbf{\Lambda}\log(\mathbf{\Lambda}) = 0$. Given $\mathbf{\Lambda} > 0$, this is equivalent to $c_2 \log\det(\mathbf{X})\mathbf{I} = 2c_3 \log(\mathbf{\Lambda})$. Hence the $\lambda_i(\mathbf{Y}) = \exp(\frac{c_2}{2c_3} \log\det(\mathbf{X}))$ for all $i$ if and only if $\mathrm{grad}_y f(\mathbf{X}, \mathbf{Y}) = 0$. Given that $\mathrm{dist}^2(\mathbf{I}, \mathbf{X}) = \sum_{i=1}^d \big(\log(\lambda_i(\mathbf{X}))\big)^2 \leq \delta^2$ and suppose $\lambda_i(\mathbf{Y}) = \exp(\frac{c_2}{2c_3} \log\det(\mathbf{X}))$. Then,

$$\mathrm{dist}^2(\mathbf{I}, \mathbf{Y}) = \sum_{i=1}^d \big(\log(\lambda_i(\mathbf{Y}))\big)^2 = \frac{dc_2^2}{4c_3^2}(\log\det(\mathbf{X}))^2 = \frac{dc_2^2}{4c_3^2}(\sum_{i=1}^d \log(\lambda_i(\mathbf{X})))^2 \leq \frac{d^2 c_2^2}{4c_3^2}\delta^2,$$

where we use the fact that $(\sum_{i=1}^d a_i)^2 \leq d\sum_{i=1}^d a_i^2$ in the last inequality. Let $h(\delta) = \frac{dc}{2}\delta$. Then, we can always ensure $\mathrm{grad}_y f(\mathbf{X}, \mathbf{Y}) = 0$ by choosing $\mathbf{Y}$ with $\lambda_i(\mathbf{Y}) = \exp(\frac{c}{2} \log\det(\mathbf{X}))$, for all $i$.

Under a such choice of $\mathbf{Y}$, we obtain $\log\det(\mathbf{Y}) = \frac{dc_2}{2c_3} \log\det(\mathbf{X})$ and $\mathrm{dist}^2(\mathbf{I}, \mathbf{Y}) = \frac{dc_2^2}{4c_3^2}(\log\det(\mathbf{X}))^2$, which gives

$$\phi(\mathbf{X}) = \max_{\mathbf{Y}:\mathrm{dist}(\mathbf{I},\mathbf{Y})\leq h(\delta)} f(\mathbf{X}, \mathbf{Y}) = -c_1(\log\det(\mathbf{X}))^2 + \frac{dc_2^2}{2c_3}(\log\det(\mathbf{X}))^2 - \frac{dc_2^2}{4c_3}(\log\det(\mathbf{X}))^2$$
$$= (\frac{dc_2^2}{4c_3} - c_1)(\log\det(\mathbf{X}))^2.$$

When $\frac{dc_2^2}{4c_3} - c_1 \geq 0$, we have $\phi(\mathbf{X}) \geq 0 = f(\mathbf{X}^*, \mathbf{Y}^*)$. Hence, the proof is complete. $\qquad\square$

## G  Proofs of Section 5

*Proof of Proposition 5.* First, we study the linearization of the dynamics $\mathrm{Log}_{z^*}(z(t))$ on $T_{z^*}\mathcal{M}$ as follows.

$$\frac{d}{dt}\mathrm{Log}_{z^*}(z(t)) = \mathrm{DLog}_{z^*}(z(t))[\dot{z}(t)] = \mathrm{DLog}_{z^*}(z(t))[G(z(t))] = \big(\mathrm{DLog}_{z^*}(z(t)) \circ \Gamma_{z^*}^{z(t)}\big)[\Gamma_{z(t)}^{z^*}G(z(t))]$$
$$= \Gamma_{z(t)}^{z^*}G(z(t)) + o(\|\mathrm{Log}_{z^*}(z(t))\|_{z^*}), \quad (8)$$

where the last equality is from the proof of (Han et al., 2023a, Lemma 3) where it has been shown that $\mathrm{DLog}_{z^*}(z(t)) \circ \Gamma_{z^*}^{z(t)}$ is locally identity.

Next from Taylor expansion on $G(z(t))$, we obtain

$$\Gamma_{z(t)}^{z^*}G(z(t)) = \boldsymbol{\nabla}G(z^*)[\mathrm{Log}_{z^*}(z(t))] + o(\|\mathrm{Log}_{z^*}(z(t))\|_{z^*}), \quad (9)$$

where we notice $G(z^*) = 0$. Finally combining the results (9) with (8) gives the first result. Next,

$$\frac{d}{dt}V(z(t)) = 2\big\langle \mathrm{Log}_{z^*}(z(t)), H\big[\frac{d}{dt}\mathrm{Log}_{z^*}(z(t))\big]\big\rangle_{z^*}$$
$$= 2\big\langle \mathrm{Log}_{z^*}(z(t)), H\big[\boldsymbol{\nabla}G(z^*)[\mathrm{Log}_{z^*}(z(t))]\big]\big\rangle_{z^*} + o(\|\mathrm{Log}_{z^*}(z(t))\|_{z^*}^2)$$
$$= \big\langle \mathrm{Log}_{z^*}(z(t)), \big(H \circ \boldsymbol{\nabla}G(z^*) + (\boldsymbol{\nabla}G(z^*))^\dagger \circ H\big)\big[\mathrm{Log}_{z^*}(z(t))\big]\big\rangle_{z^*} + o(\|\mathrm{Log}_{z^*}(z(t))\|_{z^*}^2),$$

where $(\boldsymbol{\nabla}G(z^*))^\dagger$ represents the adjoint operator of $\boldsymbol{\nabla}G(z^*)$. $\qquad\square$

*Proof of Proposition 6.* From Proposition 5, we consider the linear dynamical system locally as $\frac{d}{dt}\text{Log}_{z^*}(z(t)) = \boldsymbol{\nabla} F(z^*)[\text{Log}_{z^*}(z(t))]$. Let $\phi : T_{z^*}\mathcal{M} \to \mathbb{R}^d$ be a coordinate transform via an orthonormal basis, which represents tangent vectors in terms of its coordinates. It can be easily verified that $\phi$ defines an isomorphism between the tangent space and the $\mathbb{R}^d$. Thus under the coordinate system, let $u(t) := \phi(\text{Log}_{z^*}(z(t))$ and then the dynamical system can be described as

$$\dot{u}(t) = \left(\text{D}\phi(\phi^{-1}(u(t))) \circ \boldsymbol{\nabla} F(z^*)\right)[(\phi^{-1}(u(t))],$$

with an equilibrium point $u^* = \phi(0_{z^*})$, where $0_{z^*}$ denotes the 0 tangent vector at $T_{z^*}\mathcal{M}$. Consider a Lyapunov function $V(\text{Log}_{z^*}(z(t))) = \langle \text{Log}_{z^*}(z(t)), H[\text{Log}_{z^*}(z(t))] \rangle_{z^*}$. The induced Lyapunov function under the coordinate system is given by $V_\phi(u(t)) := V(\phi^{-1}(u(t)))$. We can show that

$$\begin{aligned}
\frac{d}{dt}V_\phi(u(t)) &= \left(\text{D}V(\phi^{-1}(u(t))) \circ \text{D}\phi^{-1}(u(t))\right)[\dot{u}(t)] \\
&= \left(\text{D}V(\phi^{-1}(u(t))) \circ [\text{D}\phi(\phi^{-1}(u(t)))]^{-1}\right)[\dot{u}(t)] \\
&= \left(\text{D}V(\phi^{-1}(u(t))) \circ [\text{D}\phi(\phi^{-1}(u(t)))]^{-1} \circ \text{D}\phi(\phi^{-1}(u(t))) \circ \boldsymbol{\nabla} F(z^*)\right)[(\phi^{-1}(u(t))] \\
&= \left(\text{D}V(\phi^{-1}(u(t))) \circ \boldsymbol{\nabla} F(z^*)\right)[(\phi^{-1}(u(t))] = \frac{d}{dt}V((\phi^{-1}(u(t))),
\end{aligned}$$

where the second equality is due to the inverse function theorem. Hence we see the stability analysis of $z(t)$ is equivalent to that of $u(t)$.

Finally, we notice that $\phi$ is linear and is independent of $t$. Hence, we have $\dot{u}(t) = \left(\phi \circ \boldsymbol{\nabla} F(z^*) \circ \phi^{-1}\right)[u(t)]$. Let $G := \phi \circ \boldsymbol{\nabla} F(z^*) \circ \phi^{-1}$ be the matrix representation of $\boldsymbol{\nabla} F(z^*)$ under the coordinate system. Hence the dynamical system of interest is completely described in Euclidean space as $\dot{u}(t) = G[u(t)]$. By Definition 4, $u^* = \phi(0_{z^*})$ is strictly stable if there exists a positive definite matrix $H$ such that $HG + G^\top H$ is negative definite. This is equivalent to requiring $G$ to have all real parts of its eigenvalues negative (see for example (Khalil, 2002, Theorem 3.6)). $\qquad\square$

*Proof of Proposition 7.* The first result follows similarly from the proof of (Han et al., 2023a, Lemma 3). And the second result follows by substituting the first result in the definition of the Lyapunov function. $\qquad\square$

*Proof of Proposition 8.* The proof follows similarly as the proof of Proposition 6 and applies the equivalence between the eigenvalue and matrix definiteness characterization of stability (see for example (Khalil, 2002, Exercise 3.52)). $\qquad\square$

*Proof of Proposition 9.* First (1)$\Leftrightarrow$(2) is verified in Proposition 8. Now we verify the equivalence (2)$\Leftrightarrow$(3). First, when (2) holds, let $\lambda_i = a_i + b_i\text{i}$ be the eigenvalues of matrix representation of $\boldsymbol{\nabla} G(z^*)$. Then (2) states $|1 + \eta a_i + \eta b_i\text{i}| = \sqrt{(1 + \eta a_i)^2 + \eta^2 b_i^2} < 1$ for all $i$. Given $\eta > 0$, we must have $a_i < 0$ for all $i$. For the converse, when (3) holds, i.e., all $a_i < 0$, then we can see $(1 + \eta a_i)^2 + \eta^2 b_i^2 = 1 + 2\eta a_i + \eta^2(a_i^2 + b_i^2) = 1 + 2\eta a_i + o(\eta)$. Hence there always exists a sufficiently small $\eta > 0$ such that $1 + 2\eta a_i + o(\eta) < 1$ and (2) is satisfied. $\qquad\square$

*Proof of Proposition 10.* We start from the linearization of the system on $T_{z^*}\mathcal{M}$ as $\Delta_{k+1} = (\text{id} + \eta\boldsymbol{\nabla} G(z^*))[\Delta_k] + o(\|\Delta_k\|_{z^*})$, where we let $\Delta_k := \text{Log}_{z^*}(z_k) \in T_{z^*}\mathcal{M}$. Then we consider the coordinate transform $\phi$ under an orthonormal basis where we let $\overrightarrow{\Delta}_k := \phi(\Delta_k)$. By the isometric property of such coordinate transform (Lemma 3), we have the equivalent characterization of the system as $\overrightarrow{\Delta}_{k+1} = (I + \eta\overrightarrow{\boldsymbol{\nabla} G(z^*)})\overrightarrow{\Delta}_k + o(\|\overrightarrow{\Delta}_k\|_2)$ where we notice that $\|\overrightarrow{\Delta}_k\|_2 = \|\Delta_k\|_{z^*}$. By the definition of the asymptotic convergence rate (Definition 5) and following (Wang et al., 2019, Proposition 4), we have $\|\overrightarrow{\Delta}_k\|_2 \le C(1 - \kappa/2)^k\|\overrightarrow{\Delta}_0\|_2$. Finally, applying the inverse coordinate transform and using the isometric property again, we have the desired result. $\qquad\square$

## H   Proofs of Section 5.4

*Proof of Proposition 11.* To show $\mathtt{LNS} \subset \mathtt{TSRGDA}$, it suffices to write out the matrix representation of $\boldsymbol{\nabla} G_\tau(z^*)$. We consider an orthonormal basis on tangent space of product manifold as $\mathcal{B} = \mathcal{B}_x \times \mathcal{B}_y$ where $\mathcal{B}_x = (e_1, ..., e_{d_x})$ and $\mathcal{B}_y = (w_1, ..., w_{d_y})$, each orthonormal. From Lemma 2, we can write the matrix representation of $\boldsymbol{\nabla} G_\tau(z^*)$ as $\begin{bmatrix} -\tau H_{xx} & -\tau H_{xy} \\ H_{yx} & H_{yy} \end{bmatrix}$ where $H_{xx} := [\langle e_i, \mathrm{Hess}_x f(z^*)[e_j]\rangle_x]_{ij}$, $H_{xy} := [\langle e_i, \mathrm{grad}^2_{xy} f(z^*)[w_j]\rangle_x]_{ij}$, $H_{yx} := [\langle w_i, \mathrm{grad}^2_{yx} f(z^*)[e_j]\rangle_y]_{ij}$, $H_{yy} := [\langle w_i, \mathrm{Hess}_y f(z^*)[w_j]\rangle_y]_{ij}$. Now since $z^* \in \mathtt{LNS}$, $\mathrm{Hess}_x f(z^*) \succ 0$ and $\mathrm{Hess}_y f(z^*) \prec 0$ and by Lemma 1, we see all (real) eigenvalues of $\mathrm{Hess}_x f(z^*), \mathrm{Hess}_y f(z^*)$ are equal to that of $H_{xx}, H_{yy}$ respectively. Thus we have $H_{xx} \succ 0, H_{yy} \prec 0$ in the matrix sense. Therefore, the following proof readily follows from that of (Jin et al., 2020, Proposition 26). For the converse, since Euclidean space is a special case of Riemannian manifold, it suffices to consider the counter-example given in (Jin et al., 2020, Proposition 26).

For the second claim, the counter-examples given in (Jin et al., 2020, Proposition 27) suffice for the purpose. $\qquad\square$

*Proof of Proposition 12.* Similar to Proposition 11, we adopt the matrix representation and the subsequent proof follows from (Fiez & Ratliff, 2021, Theorem 1). $\qquad\square$

*Proof of Theorem 1.* The asymptotic convergence rate is readily obtained from Definition 5. The requirement on $\eta$ is to ensure the convergence rate is smaller than 1, i.e., $|1 + \eta \nu_i| < 1$, for all $\nu_i$. $\qquad\square$

## I   Proofs of Section 6

*Proof of Theorem 2.* First from Proposition 9, we start by writing the covariant derivative at $z^*$ for RTGDA as $\boldsymbol{\nabla}\big(G_{\mathrm{RTGDA}}[\mathrm{grad} f]\big)(z^*) = G_{\mathrm{RTGDA}}(z^*)[\mathrm{Hess} f(z^*)]$ where

$$G_{\mathrm{RTGDA}}(z^*)[\mathrm{Hess} f(z^*)] = \begin{pmatrix} -\tau \mathrm{id} & \tau\big(\mathrm{grad}^2_{xy} f \circ \mathrm{Hess}^{-1}_y f\big)(z^*) \\ 0 & \mathrm{id} \end{pmatrix} \begin{pmatrix} \mathrm{Hess}_x f(z^*) & \mathrm{grad}^2_{xy} f(z^*) \\ \mathrm{grad}^2_{yx} f(z^*) & \mathrm{Hess}_y f(z^*) \end{pmatrix}$$
$$= \begin{pmatrix} -\tau(\mathrm{Hess}_x f - \mathrm{grad}^2_{xy} f \circ \mathrm{Hess}^{-1}_y f \circ \mathrm{grad}^2_{yx} f)(z^*) & 0 \\ \mathrm{grad}^2_{yx} f(z^*) & \mathrm{Hess}_y f(z^*) \end{pmatrix}$$

By transforming the operator into its matrix representation and using Lemma 2 for eigenvalue equivalence in basis, we see the eigenvalues of $G_{\mathrm{RTGDA}}(z^*)[\mathrm{Hess} f(z^*)]$ are precisely the eigenvalues of $-\tau(\mathrm{Hess}_x f - \mathrm{grad}^2_{xy} f \circ \mathrm{Hess}^{-1}_y f \circ \mathrm{grad}^2_{yx} f)(z^*)$ and $\mathrm{Hess}_y f(z^*)$ (due to the block triangular structure). Hence if $z^*$ is a strictly local minimax point, then $\mathrm{Hess}_y f(z^*) \prec 0$ and $(\mathrm{Hess}_x f - \mathrm{grad}^2_{xy} f \circ \mathrm{Hess}^{-1}_y f \circ \mathrm{grad}^2_{yx} f)(z^*) \succ 0$, which suggests eigenvalues of $G_{\mathrm{RTGDA}}(z^*)[\mathrm{Hess} f(z^*)]$ are all negative. The converse is also true for the same reason.

For the RFR, the covariant derivative at $z^*$ is given by $\boldsymbol{\nabla}\big(G_{\mathrm{RFR}}[\mathrm{grad} f]\big)(z^*) = G_{\mathrm{RFR}}(z^*)[\mathrm{Hess} f(z^*)]$ where

$$G_{\mathrm{RFR}}(z^*)[\mathrm{Hess} f(z^*)] = \begin{pmatrix} -\mathrm{id} & 0 \\ (\mathrm{Hess}^{-1}_y f \circ \mathrm{grad}^2_{yx} f)(z^*) & \mathrm{id} \end{pmatrix} \begin{pmatrix} \tau\mathrm{Hess}_x f(z^*) & \tau\mathrm{grad}^2_{xy} f(z^*) \\ \mathrm{grad}^2_{yx} f(z^*) & \mathrm{Hess}_y f(z^*) \end{pmatrix}$$

which is similar to the following operator

$$\begin{pmatrix} -\mathrm{id} & 0 \\ (\mathrm{Hess}^{-1}_y f \circ \mathrm{grad}^2_{yx} f)(z^*) & \mathrm{id} \end{pmatrix} G_{\mathrm{RFR}}(z^*)[\mathrm{Hess} f(z^*)] \begin{pmatrix} -\mathrm{id} & 0 \\ (\mathrm{Hess}^{-1}_y f \circ \mathrm{grad}^2_{yx} f)(z^*) & \mathrm{id} \end{pmatrix}$$
$$= \begin{pmatrix} -\tau\big(\mathrm{Hess}_x f - \mathrm{grad}^2_{xy} \circ \mathrm{Hess}^{-1}_y f \circ \mathrm{grad}^2_{yx}\big)(z^*) & \tau\mathrm{grad}^2_{xy} f(z^*) \\ 0 & \mathrm{Hess}_y f(z^*) \end{pmatrix}.$$

By the eigenvalue property of similar operators (Proposition 14) as well as similar arguments for RTGDA, the eigenvalues of $G_{\mathrm{RFR}}(z^*)[\mathrm{Hess} f(z^*)]$ are the eigenvalues of $-\tau\big(\mathrm{Hess}_x f - \mathrm{grad}^2_{xy} \circ \mathrm{Hess}^{-1}_y f \circ \mathrm{grad}^2_{yx}\big)(z^*)$ and $\mathrm{Hess}_y f(z^*)$. Thus the same equivalence between strictly local minimax point and strictly stable fixed point.

To obtain the asymptotic convergence rate, we consider the discrete case where the dynamics are given by $z_{k+1} = \mathrm{Exp}_{z_k}\big(\eta\, G_{\mathrm{RFR}}(z_k)[\mathrm{grad} f(z_k)]\big)$ and $z_{k+1} = \mathrm{Exp}_{z_k}\big(\eta\, G_{\mathrm{RTGDA}}(z_k)[\mathrm{grad} f(z_k)]\big)$. First consider RTGDA and denote $\tilde{\lambda}_\ell$ as the eigenvalues of $\boldsymbol{\nabla}(G_{\mathrm{RTGDA}}[\mathrm{grad} f])(z^*)$. The asymptotic convergence rate is given by $\rho_{\mathrm{RTGDA}} = \max_\ell |1 + \eta\tilde{\lambda}_\ell|$ for some $\eta > 0$ such that $|1 + \eta\tilde{\lambda}_\ell| < 1$ for all $\ell$. Let $\lambda_i$ and $\mu_j$ be the eigenvalues of $(\mathrm{Hess}_x f - \mathrm{grad}^2_{xy} f \circ \mathrm{Hess}^{-1}_y f \circ \mathrm{grad}^2_{yx} f)(z^*)$ and $-\mathrm{Hess}_y f(z^*)$ respectively. Hence for strictly local minimax point $z^*$ we have $\lambda_i, \mu_j > 0$ and we can see $\tilde{\lambda}_\ell \in \{-\tau\lambda_i\}_{i=1}^{d_x} \cup \{-\mu_j\}_{j=1}^{d_y}$. Thus the convergence rate $\rho_{\mathrm{RTGDA}} = \max_i |1 - \eta\tau\lambda_i| \vee \max_j |1 - \eta\mu_j|$. The requirement on $\eta$ such that $\rho_{\mathrm{RTGDA}} < 1$ is again $\eta < \frac{2}{\tau\lambda_1 \vee \mu_1}$. This proves the convergence rate for RTGDA. For RFR, the proof is exactly the same. $\qquad\square$

*Proof of Theorem 3.* The derivative of $G_{\mathrm{RNFR}}[\mathrm{grad} f]$ at $z^*$ is given by

$$\boldsymbol{\nabla}(G_{\mathrm{RNFR}}[\mathrm{grad} f])(z^*) = G_{\mathrm{RNFR}}(z^*)\mathrm{Hess} f(z^*)$$
$$= \begin{pmatrix} -\tau\mathrm{id} & 0 \\ \tau(\mathrm{Hess}^{-1}_y f \circ \mathrm{grad}^2_{yx} f)(z^*) & -\zeta\mathrm{Hess}^{-1}_y f(z^*) \end{pmatrix} \begin{pmatrix} \mathrm{Hess}_x f(z^*) & \mathrm{grad}^2_{xy} f(z^*) \\ \mathrm{grad}^2_{yx} f(z^*) & \mathrm{Hess}_y f(z^*) \end{pmatrix}$$

which is similar to

$$\begin{pmatrix} -\mathrm{id} & 0 \\ (\mathrm{Hess}^{-1}_y f \circ \mathrm{grad}^2_{yx} f)(z^*) & \mathrm{id} \end{pmatrix} G_{\mathrm{RNFR}}(z^*)\mathrm{Hess} f(z^*) \begin{pmatrix} -\mathrm{id} & 0 \\ (\mathrm{Hess}^{-1}_y f \circ \mathrm{grad}^2_{yx} f)(z^*) & \mathrm{id} \end{pmatrix}$$
$$= \begin{pmatrix} -\tau\big(\mathrm{Hess}_x f - \mathrm{grad}^2_{xy} \circ \mathrm{Hess}^{-1}_y f \circ \mathrm{grad}^2_{yx}\big)(z^*) & \tau\mathrm{grad}^2_{xy} f(z^*) \\ 0 & -\zeta\mathrm{id} \end{pmatrix}.$$

Thus, the eigenvalues of $\boldsymbol{\nabla}(G_{\mathrm{RNFR}}[\mathrm{grad} f])(z^*)$ are the union of eigenvalues of $-\tau\big(\mathrm{Hess}_x f - \mathrm{grad}^2_{xy} \circ \mathrm{Hess}^{-1}_y f \circ \mathrm{grad}^2_{yx}\big)(z^*)$ and $-\zeta\mathrm{id}$. If $z^*$ is a strict local minimax, then we have all eigenvalues of $\boldsymbol{\nabla}(G_{\mathrm{RNFR}}[\mathrm{grad} f])(z^*)$ negative, which suggests $z^*$ is a strictly stable point. On the other hand, if $z^*$ is a strictly stable fixed point, then $-\tau\big(\mathrm{Hess}_x f - \mathrm{grad}^2_{xy} \circ \mathrm{Hess}^{-1}_y f \circ \mathrm{grad}^2_{yx}\big)(z^*) \prec 0$ and hence $\boldsymbol{\nabla}(G_{\mathrm{RNFR}}[\mathrm{grad} f])(z^*)$ has all eigenvalues negative. Similar as before, the asymptotic convergence rate is given by $\max_i |1 - \eta\tau\lambda_i| \vee \max_j |1 - \eta\zeta|$ where $\lambda_i$ are the eigenvalues of $\big(\mathrm{Hess}_x f - \mathrm{grad}^2_{xy} \circ \mathrm{Hess}^{-1}_y f \circ \mathrm{grad}^2_{yx}\big)(z^*)$. Suppose we choose $\zeta = 1/\eta$, then the convergence rate simplifies to $\max_i |1 - \eta\tau\lambda_i|$ where $\eta < \frac{2}{\tau\lambda_1}$ to ensure the rate is less than one.

For the RNTGDA algorithm, we derive

$$\boldsymbol{\nabla}(G_{\mathrm{RNTGDA}}[\mathrm{grad} f])(z^*) = G_{\mathrm{RNTGDA}}(z^*)\mathrm{Hess} f(z^*)$$
$$= \begin{pmatrix} -\tau\mathrm{id} & \tau(\mathrm{grad}^2_{xy} f \circ \mathrm{Hess}^{-1}_y f)(z^*) \\ 0 & -\zeta\mathrm{Hess}^{-1}_y f(z^*) \end{pmatrix} \begin{pmatrix} \mathrm{Hess}_x f(z^*) & \mathrm{grad}^2_{xy} f(z^*) \\ \mathrm{grad}^2_{yx} f(z^*) & \mathrm{Hess}_y f(z^*) \end{pmatrix}$$
$$= \begin{pmatrix} -\tau\big(\mathrm{Hess}_x f - \mathrm{grad}^2_{xy} f \circ \mathrm{Hess}^{-1}_y f \circ \mathrm{grad}^2_{yx} f\big)(z^*) & 0 \\ -\zeta\mathrm{Hess}^{-1}_y f(z^*)\mathrm{grad}^2_{yx} f(z^*) & -\zeta\mathrm{id} \end{pmatrix}$$

which has the same eigenvalues as $\boldsymbol{\nabla}(G_{\mathrm{RNFR}}[\mathrm{grad} f])(z^*)$. Thus the same results follow. $\qquad\square$

*Proof of Proposition 13.* We compute the derivative of $H$ at stationary points $z^*$ as

$$\boldsymbol{\nabla} H(z^*) = \big(\beta\mathrm{id} - (\boldsymbol{\nabla} G(z^*))^\dagger\big) \circ \boldsymbol{\nabla} G(z^*).$$

First, we analyze the asymptotic behavior of RHM (i.e., when $\beta = 0$). We see RHM converges to a stationary point as long as $\boldsymbol{\nabla} G(z^*)$ is invertible, and the asymptotic convergence rate is indeed linear that depends on the spectrum of $\mathrm{Hess} f(z^*)$. More superficially for RHM, $\mathrm{id} + \eta\boldsymbol{\nabla} H(z^*) = \mathrm{id} - \eta(\boldsymbol{\nabla} G(z^*))^\dagger \circ \boldsymbol{\nabla} G(z^*) = \mathrm{id} - \eta(\mathrm{Hess} f(z^*))^2$. As long as $\mathrm{Hess} f(z^*)$ is invertible, then we see $z^*$ is a strictly stable fixed point of RHM. Let $\delta_i$ be the eigenvalues of $\mathrm{Hess} f(z^*)$. Then the asymptotic convergence rate is given by $\rho_{\mathrm{RHM}} = \max_i |1 - \eta\delta_i^2|$ and $\eta < \frac{2}{\max_i \delta_i^2}$. However, because the only condition on $z^*$ being a strictly stable fixed point is that $\mathrm{Hess} f(z^*)$ is invertible. It may happen that $z^*$ is a local minimum, local maximum, local minimax or local maximin.

For RCON with $\beta > 0$, we have $\mathrm{id} + \eta\boldsymbol{\nabla} H(z^*) = \mathrm{id} + \eta\beta\boldsymbol{\nabla} G(z^*) - \eta(\boldsymbol{\nabla} G(z^*))^\dagger \circ \boldsymbol{\nabla} G(z^*)$. To analyze the asymptotic behavior, we first consider the real part of eigenvalues of $\boldsymbol{\nabla} H(z^*)$, i.e., $\mathfrak{R}(\lambda(\boldsymbol{\nabla} H(z^*)))$. From

classic results (Garren, 1968) on real parts of eigenvalues, we have $\lambda_{\min}(\{A\}_S) \le \Re(\lambda(\vec{A})) \le \lambda_{\max}(\{A\}_S)$ where we denote $\{A\}_S := (A + A^\dagger)/2$ for any linear operator $A$. Then we see

$$\{\boldsymbol{\nabla}H(z^*)\}_S = -(\mathrm{Hess}f(z^*))^2 + \beta \begin{pmatrix} -\mathrm{Hess}_x f(z^*) & 0 \\ 0 & \mathrm{Hess}_y f(z^*) \end{pmatrix}.$$

Further we have from (Merikoski & Kumar, 2004, Theorem 1)

$$\lambda_{\max}(\{\boldsymbol{\nabla}H(z^*)\}_S) \le -\min_i \delta_i^2 - \beta(\min_j \upsilon_j \wedge \min_\ell \mu_\ell)$$
$$\lambda_{\min}(\{\boldsymbol{\nabla}H(z^*)\}_S) \ge -\max_i \delta_i^2 - \beta(\max_j \upsilon_j \vee \max_\ell \mu_\ell)$$

where we let $\upsilon_j, \mu_\ell$ be the eigenvalues of $\mathrm{Hess}_x f(z^*)$ and $-\mathrm{Hess}_y f(z^*)$ respectively. A sufficient condition for $z^*$ to be strictly stable fixed point of RCON is thus $\min_i \delta_i^2 + \beta(\min_j \upsilon_j \wedge \min_\ell \mu_\ell) > 0$. Similar to the case of RHM, there are many such points as long as $\delta_i^2$ is sufficiently larger than the negative spectrum of $\mathrm{Hess}_x f(z^*), -\mathrm{Hess}_y f(z^*)$. On the contrary, a sufficient condition for $z^*$ to be unstable is where $\max_i \delta_i^2 + \beta(\max_j \upsilon_j \vee \max_\ell \mu_\ell) < 0$. $\qquad\square$

## J  Proofs of Section 7

**Lemma 4.** *The Riemannian gradient and second-order derivatives for the logdet quadratic problem*

$$f(\mathbf{X}, \mathbf{Y}) = c_1(\log\det(\mathbf{X}))^2 + c_2 \log\det(\mathbf{X})\log\det(\mathbf{Y}) + c_3(\log\det(\mathbf{Y}))^2$$

*are derived as follows, where we let $\mathbf{Z} = (\mathbf{X}, \mathbf{Y})$ for simplicity.*

$$\mathrm{grad}_x f(\mathbf{Z}) = (2c_1 \log\det(\mathbf{X}) + c_2 \log\det(\mathbf{Y}))\mathbf{X}, \quad \mathrm{grad}_y f(\mathbf{Z}) = (2c_3 \log\det(\mathbf{Y}) + c_2 \log\det(\mathbf{X}))\mathbf{Y}$$
$$\mathrm{Hess}_x f(\mathbf{Z})[\mathbf{U}] = 2c_1 \mathrm{tr}(\mathbf{X}^{-1}\mathbf{U})\mathbf{X}, \quad \mathrm{Hess}_y f(\mathbf{Z})[\mathbf{V}] = 2c_3 \mathrm{tr}(\mathbf{Y}^{-1}\mathbf{V})\mathbf{Y}$$
$$\mathrm{grad}_{xy}^2 f(\mathbf{Z})[\mathbf{V}] = c_2 \mathrm{tr}(\mathbf{Y}^{-1}\mathbf{V})\mathbf{X}, \quad \mathrm{grad}_{yx}^2 f(\mathbf{Z})[\mathbf{U}] = c_2 \mathrm{tr}(\mathbf{X}^{-1}\mathbf{U})\mathbf{Y}$$

*Proof of Lemma 4.* The Riemannian gradient is given by

$$\mathrm{grad}_x f(\mathbf{Z}) = \mathbf{X}\nabla f(\mathbf{Z})\mathbf{X} = \mathbf{X}(2c_1 \log\det(\mathbf{X})\mathbf{X}^{-1} + c_2 \log\det(\mathbf{Y})\mathbf{X}^{-1})\mathbf{X}$$
$$= (2c_1 \log\det(\mathbf{X}) + c_2 \log\det(\mathbf{Y}))\mathbf{X}$$

and similarly for $\mathbf{Y}$. For Riemannian Hessian $\mathrm{Hess}_x f(\mathbf{Z})[\mathbf{U}]$ for any $\mathbf{U} \in T_{\mathbf{X}}\mathbb{S}_{++}^d$, we use the formula $\mathrm{Hess}_x f(\mathbf{Z})[\mathbf{U}] = \mathrm{D}_x \mathrm{grad}_x f(\mathbf{Z})[\mathbf{U}] - \{\mathbf{U}\mathbf{X}^{-1}\mathrm{grad}_x f(\mathbf{Z})\}_S$, which gives

$$\mathrm{Hess}_x f(\mathbf{Z})[\mathbf{U}] = 2c_1 \mathrm{tr}(\mathbf{X}^{-1}\mathbf{U})\mathbf{X} + 2c_1 \log\det(\mathbf{X})\mathbf{U} + c_2 \log\det(\mathbf{Y})\mathbf{U} - \{\mathbf{U}\mathbf{X}^{-1}\mathrm{grad}_x f(\mathbf{Z})\}_S$$
$$= 2c_1 \mathrm{tr}(\mathbf{X}^{-1}\mathbf{U})\mathbf{X}.$$

The cross derivatives are given from the definition as $\mathrm{grad}_{xy}^2 f(\mathbf{Z})[\mathbf{V}] = \mathrm{D}_y \mathrm{grad}_x f(\mathbf{Z})[\mathbf{V}] = c_2 \mathrm{tr}(\mathbf{Y}^{-1}\mathbf{V})\mathbf{X}$. Similar arguments hold for $\mathrm{Hess}_y f(\mathbf{Z})[\mathbf{V}]$ and $\mathrm{grad}_{yx}^2 f(\mathbf{Z})[\mathbf{U}]$. $\qquad\square$

**Lemma 5.** *The Riemannian gradient and second-order derivatives for problem* (4), *i.e.,*

$$f(\mathbf{X}, \mathbf{Y}) = c_1(\log\det(\mathbf{X}))^2 + c_2 \log\det(\mathbf{X})\log\det(\mathbf{Y}) + c_3 \mathrm{dist}^2(\mathbf{I}, \mathbf{Y})$$

*are derived as follows.*

$$\mathrm{grad}_x f(\mathbf{Z}) = (2c_1 \log\det(\mathbf{X}) + c_2 \log\det(\mathbf{Y}))\mathbf{X},$$
$$\mathrm{grad}_y f(\mathbf{Z}) = c_2 \log\det(\mathbf{X})\mathbf{Y} - 2c_3 \mathrm{Log}_{\mathbf{Y}}(\mathbf{I}) = c_2 \log\det(\mathbf{X})\mathbf{Y} - 2c_3\{\mathbf{Y}\mathrm{logm}(\mathbf{Y}^{-1})\}_S$$
$$\mathrm{Hess}_x f(\mathbf{Z})[\mathbf{U}] = 2c_1 \mathrm{tr}(\mathbf{X}^{-1}\mathbf{U})\mathbf{X},$$
$$\mathrm{Hess}_y f(\mathbf{Z})[\mathbf{V}] = -2c_3\{\mathbf{V}\mathrm{logm}(\mathbf{Y}^{-1}) - \mathrm{Dlogm}(\mathbf{Y}^{-1})[\mathbf{Y}^{-1}\mathbf{V}\mathbf{Y}^{-1}]\}_S + 2c_3\{\mathbf{V}\mathbf{Y}^{-1}\{\mathbf{Y}\mathrm{logm}(\mathbf{Y}^{-1})\}_S\}_S$$
$$\mathrm{grad}_{xy}^2 f(\mathbf{Z})[\mathbf{V}] = c_2 \mathrm{tr}(\mathbf{Y}^{-1}\mathbf{V})\mathbf{X}, \quad \mathrm{grad}_{yx}^2 f(\mathbf{Z})[\mathbf{U}] = c_2 \mathrm{tr}(\mathbf{X}^{-1}\mathbf{U})\mathbf{Y}.$$

*Proof of Lemma 5.* Given we only change the quadratic term on $\mathbf{Y}$, the only differences come in the computation of $\mathrm{grad}_y f(\mathbf{Z})$ and $\mathrm{Hess}_y f(\mathbf{Z})$. It has been shown by Alimisis et al. (2020) that the Riemannian gradient of the squared Riemannian distance is $\mathrm{grad}_{\mathbf{A}} \mathrm{dist}^2(\mathbf{A}, \mathbf{B}) = -2\mathrm{Log}_{\mathbf{A}}(\mathbf{B})$. Substituting $\mathbf{B} = \mathbf{I}$, we obtain the expression for $\mathrm{grad}_y f(\mathbf{Z})$. In terms of the Riemannian Hessian, i.e., $\mathrm{Hess}_y f(\mathbf{Z})[\mathbf{V}]$ for any $\mathbf{V} \in T_{\mathbf{Y}} \mathbb{S}_{++}^d$, we use the same formula as in Lemma 4, which leads to the desired expression. $\qquad\square$

**Lemma 6.** *The Riemannian gradient and Riemannian second-order derivatives of the robust Fréchet mean (RFM) problem* (5)

$$f(\mathbf{M}, \mathbf{x}) = \mathbf{x}^\top \mathbf{M} \mathbf{x} + \frac{\alpha}{n} \sum_{i=1}^n \mathrm{dist}^2(\mathbf{M}, \mathbf{M}_i)$$

*are computed as*

$$\mathrm{grad}_{\mathbf{M}} f(\mathbf{M}, \mathbf{x}) = \mathbf{M} \mathbf{x} \mathbf{x}^\top \mathbf{M} + \frac{2\alpha}{n} \sum_{i=1}^n \{\mathrm{logm}(\mathbf{M}\mathbf{M}_i^{-1})\mathbf{M}\}_{\mathrm{S}}, \qquad \mathrm{grad}_{\mathbf{x}} f(\mathbf{M}, \mathbf{x}) = 2(\mathbf{I} - \mathbf{x}\mathbf{x}^\top)\mathbf{M}\mathbf{x}$$

$$\mathrm{Hess}_{\mathbf{M}} f(\mathbf{M}, \mathbf{x})[\mathbf{U}] = \{\mathbf{U}\mathbf{x}\mathbf{x}^\top\mathbf{M}\}_{\mathrm{S}} + \frac{2\alpha}{n} \sum_{i=1}^n \Big\{ \mathrm{Dlogm}(\mathbf{M}\mathbf{M}_i^{-1})[\mathbf{U}\mathbf{M}_i^{-1}]\mathbf{M} + \mathrm{logm}(\mathbf{M}\mathbf{M}_i^{-1})\mathbf{U}$$

$$- \mathbf{U}\mathbf{M}^{-1}\{\mathrm{logm}(\mathbf{M}\mathbf{M}_i^{-1})\mathbf{M}\}_{\mathrm{S}} \Big\}_{\mathrm{S}}$$

$$\mathrm{Hess}_{\mathbf{x}} f(\mathbf{M}, \mathbf{x})[\mathbf{v}] = 2(\mathbf{I} - \mathbf{x}\mathbf{x}^\top)\mathbf{M}\mathbf{v} - 2(\mathbf{x}^\top\mathbf{M}\mathbf{x})\mathbf{v}$$

$$\mathrm{grad}_{\mathbf{M},\mathbf{x}} f(\mathbf{M}, \mathbf{x})[\mathbf{v}] = \mathbf{M}\mathbf{v}\mathbf{x}^\top\mathbf{M} + \mathbf{M}\mathbf{x}\mathbf{v}^\top\mathbf{M} \qquad \mathrm{grad}_{\mathbf{x},\mathbf{M}} f(\mathbf{M}, \mathbf{x})[\mathbf{U}] = 2(\mathbf{I} - \mathbf{x}\mathbf{x}^\top)\mathbf{U}\mathbf{x}$$

*Proof of Lemma 6.* First, we recall that the Riemannian gradient and Hessian on a sphere are given by $\mathrm{grad}g(\mathbf{x}) = (\mathbf{I} - \mathbf{x}\mathbf{x}^\top)\nabla g(\mathbf{x})$ and for any $\mathbf{v} \in T_{\mathbf{x}}\mathcal{S}^d$, $\mathrm{Hess}g(\mathbf{x})[\mathbf{v}] = (\mathbf{I} - \mathbf{x}\mathbf{x}^\top)\nabla^2 g(\mathbf{x})[\mathbf{v}] - \mathbf{x}^\top\nabla f(\mathbf{x})\mathbf{v}$. Substituting the results $\nabla_{\mathbf{x}} f(\mathbf{M}, \mathbf{x}) = 2\mathbf{M}\mathbf{x}$ and $\nabla^2 f(\mathbf{M}, \mathbf{x})[\mathbf{v}] = 2\mathbf{M}\mathbf{v}$ in the Riemannian gradient and Hessian, we obtain the desired expressions. Then for $\mathrm{grad}_{\mathbf{x},\mathbf{M}} f(\mathbf{M}, \mathbf{x})[\mathbf{U}] = \mathrm{D}_{\mathbf{M}} \mathrm{grad}_x f(\mathbf{M}, \mathbf{x})[\mathbf{U}] = 2(\mathbf{I} - \mathbf{x}\mathbf{x}^\top)\mathbf{U}\mathbf{x}$.

The Riemannian gradient for $\mathbf{M}$ can be derived as

$$\mathrm{grad}_{\mathbf{M}} f(\mathbf{M}, \mathbf{x}) = \mathbf{M}\mathbf{x}\mathbf{x}^\top\mathbf{M} - \frac{2\alpha}{n}\mathrm{Log}_{\mathbf{M}}(\mathbf{M}_i) = \mathbf{M}\mathbf{x}\mathbf{x}^\top\mathbf{M} - \frac{2\alpha}{n}\sum_{i=1}^n\{\mathbf{M}\mathrm{logm}(\mathbf{M}^{-1}\mathbf{M}_i)\}_{\mathrm{S}}.$$

We highlight that the second term in the expression of $\mathrm{grad}_{\mathbf{M}} f(\mathbf{M}, \mathbf{x})$ can be costly to evaluate because $\mathbf{M}^{-1}$ needs to be computed every iteration. To simplify the computation, we can show $\{\mathbf{M}\mathrm{logm}(\mathbf{M}^{-1}\mathbf{M}_i)\}_{\mathrm{S}} = -\{\mathrm{logm}(\mathbf{M}\mathbf{M}_i^{-1})\mathbf{M}\}_{\mathrm{S}}$ where $\mathbf{M}_i^{-1}$ can be pre-computed.

To see why this is the case, we consider the generalized eigenvalue problem $\mathbf{M}_i\boldsymbol{\phi} = \lambda\mathbf{M}\boldsymbol{\phi}$. Since $\mathbf{M}, \mathbf{M}_i \in \mathbb{S}_{++}^d$, $\lambda > 0$ and we have $\mathbf{M}^{-1}\mathbf{M}_i\boldsymbol{\phi} = \lambda\boldsymbol{\phi}$ and $\mathbf{M}_i^{-1}\mathbf{M}\boldsymbol{\phi} = \lambda^{-1}\boldsymbol{\phi}$. Written compactly, let $\boldsymbol{\Phi}$ be the matrix collecting $\boldsymbol{\phi}$ and thus we have $\mathbf{M}^{-1}\mathbf{M}_i = \boldsymbol{\Phi}\boldsymbol{\Lambda}\boldsymbol{\Phi}^{-1}$ and $\mathbf{M}_i^{-1}\mathbf{M} = \boldsymbol{\Phi}\boldsymbol{\Lambda}^{-1}\boldsymbol{\Phi}^{-1}$ where $\boldsymbol{\Lambda} > 0$ is the diagonal matrix collecting the generalized eigenvalues. Then

$$\mathbf{M}\mathrm{logm}(\mathbf{M}^{-1}\mathbf{M}_i) = \mathbf{M}\boldsymbol{\Phi}\log(\boldsymbol{\Lambda})\boldsymbol{\Phi}^{-1} = -\mathbf{M}\boldsymbol{\Phi}\log(\boldsymbol{\Lambda}^{-1})\boldsymbol{\Phi}^{-1} = -(\boldsymbol{\Phi}^{-\top}\log(\boldsymbol{\Lambda}^{-1})\boldsymbol{\Phi}^\top\mathbf{M})^\top$$

$$= -(\mathrm{logm}(\mathbf{M}\mathbf{M}_i^{-1})\mathbf{M})^\top.$$

Hence, we can see $\{\mathbf{M}\mathrm{logm}(\mathbf{M}^{-1}\mathbf{M}_i)\}_{\mathrm{S}} = -\{\mathrm{logm}(\mathbf{M}\mathbf{M}_i^{-1})\mathbf{M}\}_{\mathrm{S}}$, and therefore,

$$\mathrm{grad}_{\mathbf{M}} f(\mathbf{M}, \mathbf{x}) = \mathbf{M}\mathbf{x}\mathbf{x}^\top\mathbf{M} + \frac{2\alpha}{n}\sum_{i=1}^n\{\mathrm{logm}(\mathbf{M}\mathbf{M}_i^{-1})\mathbf{M}\}_{\mathrm{S}}.$$

We derive $\operatorname{grad}_{\mathbf{M},\mathbf{x}}f(\mathbf{M},\mathbf{x})[\mathbf{v}] = \mathrm{D}_{\mathbf{x}}\operatorname{grad}_{\mathbf{M}}f(\mathbf{M},\mathbf{x})[\mathbf{v}] = \mathbf{M}\mathbf{v}\mathbf{x}^\top\mathbf{M} + \mathbf{M}\mathbf{x}\mathbf{v}^\top\mathbf{M}$. It only remains to derive $\operatorname{Hess}_{\mathbf{M}}f(\mathbf{M},\mathbf{x})[\mathbf{U}]$, which is

$$\operatorname{Hess}_{\mathbf{M}}f(\mathbf{M},\mathbf{x})[\mathbf{U}]$$
$$= \mathrm{D}_{\mathbf{M}}\operatorname{grad}_{\mathbf{M}}f(\mathbf{M},\mathbf{x})[\mathbf{U}] - \{\mathbf{U}\mathbf{M}^{-1}\operatorname{grad}_{\mathbf{M}}f(\mathbf{M},\mathbf{x})\}_{\mathrm{S}}$$
$$= 2\{\mathbf{U}\mathbf{x}\mathbf{x}^\top\mathbf{M}\}_{\mathrm{S}} - \{\mathbf{U}\mathbf{x}\mathbf{x}^\top\mathbf{M}\}_{\mathrm{S}} + \frac{2\alpha}{n}\sum_{i=1}^{n}\Big(\{\mathrm{Dlogm}(\mathbf{M}\mathbf{M}_i^{-1})[\mathbf{U}\mathbf{M}_i^{-1}]\mathbf{M} + \operatorname{logm}(\mathbf{M}\mathbf{M}_i^{-1})\mathbf{U}\}_{\mathrm{S}}$$
$$- \{\mathbf{U}\mathbf{M}^{-1}\{\operatorname{logm}(\mathbf{M}\mathbf{M}_i^{-1})\mathbf{M}\}_{\mathrm{S}}\}_{\mathrm{S}}\Big)$$
$$= \{\mathbf{U}\mathbf{x}\mathbf{x}^\top\mathbf{M}\}_{\mathrm{S}} + \frac{2\alpha}{n}\sum_{i=1}^{n}\Big\{\mathrm{Dlogm}(\mathbf{M}\mathbf{M}_i^{-1})[\mathbf{U}\mathbf{M}_i^{-1}]\mathbf{M} + \operatorname{logm}(\mathbf{M}\mathbf{M}_i^{-1})\mathbf{U}$$
$$- \mathbf{U}\mathbf{M}^{-1}\{\operatorname{logm}(\mathbf{M}\mathbf{M}_i^{-1})\mathbf{M}\}_{\mathrm{S}}\Big\}_{\mathrm{S}}.$$

$\square$

**Lemma 7.** *The Riemannian gradient and second-order derivatives expressions for the robust maximum likelihood estimation problem* (7)

$$f(\boldsymbol{\delta},\mathbf{S}) := -\frac{n}{2}\log\det(\mathbf{S}) - \frac{1}{2}\sum_{i=1}^{n}(\mathbf{y}_i - \boldsymbol{\delta})^\top\mathbf{S}^{-1}(\mathbf{y}_i - \boldsymbol{\delta})$$

*are derived as*

$$\operatorname{grad}_{\boldsymbol{\delta}}f(\boldsymbol{\delta},\mathbf{S}) = (\mathbf{I} - \boldsymbol{\delta}\boldsymbol{\delta}^\top)\mathbf{S}^{-1}\Big(\sum_{i=1}^{n}\mathbf{y}_i - n\boldsymbol{\delta}\Big), \quad \operatorname{grad}_{\mathbf{S}}f(\boldsymbol{\delta},\mathbf{S}) = -\frac{n}{2}\mathbf{S} + \frac{1}{2}\sum_{i=1}^{n}(\mathbf{y}_i - \boldsymbol{\delta})(\mathbf{y}_i - \boldsymbol{\delta})^\top$$

$$\operatorname{grad}^2_{\boldsymbol{\delta},\mathbf{S}}f(\boldsymbol{\delta},\mathbf{S})[\widetilde{\mathbf{S}}] = -(\mathbf{I} - \boldsymbol{\delta}\boldsymbol{\delta}^\top)(\mathbf{S}^{-1}\widetilde{\mathbf{S}}\mathbf{S}^{-1})\Big(\sum_{i=1}^{n}\mathbf{y}_i - n\boldsymbol{\delta}\Big), \quad \operatorname{grad}^2_{\mathbf{S},\boldsymbol{\delta}}f(\boldsymbol{\delta},\mathbf{S})[\widetilde{\boldsymbol{\delta}}] = -\sum_{i=1}^{n}\{\widetilde{\boldsymbol{\delta}}(\mathbf{y}_i - \boldsymbol{\delta})^\top\}_{\mathrm{S}},$$

$$\operatorname{Hess}_{\mathbf{S}}f(\boldsymbol{\delta},\mathbf{S})[\widetilde{\mathbf{S}}] = -\frac{1}{2}\sum_{i=1}^{n}\{\widetilde{\mathbf{S}}\mathbf{S}^{-1}(\mathbf{y}_i - \boldsymbol{\delta})(\mathbf{y}_i - \boldsymbol{\delta})^\top\}_{\mathrm{S}},$$

$$\operatorname{Hess}_{\boldsymbol{\delta}}f(\boldsymbol{\delta},\mathbf{S})[\widetilde{\boldsymbol{\delta}}] = -n(\mathbf{I} - \boldsymbol{\delta}\boldsymbol{\delta}^\top)\mathbf{S}^{-1}\widetilde{\boldsymbol{\delta}} - \boldsymbol{\delta}^\top\Big(\mathbf{S}^{-1}\Big(\sum_{i=1}^{n}\mathbf{y}_i - n\boldsymbol{\delta}\Big)\Big)\widetilde{\boldsymbol{\delta}}.$$

*Proof of Lemma 7.* From the definition of the Riemannian gradient for $\mathcal{S}^d, \mathbb{S}^d_{++}$, the Riemannian gradient can be easily derived, and similarly for the cross derivatives. For the Riemannian Hessian,

$$\operatorname{Hess}_{\mathbf{S}}f(\boldsymbol{\delta},\mathbf{S})[\widetilde{\mathbf{S}}] = \mathrm{D}_{\mathbf{S}}\operatorname{grad}_{\mathbf{S}}f(\boldsymbol{\delta},\mathbf{S})[\widetilde{\mathbf{S}}] - \{\widetilde{\mathbf{S}}\mathbf{S}^{-1}\operatorname{grad}_{\mathbf{S}}f(\boldsymbol{\delta},\mathbf{S})\}_{\mathrm{S}}$$
$$= -\frac{n}{2}\widetilde{\mathbf{S}} - \{-\frac{n}{2}\widetilde{\mathbf{S}} + \frac{1}{2}\sum_{i=1}^{n}\widetilde{\mathbf{S}}\mathbf{S}^{-1}(\mathbf{y}_i - \boldsymbol{\delta})(\mathbf{y}_i - \boldsymbol{\delta})^\top\}_{\mathrm{S}} = -\frac{1}{2}\sum_{i=1}^{n}\{\widetilde{\mathbf{S}}\mathbf{S}^{-1}(\mathbf{y}_i - \boldsymbol{\delta})(\mathbf{y}_i - \boldsymbol{\delta})^\top\}_{\mathrm{S}},$$

$$\operatorname{Hess}_{\boldsymbol{\delta}}f(\boldsymbol{\delta},\mathbf{S})[\widetilde{\boldsymbol{\delta}}] = (\mathbf{I} - \boldsymbol{\delta}\boldsymbol{\delta}^\top)\nabla^2_{\boldsymbol{\delta}}f(\boldsymbol{\delta},\mathbf{S})[\widetilde{\boldsymbol{\delta}}] - \boldsymbol{\delta}^\top\nabla_{\boldsymbol{\delta}}f(\boldsymbol{\delta},\mathbf{S})\widetilde{\boldsymbol{\delta}} = -n(\mathbf{I} - \boldsymbol{\delta}\boldsymbol{\delta}^\top)\mathbf{S}^{-1}\widetilde{\boldsymbol{\delta}} - \boldsymbol{\delta}^\top\Big(\mathbf{S}^{-1}\Big(\sum_{i=1}^{n}\mathbf{y}_i - n\boldsymbol{\delta}\Big)\Big)\widetilde{\boldsymbol{\delta}}.$$

The proof is complete. $\square$

**Lemma 8.** *The Riemannian gradient and Riemannian second-order derivatives of Projection robust Wasserstein distance, i.e.,*

$$f(\Gamma,\mathbf{U}) := \sum_{i=1}^{m}\sum_{j=1}^{n}\Big(\frac{1}{mn}\Gamma_{ij}\|\mathbf{U}^\top\mathbf{x}_i - \mathbf{U}^\top\mathbf{y}_j\|_2^2 + \epsilon\,\Gamma_{i,j}\big(\log(\Gamma_{ij}) - 1\big)\Big)$$

*are computed as follows. First, the Riemannian gradient and Riemannian Hessian can be computed as long as the Euclidean gradient and Hessian are computed, i.e.,*

$$\nabla_{\mathbf{U}} f(\Gamma, \mathbf{U}) = \frac{2}{mn} \big(\mathbf{X}^\top \mathrm{diag}(\Gamma \mathbf{1}_n) \mathbf{X} + \mathbf{Y}^\top \mathrm{diag}(\Gamma^\top \mathbf{1}_m) \mathbf{Y} - \mathbf{X}^\top \Gamma \mathbf{Y} - \mathbf{Y}^\top \Gamma^\top \mathbf{X}\big) \mathbf{U},$$

$$\nabla_{\Gamma} f(\Gamma, \mathbf{U}) = \frac{1}{mn} \big(\mathrm{diag}(\mathbf{X}\mathbf{U}\mathbf{U}^\top \mathbf{X}^\top) \mathbf{1}_n^\top + \mathbf{1}_m^\top \mathrm{diag}(\mathbf{Y}\mathbf{U}\mathbf{U}^\top \mathbf{Y}^\top)^\top - 2\mathbf{X}\mathbf{U}\mathbf{U}^\top \mathbf{Y}^\top\big) + \epsilon \log(\Gamma),$$

$$\nabla_{\mathbf{U}}^2 f(\Gamma, \mathbf{U})[\widetilde{\mathbf{U}}] = \frac{2}{mn} \big(\mathbf{X}^\top \mathrm{diag}(\Gamma \mathbf{1}_n) \mathbf{X} + \mathbf{Y}^\top \mathrm{diag}(\Gamma^\top \mathbf{1}_m) \mathbf{Y} - \mathbf{X}^\top \Gamma \mathbf{Y} - \mathbf{Y}^\top \Gamma^\top \mathbf{X}\big) \widetilde{\mathbf{U}}$$

$$\nabla_{\Gamma}^2 f(\Gamma, \mathbf{U})[\widetilde{\Gamma}] = \epsilon \widetilde{\Gamma} \oslash \Gamma,$$

*where we denote $\oslash$ as elementwise division. In addition, the cross derivatives are given by*

$$\mathrm{grad}_{\mathbf{U},\Gamma}^2 f(\Gamma, \mathbf{U})[\widetilde{\Gamma}] = \frac{2}{mn}(\mathbf{I} - \mathbf{U}\mathbf{U}^\top)\Big(\mathbf{X}^\top \mathrm{diag}(\widetilde{\Gamma} \mathbf{1}_n) \mathbf{X} + \mathbf{Y}^\top \mathrm{diag}(\widetilde{\Gamma}^\top \mathbf{1}_m) \mathbf{Y} - \mathbf{X}^\top \widetilde{\Gamma} \mathbf{Y} - \mathbf{Y}^\top \widetilde{\Gamma}^\top \mathbf{X}\Big) \mathbf{U}$$

$$\mathrm{grad}_{\Gamma,\mathbf{U}}^2 f(\Gamma, \mathbf{U})[\widetilde{\mathbf{U}}] = \mathrm{P}_\Gamma(\Gamma \odot \mathrm{D}_{\mathbf{U}} \nabla_\Gamma f[\widetilde{\mathbf{U}}]),$$

*where $\mathrm{P}_\Gamma(\mathbf{V})$ is the orthogonal projection of $\mathbf{V} \in \mathbb{R}^{m \times n}$ to the tangent space $T_\Gamma \Pi(\mu, \nu)$.*

*Proof of Lemma 8.* To derive the gradient and second-order derivatives, we first rewrite the objective as follows. Let $\mathbf{X} \in \mathbb{R}^{m \times d}, \mathbf{Y} \in \mathbb{R}^{n \times d}$ be the data matrix. Then the objective is

$$f(\Gamma, \mathbf{U}) = \frac{1}{mn} \Big\langle \Gamma, \mathrm{diag}(\mathbf{X}\mathbf{U}\mathbf{U}^\top \mathbf{X}^\top) \mathbf{1}_n^\top + \mathbf{1}_m^\top \mathrm{diag}(\mathbf{Y}\mathbf{U}\mathbf{U}^\top \mathbf{Y}^\top) - 2\mathbf{X}\mathbf{U}\mathbf{U}^\top \mathbf{Y}^\top \Big\rangle_2 + \epsilon \big\langle \Gamma, \log(\Gamma) - \mathbf{1}_m \mathbf{1}_n^\top \big\rangle_2.$$

The Euclidean gradients are given by

$$\nabla_{\mathbf{U}} f(\Gamma, \mathbf{U}) = \frac{2}{mn} \big(\mathbf{X}^\top \mathrm{diag}(\Gamma \mathbf{1}_n) \mathbf{X} + \mathbf{Y}^\top \mathrm{diag}(\Gamma^\top \mathbf{1}_m) \mathbf{Y} - \mathbf{X}^\top \Gamma \mathbf{Y} - \mathbf{Y}^\top \Gamma^\top \mathbf{X}\big) \mathbf{U},$$

$$\nabla_{\Gamma} f(\Gamma, \mathbf{U}) = \frac{1}{mn} \big(\mathrm{diag}(\mathbf{X}\mathbf{U}\mathbf{U}^\top \mathbf{X}^\top) \mathbf{1}_n^\top + \mathbf{1}_m^\top \mathrm{diag}(\mathbf{Y}\mathbf{U}\mathbf{U}^\top \mathbf{Y}^\top)^\top - 2\mathbf{X}\mathbf{U}\mathbf{U}^\top \mathbf{Y}^\top\big) + \epsilon \log(\Gamma).$$

From the derivations of Riemannian gradients of the Stiefel manifold and doubly stochastic manifold, we have $\mathrm{grad}_{\mathbf{U}} f(\Gamma, \mathbf{U}) = (\mathbf{I} - \mathbf{U}\mathbf{U}^\top) \nabla_{\mathbf{U}} f(\mathbf{U}, \Gamma)$, and $\mathrm{grad}_{\Gamma} f(\Gamma, \mathbf{U}) = \mathrm{P}_\Gamma(\Gamma \odot \nabla_\Gamma f(\Gamma, \mathbf{U})) = \Gamma \odot \big(\nabla_\Gamma f(\Gamma, \mathbf{U}) - \alpha \mathbf{1}_n^\top - \mathbf{1}_m \beta^\top\big)$, where $\alpha \in \mathbb{R}^m, \beta \in \mathbb{R}^n$ are solutions from the following linear system, i.e.,

$$\begin{cases} \alpha \odot \mathbf{a} + \Gamma \beta = (\Gamma \odot \nabla_\Gamma f(\Gamma, \mathbf{U})) \mathbf{1}_m \\ \beta \odot \mathbf{b} + \Gamma^\top \alpha = (\Gamma \odot \nabla_\Gamma f(\Gamma, \mathbf{U}))^\top \mathbf{1}_n. \end{cases}$$

For the second-order derivatives, first we see that it is easy to derive the Riemannian Hessian $\mathrm{Hess}_{\mathbf{U}} f(\Gamma, \mathbf{U})[\widetilde{\mathbf{U}}]$ and $\mathrm{Hess}_\Gamma f(\Gamma, \mathbf{U})[\widetilde{\Gamma}]$ for any $\widetilde{\mathbf{U}} \in T_{\mathbf{U}} \mathrm{St}(d, r), \widetilde{\Gamma} \in T_\Gamma \Pi(\mu, \nu)$ as long as the Euclidean Hessian can be derived. To this end, we obtain

$$\nabla_{\mathbf{U}}^2 f(\Gamma, \mathbf{U})[\widetilde{\mathbf{U}}] = \frac{2}{mn} \big(\mathbf{X}^\top \mathrm{diag}(\Gamma \mathbf{1}_n) \mathbf{X} + \mathbf{Y}^\top \mathrm{diag}(\Gamma^\top \mathbf{1}_m) \mathbf{Y} - \mathbf{X}^\top \Gamma \mathbf{Y} - \mathbf{Y}^\top \Gamma^\top \mathbf{X}\big) \widetilde{\mathbf{U}}$$

$$\nabla_{\Gamma}^2 f(\Gamma, \mathbf{U})[\widetilde{\Gamma}] = \epsilon \widetilde{\Gamma} \oslash \Gamma.$$

For the cross-derivatives, we first compute

$$\mathrm{grad}_{\mathbf{U},\Gamma}^2 f(\Gamma, \mathbf{U})[\widetilde{\Gamma}] = \frac{2}{mn}(\mathbf{I} - \mathbf{U}\mathbf{U}^\top)\Big(\mathbf{X}^\top \mathrm{diag}(\widetilde{\Gamma} \mathbf{1}_n) \mathbf{X} + \mathbf{Y}^\top \mathrm{diag}(\widetilde{\Gamma}^\top \mathbf{1}_m) \mathbf{Y} - \mathbf{X}^\top \widetilde{\Gamma} \mathbf{Y} - \mathbf{Y}^\top \widetilde{\Gamma}^\top \mathbf{X}\Big) \mathbf{U}.$$

To derive $\mathrm{grad}_{\Gamma,\mathbf{U}}^2 f(\Gamma, \mathbf{U})[\widetilde{\mathbf{U}}]$, we let $f_1 := \alpha \odot \mathbf{a} + \Gamma \beta - (\Gamma \odot \nabla_\Gamma f) \mathbf{1}_n = 0$ and $f_2 := \beta \odot \mathbf{b} + \Gamma^\top \alpha - (\Gamma \odot \nabla_\Gamma f)^\top \mathbf{1}_m$, where we omit the evaluation of function $f$ at $(\mathbf{U}, \Gamma)$ for clarity. Then we have

$$\mathrm{D}_{\mathbf{U}} f_1[\widetilde{\mathbf{U}}] = \mathrm{D}_{\mathbf{U}} \alpha[\widetilde{\mathbf{U}}] \odot \mathbf{a} + \Gamma \mathrm{D}_{\mathbf{U}} \beta[\widetilde{\mathbf{U}}] - \big(\Gamma \odot \mathrm{D}_{\mathbf{U}} \nabla_\Gamma f[\widetilde{\mathbf{U}}]\big) \mathbf{1}_n = 0$$

$$\mathrm{D}_{\mathbf{U}} f_2[\widetilde{\mathbf{U}}] = \mathrm{D}_{\mathbf{U}} \beta[\widetilde{\mathbf{U}}] \odot \mathbf{b} + \Gamma^\top \mathrm{D}_{\mathbf{U}} \alpha[\widetilde{\mathbf{U}}] - \big(\Gamma \odot \mathrm{D}_{\mathbf{U}} \nabla_\Gamma f[\widetilde{\mathbf{U}}]\big)^\top \mathbf{1}_m = 0.$$

Thus we can compute $D_{\mathbf{U}}\boldsymbol{\alpha}[\widetilde{\mathbf{U}}]$ and $D_{\mathbf{U}}\boldsymbol{\beta}[\widetilde{\mathbf{U}}]$ from the above linear system. Then we have

$$\mathrm{grad}^2_{\mathbf{\Gamma},\mathbf{U}} f(\Gamma, \mathbf{U})[\widetilde{\mathbf{U}}] = \mathbf{\Gamma} \odot \big(D_{\mathbf{U}}\nabla_{\mathbf{\Gamma}} f[\widetilde{\mathbf{U}}] - D_{\mathbf{U}}\boldsymbol{\alpha}[\widetilde{\mathbf{U}}]\mathbf{1}_n^\top - \mathbf{1}_m (D_{\mathbf{U}}\boldsymbol{\beta}[\widetilde{\mathbf{U}}])^\top\big),$$

where $D_{\mathbf{U}}\nabla_{\mathbf{\Gamma}} f[\widetilde{\mathbf{U}}] = \frac{2}{mn}\big(\mathrm{diag}(\mathbf{X}\{\mathbf{U}\widetilde{\mathbf{U}}^\top\}_S \mathbf{X}^\top)\mathbf{1}_n^\top + \mathbf{1}_m^\top \mathrm{diag}(\mathbf{Y}\{\mathbf{U}\widetilde{\mathbf{U}}^\top\}_S \mathbf{Y}^\top)^\top - 2\mathbf{X}\{\mathbf{U}\widetilde{\mathbf{U}}^\top\}_S \mathbf{Y}^\top\big)$. This is equivalent to $\mathrm{grad}^2_{\mathbf{\Gamma},\mathbf{U}} f(\Gamma, \mathbf{U})[\widetilde{\mathbf{U}}] = P_{\mathbf{\Gamma}}(\mathbf{\Gamma} \odot D_{\mathbf{U}}\nabla_{\mathbf{\Gamma}} f[\widetilde{\mathbf{U}}])$. $\qquad\square$

