# OpenReview forum: "Nonconvex-nonconcave min-max optimization on Riemannian manifolds"
_TMLR — Accepted by TMLR_

### Review · Reviewer_yZgL · 2023-05-03

**Summary Of Contributions:**

This work extends the notions of local minimax points and stability of continuous and discrete dynamics for a nonconvex-nonconcave minimax optimization from Euclidean space to Riemannian manifolds. Equivalent conditions of stability are also given. Then this work obtains asymptotic convergence rates of the existing first-order TSRGDA algorithm and the proposed second-order algorithms.

**Audience:**

Yes

**Broader Impact Concerns:**

I did not find out any ethical concerns.

**Claims And Evidence:**

Yes

**Requested Changes:**

**Requested changes or answers (critical for acceptance):**

(1) The discriminator prediction in Figure 3 looks confusing to me. Could you explain the meaning? For example, does the discriminator classify between the real points from the Gaussian mixture and fake points from noise $z_i$? Does the pixel strength mean the probability of being classified as real points?

(2) In the RNGD algorithm, why do you use $-\eta$ not $+\eta$ to maximize over $y$?

(3) The algorithm proposed by (Gao et al., 2022) is called Hessian FR instead of Newton follow-the-ridge. In that algorithm, the update rule of $y$ is $y_{t+1}=y_t+\eta_{y1}\nabla_y f(x_t,y_t)-\eta_{y2}H_{yy}^{-1}\nabla_y  f(x_t,y_t)+\eta_x H_{yy}^{-1}H_{yx}\nabla_x  f(x_t,y_t)$. Therefore, shouldn't we use $\text{id}-\zeta\text{Hess}_y^{-1}f(z_k)$ to replace $-\zeta\text{Hess}_y^{-1}f(z_k)$ in your RNFR.

**Requested changes or answers (not critical for acceptance):**

(4) It seems that the global minimax point on the Riemann manifold is also proposed by your work, yes? If so, why not add that to the abstract, the research question, and the list of contributions in the Introduction?

(5) In Theorem 1, if $x$ and $y$ have different dimensionality, then $\text{grad}_{yx}^2 f(z)$ corresponds to a non-square matrix in Euclidean space and is thus not invertible so that LMiniMax=TSRGDA, right?

(6) Typo: In the final paragraph of related works, add ''s'' after the verbs, including ''generalizes'', ''completes'', ''extends''.

(7) Typo in Theorem 1: $\mu_1\ge \cdots\ge \mu_{d_y}>0$.

**Strengths And Weaknesses:**

**Strengths:** This is an excellent work that studies a nonconvex-nonconcave minimax optimization problem on Riemannian manifolds. This problem is challenging and under-explored but has important applications including robust Fréchet mean, projection robust Wasserstein distance, and orthonormal GAN. This work makes abundant and non-trivial novel extensions from Euclidean to Riemannian manifolds as summarized above in the ''Summary Of Contributions''.  The lit review, preliminary knowledge, novel notions, theories and experiments are presented clearly in a well-organized structure. Honestly, I am a novice in Riemann manifold. However, after reading this paper, I gain much knowledge about Riemann manifold. I did not check the proof but I can see that the proposed notions and theorems in this work are reasonable as they can reduce to what I already know on Euclidean space. Four (abundant) experiments cover both synthetic and application settings and the results are convincing.

**Weaknesses:** As a novice in Riemannian manifolds, I cannot find any major weaknesses of this paper. A few details could be clarified more, and some typos could be corrected, as I listed in the ''Requested Changes'' below.

---

> ### Author Response · Authors · 2023-05-15
> **Response to Reviewer yZgL**
>
> Thanks for your comments. Below are our responses to your comments.
>
> **(1)**. The discriminator aims to classify correctly the real samples from Gaussian mixture and incorrectly for the samples generated by the generator. The figure shows the prediction of discriminator on the entire 2d domain, and yes the pixel strength refers to the predicted probability.
>
> **(2)**. Because the sign cancels in the gradient and Hessian.
>
> **(3)**. Since we motivate our RNFR as approximation to RNGD, we choose not to add ${\rm id}$ to our formulation (where ${\rm id}$ does not appear in the Taylor approximation). Such a distinction has been mentioned in Section 3.1.2 of (Gao et al., 2022). The analysis does not differ too much with or without ${\rm id}$.
>
> **(4)**. That is correct. However, in this work, we focus on proper local optimality for nonconvex nonconcave problems and study algorithms converging to such local optimality. Hence we choose not to stress on global minimax point, which can be difficult to obtain.
>
> **(5)**. This is in fact a typo and is meant to be ${\rm Hess}_y f(z)$ not invertible. Sorry for the confusion.
>
> **(6) & (7)**. We have fixed the typos.

---

> ### Comment · Reviewer_yZgL · 2023-06-05
> **My concerns are not addressed.**
>
> Dear authors,
>
> My first 3 suggestions are not addressed in your revision.
> If you think my suggestions are not proper, you can explain in your reply.
> Thanks.
>
> Reviewer yZgL

---

> > ### Author Response · Authors · 2023-06-06
> > **Further reponses to Reviewer yZgL**
> >
> > Thanks for your comments. We thought it was sufficient to address your concerns in our response, which had been done before.
> >
> > As asked now, we have included the modifications regarding 1-3 concerns in the revised manuscript. Please check page 8 and page 13 (in the Caption of Figure 4).
> >
> > Please let us know if anything in particular required.

---

> > > ### Comment · Reviewer_yZgL · 2023-06-06
> > > **Reviewer yZgL is now satisfied**
> > >
> > > Thanks, authors.
> > >
> > > Sorry that the authors' response written on May 15 did not show up on my computer until just now. I'm not sure why.
> > > I just checked the response and the revision, and. now feel satisfied.
> > >
> > > Reviewer yZgL

---

### Review · Reviewer_z28z · 2023-05-07

**Summary Of Contributions:**

The paper focuses on solving nonconvex-nonconcave min-max optimization problems on Riemannian manifolds.

1. Section 4 provides the definitions and necessary conditions of local solutions and illustrates them in examples in Prop 4. This is an extension of previous results to Riemannian manifolds.

2. Section 5 first provides the definitions and characterization for strictly stable points for continuous and discrete dynamics, and proves the relationship between them. After that, it discussed the relationship between the limit point of TSRGDA and local minmax points and provides a local convergence rate.

3. Section 6 further considers several second-order methods, and provides a relationship between their stable fix points and local minmax points.

4. Section 7 conducts numerical experiments to compare the algorithms above.

**Audience:**

Yes

**Claims And Evidence:**

No

**Requested Changes:**

Please see the weakness part above.

**Strengths And Weaknesses:**

Strengths:

1. The paper includes a wide range of results, which provides a big picture for the topic.

2. The examples in experiments show the meaningfulness of such a setting, even though most results are extended from Euclidean space. I believe the setting may have more applications in the future.

Weakness: I have several major concerns about the paper.

1. I think the paper missed an important reference:

    [1] Tanner Fiez, Lillian J. Ratliff. Local Convergence Analysis of Gradient Descent Ascent with Finite Timescale Separation. ICLR 2021.

    [1] shows the local behavior GDA when the separation $\tau$ does not need to converge to 0, while similar results in [Jin et al., 2020] need $\tau \rightarrow 0$.

2. Regarding the main result of Theorem 1:

     a.  The complete proof of Theorem 1 is not presented in the paper. It states that the result of stable points follows from [Jin et al., 2020]. The theorem in the paper holds for a finite $\tau_0 > 0$. However, based on my understanding, Theorem 28 in [Jin et al., 2020] requires $\tau \rightarrow 0$.

     b. The second half of Theorem 1 states TSRGDA converges to local minmax points with a rate. However, as the stable equilibrium of TSRGDA is a superset of local minmax points, how can we guarantee that it only finds local minmax points? I also wonder why it does not depend on the initial point. If the algorithm starts from a point with derivatives 0 but not a local minmax point, will it stop there?

     c. I didn't follow what the $\delta$ is in this theorem, and how it affects the relationship between local minmax points and stable points.

     d. I suggest adding the requirement for $\eta$ in the theorem.

3. At the beginning of Section 6, it states that "without requiring a sufficiently small stepsize for the minimization variable as in TSRGDA".  I didn't follow this sentence. If Theorem 1 is correct, there exists $\tau>0$ for TSRGDA. In Theorem 2 and 3, there are also requirements for $\eta$ and $\tau$ (hidden in the proof. I suggest writing them in the statement of the theorems).

4. For experiments, I wonder why some algorithms stop earlier, e.g., the blue line in Figure 1(d). I also suggest using some different colors in the experiments, since some of them are similar to each other.

5. I also think it is necessary to compare with the following papers.

   [2] Constantinos Daskalakis, Stratis Skoulakis, Manolis Zampetakis: The Complexity of Constrained Min-Max Optimization. STOC 2021.

   [3] Gidel et al. Negative momentum for improved game dynamics. AISTATS 2019.

   [3] gives a bilinear minmax problem that GDA with any time separation might not converge. I wonder if such a pathological example is excluded by the assumption in this paper.

---

> ### Author Response · Authors · 2023-05-15
> **Response to Reviewer z28z**
>
> Thanks for your comments. Before we address the comments, we would like to highlight the changes we have made regarding your concerns on Theorem 1. In summary, we have restated Theorem 1 for better clarity and simplifies the convergence rate to avoid confusion. Nevertheless, the updated version of Theorem 1 does not negate the former version. Specifically, the changes are
>
> *  We separate out a new Proposition (Proposition 12) to formally call out the non-asymptotic statement.
>
> * We simplify the convergence rate analysis using eigenvalues of $\boldsymbol{\nabla} G_\tau(z^*)$ to avoid confusion over $\delta$.
>
> * We add a remark on the limiting case of the convergence rate.
>
> It should be noted that the restatement of Theorem 1 does not affect our other convergence results in Theorems 2, 3 for second-order methods.
>
> *1. I think the paper missed an important reference: [1] Tanner Fiez, Lillian J. Ratliff. Local Convergence Analysis of Gradient Descent Ascent with Finite Timescale Separation. ICLR 2021.*
>
> **Reply**: Thanks for your suggestion. We have now added this reference along with a modification of Theorem 1. This modification includes separating out a Proposition (Proposition 12) for non-asymptotic result and a simplification of the convergence rate analysis. It should be emphasized that the modification does not fundamentally change our Theorem 1. It should be also mentioned that our other results remain unaffected by this modification.
>
>
>
> *2. Regarding the main result of Theorem 1.*
>
> **Reply**:
>
> **a**. Our non-asymptotic result can be seen from the proof of Theorem 28 in (Jin et al. 2020), i.e., "we know there exists sufficiently small $\epsilon_0$ such that for any $\epsilon < \epsilon_0$" and "for sufficiently small $\epsilon$". Furthermore, the suggested reference [1] instrumentally characterizes such non-asymptotic result. For better clarity, we have, therefore, introduced a separate proposition (Proposition 12) to include the non-asymptotic result. Also, we restated the asymptotic result by only considering non-degenerate points, i.e., with ${\rm Hess}_y f(z)$ non-degenerate. This is because only such points are interesting for second-order methods as we introduce in Section 6.
>
> **b**. Theorem 1 states stable equilibrium of TSRGDA is equivalent to local minimax points up to some degenerate points. This appears to be the best we can achieve according to Jin et al. 2020 and also the new reference [1] that the reviewer has provided.
>
> **c**. We agree the introduction of $\delta$ here is confusing, and thus, we have removed it in the revision and restate the rate with eigenvalues of $\boldsymbol{\nabla} G_\tau(z)$ and only connects to eigenvalues $\lambda_i$, $\mu_j$ in the limiting case as shown in Remark 1.
>
> **d**. The requirement for $\eta$, i.e., $\eta < \min_i (-2 \mathcal{R}(\nu_i)/|\nu_i|^2)$ is now presented in Theorem 1.
>
> *3. At the beginning of Section 6, it states that "without requiring a sufficiently small stepsize for the minimization variable as in TSRGDA". I didn't follow this sentence...*
>
> **Reply**: Theorem 1 requires sufficiently small $\tau$, i.e., $\tau < \tau_0$. However, for Theorem 2 and 3, the choice of $\tau$ is arbitrary and there is no requirement on $\tau$. We have emphasized "for any $\tau > 0$" in Theorems 2 and 3. The requirement for $\eta$ is presented in Theorem 2 and 3 already.
>
> *4. For experiments, I wonder why some algorithms stop earlier, e.g., the blue line in Figure 1(d). I also suggest using some different colors in the experiments, since some of them are similar to each other.*
>
> **Reply**: This is because we set a maximum number of iterations for all algorithms (e.g., Figure 1(c), we set max iteration to be 100). We have added a statement before Section 7.1. Also, because RGDA requires less per-iteration computational cost, its requires less time for the same number of iterations (e.g., Figure 1(d)). Thanks for your suggestions on the color usage. We will definitely consider it when preparing the final submission.
>
> *5. I also think it is necessary to compare with the following papers [2], [3].*
>
> **Reply**: We have added reference to [2] in related work (Section 2). In summary, [2] considers approximate local min-max saddle points and linear convex constraints, which is different to what we consider in this work, which are local minimax points and general nonlinear nonconvex constraints. Hence, [2] is not applicable to our setting.
>
> For [3], we highlight that this pathological example in fact provides divergence results for GDA, while our paper studies what happens when GDA converges. Also, the work [3] focuses on bilinear problems, which are in general excluded from our work as our analysis mostly focuses on critical points/equilibrium with non-degenerate Hessian, ${\rm Hess}_y f(z)$.

---

> > ### Comment · Reviewer_z28z · 2023-06-12
> >
> > Thank you for the detailed reply.
> >
> > After the revision, Theorem 1 now makes more sense to me.

---

### Review · Reviewer_Aujm · 2023-05-31

**Summary Of Contributions:**

See below

**Audience:**

Yes

**Claims And Evidence:**

No

**Requested Changes:**

I have had the opportunity to go through the paper in question and found the concept to be quite interesting. However, I must express my concerns regarding the presentation of the work. From a reader's perspective, I found the paper's language and structure a bit challenging to navigate.

A particular issue I observed is related to the writing style. While the mathematics within the paper are correct, the lack of contextualization and intuition prior to each proof creates a barrier for the reader. To be able to appreciate the content fully, readers would greatly benefit from a more explanatory approach.

Moreover, a significant aspect that is missing from the paper is the visual representation of their algorithm. By incorporating geometrical aspects and some illustrative plots, the authors could make their work more digestible and didactic. This will not only enhance the readability of the paper but also engage the audience in a much more effective manner.

In my previous experiences with TMLR, the emphasis has been on mathematical accuracy as a sufficient condition for publication. While I understand the importance of correctness, I firmly believe that effective communication and accessibility of the work to the wider scientific community are equally crucial.

Thus, In my thorough perusal of your work, I have identified a few areas that I believe could be significantly improved to enhance the reader's understanding and the overall clarity of the manuscript.

Firstly, the inclusion of illustrative diagrams or plots which depict the geometrical facets of the algorithm would provide a substantial visual aid and support to the reader. This would not only clarify the working mechanism of the method but also aid in breaking the monotony of a text-intensive explanation. The addition of such visual elements is highly encouraged.

Secondly, it is imperative to introduce textual guidance or "foreshadowing" before each lemma in the appendix. This approach will provide a roadmap to readers on what to expect in the upcoming proofs. Without this navigational aid, readers might find themselves lost in a sea of symbols and equations, thereby failing to grasp the significance of the proofs. It's essential to remember that we're communicating a complex theory, not merely demonstrating algebraic prowess.

Lastly, I would like to see a future challenges section integrated into the manuscript. This section should consolidate observations from the experimental comparisons between different methods, delineating which aspects of these comparisons have been sufficiently elucidated by your theory or extant literature. More importantly, it should highlight which areas are yet to be adequately explained, posing them as open problems for future work. This would not only underline the current limitations of the study but also open new avenues for further research, demonstrating an appreciation for the ongoing evolution of knowledge in this area.

Remember, an academic manuscript should not only be a conduit of information but also a tool that educates and stimulates curiosity in the reader. Thus, adhering to these suggestions will significantly enhance your manuscript's pedagogical value and its scholarly impact.


**Strengths And Weaknesses:**

See Below

---

> ### Author Response · Authors · 2023-06-02
> **Response to Reviewer Aujm**
>
> Thanks for your comments and suggestions. Here are the responses to your comments.
>
> *1. On the inclusion of illustrative diagram of the algorithms proposed.*
>
> **Reply**: We have included a schematic figure (Figure 1), illustrating the iterative scheme in Riemannian optimization. This includes all the methods discussed in this work. The only difference is the way we compute the tangential vector $\xi$.
>
> *2. On textual or "foreshadowing" before each lemma in the appendix.*
>
> **Reply**: We believe you are referring to Section E Useful Results (revised manuscript) where we introduce new Lemmas for the proofs of the main results. For this, we have now added some textual guidance at the start of this section, introducing the purpose of the results. We have also included a separate section (Appendix A in the revised manuscript) outlining the structure of all Appendix sections for better clarity.
>
> In the main paper, we have also contextualized all the theorems.
>
> We will be happy to address any specific comment that you might have.
>
> *3. Introducing future challenges section.*
>
> **Reply**: Thanks for your suggestion. We have expanded the conclusion section (Section 8) and have included the future challenges. We also highlight some future directions.

---

### Decision · Action_Editors · 2023-08-20

**Recommendation:** Accept as is

**Comment:**

This paper focuses on solving min-max optimization problems on Riemannian manifolds, where the objective function is generally nonconvex and nonconcave in both the Euclidean and geodesic contexts. The authors (i) provide definitions of local optimality for such nonconvex-nonconcave min-max problems on Riemannian manifolds, (ii) establish stability and (local) asymptotic convergence rates for both continuous and discrete dynamical systems on these manifolds, and (iii) propose second-order methods that are provably guaranteed to asymptotically converge to local minimax points. The results are supported by detailed proofs and experiments involving a variety of nonconvex-nonconcave problems.

Nonconvex-nonconcave min-max optimization problems on Riemannian manifolds can be challenging and have remained relatively under-explored. This study introduces novel extensions from the Euclidean to Riemannian manifolds, as elaborated above. The reviewers have acknowledged these contributions and have either recommended acceptance or are leaning toward acceptance. I concur with the reviewers' assessment and recommend accepting.

**Audience:**

The results could potentially attract a wide range of interest from researchers working on topics such as generative adversarial networks, adversarial training, optimal transport, and more.

**Claims And Evidence:**

This paper focuses on solving min-max optimization problems on Riemannian manifolds, where the objective function is generally nonconvex and nonconcave in both the Euclidean and geodesic contexts. The authors (i) provide definitions of local optimality for such nonconvex-nonconcave min-max problems on Riemannian manifolds, (ii) establish stability and (local) asymptotic convergence rates for both continuous and discrete dynamical systems on these manifolds, and (iii) propose second-order methods that are provably guaranteed to asymptotically converge to local minimax points. The results are supported by detailed proofs and experiments involving a variety of nonconvex-nonconcave problems. Consequently, the theoretical findings and claims are substantiated by accurate, convincing, and clear evidence.

---

> ### Author Response · Authors · 2023-08-26
> **Uploaded Camera Ready Version**
>
> Dear action editors and reviewers,
>
> We appreciate the time and effort in reviewing our paper. We have now uploaded the camera ready version of our paper.
>
> Best, Authors